# Neutral ceramidase regulates breast cancer progression by metabolic programming of TREM2-associated macrophages

Rui Sun[1,2,3,8], Chao Lei[1,2,8], Zhishan Xu[1,2], Xuemei Gu[2], Liu Huang[4], Liang Chen[1,2], Yi Tan [5], Min Peng[3], Kavitha Yaddanapudi[1,2], Leah Siskind [6], Maiying Kong[2,7], Robert Mitchell [1,2], Jun Yan [1,2] & Zhongbin Deng [1,2] ✉

The tumor microenvironment is reprogrammed by cancer cells and participates in all stages of tumor progression. Neutral ceramidase is a key regulator of ceramide, the central intermediate in sphingolipid metabolism. The contribution of neutral ceramidase to the reprogramming of the tumor microenvironment is not well understood. Here, we find that deletion of neutral ceramidase in multiple breast cancer models in female mice accelerates tumor growth. Our result show that Ly6C⁺CD39⁺ tumor-infiltrating CD8 T cells are enriched in the tumor microenvironment and display an exhausted phenotype. Deletion of myeloid neutral ceramidase in vivo and in vitro induces exhaustion in tumor-infiltrating Ly6C⁺CD39⁺CD8⁺ T cells. Mechanistically, myeloid neutral ceramidase is required for the generation of lipid droplets and for the induction of lipolysis, which generate fatty acids for fatty-acid oxidation and orchestrate macrophage metabolism. Metabolite ceramide leads to reprogramming of macrophages toward immune suppressive TREM2⁺ tumor associated macrophages, which promote CD8 T cells exhaustion.

Tumor-associated macrophages (TAMs) play an important role in cancer development[1]. In breast cancer, macrophages within the tumor microenvironment (TME) correlate with poor clinical outcome[2]. TAMs play pro-tumorigenic roles by promoting angiogenesis and enhancing tumor cell invasion and metastasis[3]. TAMs also have the potential to phagocytose large amounts of tumor-associated antigens, but fail to successfully support T cell activation, thereby preventing tumor cell destruction by T lymphocytes[4,5]. CD8⁺ T cells that infiltrate into the TME are critical mediators of antitumor immune responses. However, these CD8⁺ T cells are frequently non-functional due to their exhausted state, characterized by the expression of inhibitory molecules, including PD-1, LAG3, CD39, and TOX, and the loss of cytotoxic

effector function[6–8]. Previous studies found that chronic antigen exposure and stimulation of the T cell receptor (TCR) are required for an exhaustion program in T cells[9,10]. Intravital imaging studies have shown that antigen-specific CD8⁺ T cells preferentially localize in TAM-rich areas in the TME[11,12]. How the operational network between TAMs and exhausted CD8 T cells is orchestrated in the TME remains unclear.

Microenvironmental signals including lipids play a crucial role in leading polarized programs of macrophages. TAMs have distinct metabolic phenotypes that differ from resting macrophages[13,14]. Previous studies showed TAMs from both human and murine tumor tissues were enriched with lipids due to increased lipid uptake by macrophages and use of fatty-acid oxidation (FAO) to fuel

[1]Department of Surgery, Division of Immunotherapy, University of Louisville, Louisville, KY, USA. [2]Brown Cancer Center, University of Louisville, Louisville, KY KY40202, USA. [3]Cancer Center, Renmin Hospital of Wuhan University, Wuhan, Hubei 430060, P. R. China. [4]Department of Oncology, Tongji Hospital, Tongji Medical College, Huazhong University of Science and Technology, Wuhan 430030 Hubei, P.R. China. [5]Department of Pediatrics and Pediatric Research Institute, University of Louisville, Louisville, KY, USA. [6]Department of Pharmacology & Toxicology, University of Louisville, Louisville, KY 40202, USA. [7]Department of Bioinformatics and Biostatistics, University of Louisville, Louisville, KY, USA. [8]These authors contributed equally: Rui Sun, Chao Lei. ✉e-mail: z0deng01@louisville.edu

mitochondrial oxidative phosphorylation[15]. Tumors depend on increased lipid synthesis to meet their needs for energy and structural building blocks to enable replication[16]. Moreover, lipophilic signaling molecules such as sphingolipids have been shown to promote various aspects of tumorigenesis[17]. For example, an elevated production of glucosylceramides has been reported in breast cancer[18]. Ceramide is the central molecule in sphingolipid metabolism and can be hydrolyzed by neutral ceramidase (NcDase) to sphingosine[18]. We recently reported that NcDase metabolite sphingosine directs macrophage glycolytic metabolism in intestinal macrophage activation, which prevents pathogen colonization of the gut[19]. Notably, ceramide has been shown to play a critical role in the development and functionality of myeloid cell types[20,21]. However, the extent to which NcDase function regulates TAM-T cell crosstalk within the TME is not known.

In this work, we show that myeloid NcDase deficiency can facilitate the metabolism polarization of macrophages toward an immune-suppressive state in the TME, an action that is characterized by an enhanced ability to promote tumor growth and, in an indirect manner, induce the exhaustion of CD8+ T cells. Deletion of NcDase induces the generation of lipid droplets and promotes lipolysis, which serves to generate fatty acids for fatty-acid oxidation (FAO) in M2-like macrophages[22]. We further provide evidence that NcDase deficiency or ceramide accumulation is responsible for the enriched expression of transmembrane protein triggering receptor expressed on myeloid cells 2 (TREM2) of TAMs. TREM2 is an immune suppressive receptor on lipid-associated macrophages (LAMs) in tumors[23–25]. Importantly, ceramide induced TREM2+ TAMs are associated with T cell exhaustion in breast cancer.

## Results

### NcDase is associated with immune regulation of cancers

To investigate the role of NcDase in the development of breast cancer, we induced mammary carcinogenesis in wild-type (WT) and NcDase−/− mice by using the MMTV-PyMT transgenic model. NcDase−/− PyMT mice showed significant earlier onset of breast tumors (Fig. 1a), increased tumor growth and tumor weight (Fig. 1b, c and Supplementary Fig. 1a). To demonstrate the relevance of NcDase in human cancer clinical data, we used the TIMER 2.0[26], QUANTISEQ[27,28] and CIBERSORTER web[29] resources to analyze the association between NcDase expression and TME composition in public clinical breast cancer data sets for four subtypes of BRCA (Breast invasive carcinoma) including luminal A (LumA), luminal B (LumB), Her2-enriched (Her2), and basal-like (Basal). TIMER 2.0 analysis showed that NcDase mRNA expression (encoded by *ASAH2*) was correlated with CD8+ T cells and macrophages in the LumA subtype (Fig. 1d). In inflamed cancer types including breast cancer, NcDase expression correlated strongly with the expression of the markers of immune regulation (CTLA4, LAG3, PDCD1, SIGLEC15) (Fig. 1e). In addition, the activity of NcDase in mouse and human macrophage was significantly inhibited when co-cultured with tumor tissue conditioned medium (TCM) (Fig. 1f), indicating NcDase expression in TAMs is downregulated in TME. However, the TIMER analysis showed that expression of NcDase mRNA in breast cancer was not correlated with tumor cell purity and other types of immune cells (Supplementary Fig. 1b). Furthermore, the survival analysis by CIBERSORTER[29] showed that expression of NcDase mRNA was not correlated with overall survival in four subtypes of breast cancer patients (Supplementary Fig. 1c). No difference of NcDase mRNA (*ASAH2*) expression was observed between normal breast tissues and tumor tissues or between subclass of breast cancer (Supplementary Fig. 1d). Together, these results suggested that NcDase might be involved in the regulation of anticancer immunity.

To directly interrogate the functions of NcDase in the TME, we evaluated global immune profiling of the tumors using cytometry by time-of-flight (CyTOF)[30], FlowSOM[31] and (tSNE)/UMAP[32] algorithms, which identified a total of 20 immune cell subtypes (Fig. 2a). The

effector CD8 T cells were decreased in the NcDase−/− PyMT tumors (Fig. 2b, c, CD8+ T2 cluster). We also observed decreased levels of M1-like macrophages and increased levels of M2-like macrophages in NcDase−/− PyMT tumors compared with WT tumors (Fig. 2b, c, M1 mac3 and M2 mac2). A more focused analysis of the CyTOF data set on 8 subtypes within the T cell population showed the subtypes that are changed (Fig. 2d, e, and Supplementary Fig. 2a). Of note, when Ly6C expression was paired with CD39 expression, CyTOF profiling revealed that among CD8+ tumor infiltrating T lymphocytes (TIL), CD8+Ly6C+CD39- cells (CD8+ T1) were decreased to the greatest extent in the tumors from NcDase−/− PyMT versus WT PyMT mice. We observed that NcDase−/− PyMT tumors were infiltrated with a greater abundance of CD8 T cells expressing Ly6C−CD39+ (CD8+ T3 and T4) with exhausted markers (Tim-3+, PD-1+ and LAG3+) (Fig. 2d, e).

### NcDase regulates the exhaustion of cytotoxic Ly6C+CD8+ T cell in TME

Using CyTOF (Supplementary Fig. 2b, c) and flow cytometry (Fig. 2f and Supplementary Fig. 2d) analysis, four distinct subsets of CD8+ T cells could be distinguished in direct association with Ly6C and CD39 expression in TIL as: (1) Ly6C+CD39− cells (Ly6C single positive, SP), (2) Ly6C−CD39+ cells (CD39 SP), (3) Ly6C+CD39+ cells (Double positive, DP) and (4) Ly6C−CD39− cells (Double negative, DN). Compared to WT PyMT mice, the most different subset in NcDase−/− PyMT mice was the Ly6C+CD39+ CD8+ DP T cells relative to the Ly6C+CD39− T cells (Fig. 2f and Supplementary Fig. 2c), Ly6C−CD39+ CD8+ T cells expressed the highest amounts of TOX, PD-1 and CD244.2 and similar levels of 4-1BB, ICOS and TIGIT compared to other subsets (Fig. 2g and Supplementary Fig. 2e). Importantly, the expression of CD39, a marker for exhausted T cells[33,34], is only induced in tumor CD8+ T cell, during tumor growth in WT mice (Supplementary Fig. 2f), and even higher levels in tumor CD8+ T cell from NcDase−/− mice (Supplementary Fig. 2f), but not in CD8 T cells from other tissues. Moreover, when combined with Ly6C+, but not with Ly6C−, we found that the CD44+CD62L− CD8+ T effector cells were significantly decreased in NcDase−/− PyMT mice compared to WT PyMT mice (Supplementary Fig. 2g).

In TIL of WT mice, the highest IFN-γ-expressing CD8+ T cells were Ly6C+CD39− cells (6 C SP) that expressed the lowest amounts of PD-1 and TOX compared to the other three subsets (Fig. 2g), indicating they are cytotoxic CD8 T cells. Importantly, an increase in the percent of Ly6C+CD39−CD8+ T cells correlated significantly with a decreased tumor burden (Fig. 2h). We also found that Ly6C+CD8+ T cells expressed higher amounts of IFN-γ compared to Ly6C− CD8+ T cells in the spleen, neck LN, liver, blood, lung, and intestine in naïve B6 mice (Supplementary Fig. 2h). Of note, Ly6C+CD39+ T cells (DP) were the exhausted CD8 T cells because they expressed intermediate amounts of canonical exhausted markers IFN- γ, TOX and PD-1, however, a lower level of TNF-α (Fig. 2g). These results demonstrate that IFN-γ is highly expressed in Ly6C expressing CD8+ T cell. Conversely, Ly6C−CD39+ T cells (CD39 SP) have the lowest amounts of IFN-γ and TNF-α along with the highest levels of exhaustion markers. We next checked if these CD8+ subsets from two models would have different functions. Significant decreases in both IFN-γ-expressing and TNF-α-expressing Ly6C+CD39−CD8 cytotoxic T cell subsets were found in the tumors from NcDase−/− PyMT mice compared to WT PyMT mice (Fig. 2g and Supplementary Fig. 2i). The TIL of NcDase−/− PyMT mice have increased exhausted Ly6C+CD39+CD8+ T cells that express intermediate amounts of IFN-γ and produced the lowest level of TNF-α (Fig. 2g). Ly6C+CD39+CD8+ exhausted cells from NcDase−/− PyMT mice expressed lower levels of Ki-67 (Supplementary Fig. 2j). A larger fraction of exhausted Ly6C+CD39+CD8+ T cells from NcDase−/− PyMT mice stained with the apoptosis marker annexin V (Supplementary Fig. 2k). However, exhausted CD8+ T cells from NcDase−/− PyMT mice expressed similar granzyme B as did exhausted cells from WT PyMT mice (Supplementary Fig. 2e). Collectively, these data show that the Ly6C+CD8+ T

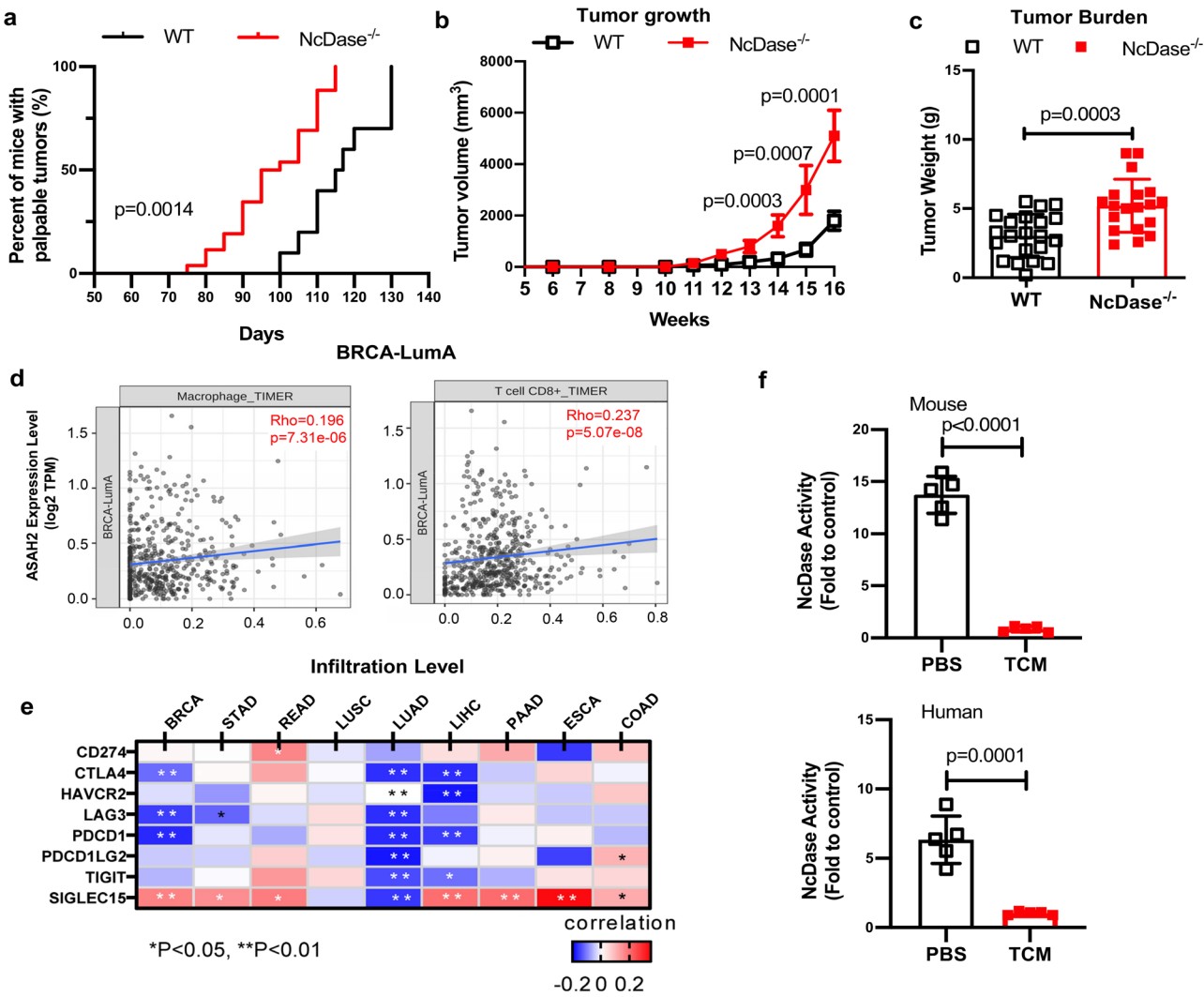

**Fig. 1 | Deletion of NcDase in the MMTV-PyMT model promotes tumor progression. a–c** Kaplan-Meier curves for tumor onsets at indicated times (**a**), tumor growth (**b**), and final tumor burden (**c**) in WT PyMT mice and NcDase$^{-/-}$ PyMT mice. $n = 21$ mice (WT) or 19 mice (NcDase$^{-/-}$). **d** Correlation of NcDase expression (*ASAH2*) with infiltration level of indicated cell types in breast cancer LumA subtypes. Data were from the TIMER 2.0 web platform ($n = 568$). Spearman's correlation coefficients and P values are shown. TPM transcript count per million reads. **e** Correlation between NcDase and immune regulatory markers across nine cancer types from TCGA are shown as a heatmap. BRCA: Breast invasive carcinoma ($n = 1098$); STAD: Stomach adenocarcinoma ($n = 443$); READ: Rectum adenocarcinoma ($n = 171$); LUSC: lung squamous cell carcinoma ($n = 504$); LUAD: Lung adenocarcinoma ($n = 584$); LIHC: Liver hepatocellular carcinoma ($n = 377$); PAAD pancreatic ductal adenocarcinoma ($n = 185$); ESCA: Esophageal carcinoma ($n = 185$); and COAD colon adenocarcinoma ($n = 460$). **f** The activity of NcDase in Raw264.7 macrophages and THP-1 derived macrophages that were co-cultured with mouse or human breast cancer conditioned medium for 48 h, respectively. $n = 5$ independent biological experiments. Statistical comparisons were performed using log-rank test (**a**), two-tailed unpaired $t$-test (**b**, **c**, **f**), Pearson's correlation coefficient (PCC) (**d**, **e**), Error bars indicate mean ± SD. Source data and exact $p$ values (**e**) are provided as a Source Data file.

cell population is heterogeneous and functionally diverse, which is divided by CD39 expression that is partially regulated by NcDase in the TME.

## Antitumor efficacy of CD8$^+$ T cells is dependent on myeloid NcDase

Since NcDase is associated to tumor-infiltrating CD8$^+$ T cells and macrophages, we investigated the spatial distribution of 18 immune cell markers and took a more detailed look at the interactions between TAMs and CD8$^+$ T cells within the TME in breast tumors by using imaging mass cytometry[35]. Ly6C$^+$ monocytes were mostly located near CD31$^+$ vessels regardless of the cancer genotype (Supplementary Fig. 3a, b). Most of the colocalizations of MHCII$^+$ M1-like macrophages with CD8$^+$ T cells in WT PyMT mice were in the tumor core, whereas colocalizations of CD206$^+$ M2-like macrophages with CD8$^+$ T cells in NcDase$^{-/-}$ PyMT mice were largely located in the tumor margin (Fig. 3a, b). When comparing

the distribution of M1/M2-like macrophages infiltrating into the tumor core and in the invasive margin, tumors in the NcDase$^{-/-}$ mice displayed significantly lower M1-like macrophages in the tumor core and margin, and higher M2-like macrophages in the margin when compared with WT mice (Fig. 3c). These data indicate that CD8$^+$ T cells and TAMs are distributed in a spatially different manner within the TME when comparing WT and NcDase$^{-/-}$ mice.

To further define the role of NcDase in anticancer immune populations, we injected tumor cells derived from a MMTV-PyMT tumor orthotopically into the mammary gland of young NcDase$^{-/-}$ PyMT mice and WT PyMT control mice without spontaneous tumors (Fig. 3d). This model allows for the assessment of the contribution of NcDase signaling in the TME to cancer progression because PyMT cancer cells express NcDase and NcDase was only depleted in the TME. Depletion of NcDase in the TME resulted in accelerated tumor onset and tumor growth (Fig. 3e–g). Orthotopic tumors from NcDase$^{-/-}$ PyMT

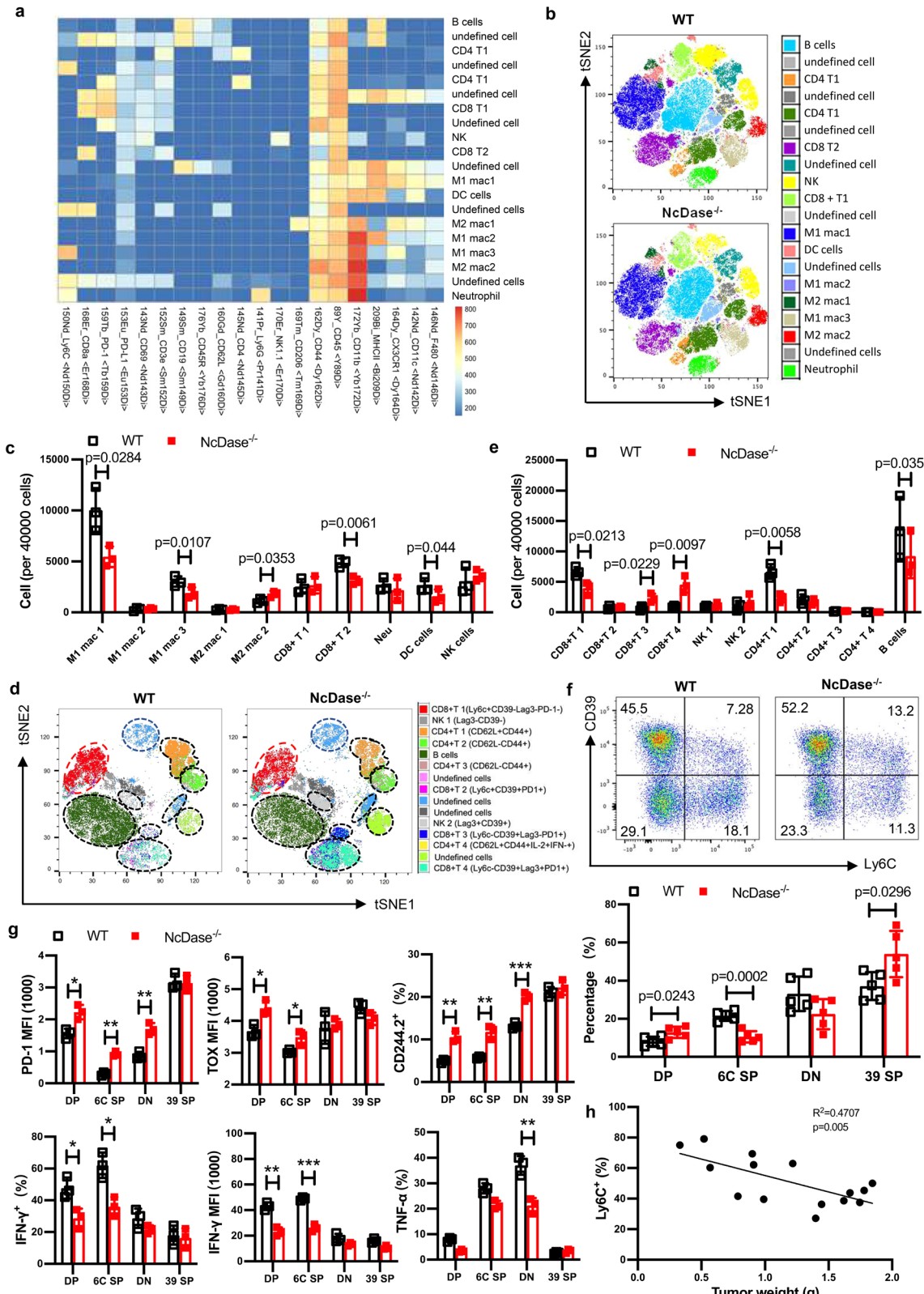

mice were marked by a strong infiltration of both MHC II^low CD206^+ TAM M2-like macrophages and Ly6C^−CD39^+ and CD244.2^+ CD39^+ exhausted CD8^+ T cells into the subcutaneous tumor (Fig. 3h). Conversely, NcDase depletion in cancer cells derived from NcDase^−/− PyMT mice did not show any effect on tumor growth and weight in WT animals compared to cancer cells derived from WT PyMT mice (Supplementary Fig. 3c–e), further emphasizing the role of NcDase in the TME.

To test the requirement for the NcDase-mediated adaptive or innate immune system in the growth of breast cancer, we deleted NcDase in Rag1-deficient mice (Rag1^−/− NcDase^−/−)[19]. Rag1^−/− mice and Rag1^−/−NcDase^−/− mice, which lack T and B cells, were orthotopically transplanted with MMTV-PyMT autochthonous tumor derived cells. In contrast to the immunocompetent mice, tumor growth in Rag1^−/− NcDase^−/− mice was similar to that in Rag1^−/− mice (Fig. 3i), indicating a

**Fig. 2 | Analysis of immune profiles within tumors based on CyTOF analysis.**
**a** CyTOF analysis based on FlowSOM clustering into 20 final major immune cell types for global immune profiling of the tumors is shown as a normalized expression heatmap. **b**, **c** Dot plot representation of t-distributed stochastic neighbor embedding (tSNE) analysis (**b**) and 40,000 cells analysis (**c**) showing different clusters that enable the distinction of immune populations in MMTV-PyMT tumors. **d**, **e** An in-depth T cell population analysis using FlowSOM clustering annotated into 8 final T cell subtypes is shown as (tSNE) analysis (**d**) and 40,000 cells - analysis (**e**). **a**–**e** n = 3 independent biological samples. **f**, **g** CD8 T cells from MMTV-PyMT tumors were analyzed by flow cytometry for expression of the indicated populations: CD39$^+$Ly6C$^-$ (39 SP), CD39$^+$Ly6C$^+$ (DP), CD39$^-$Ly6C$^-$ (DN), and CD39$^-$Ly6C$^+$ (6C SP) (**f**); The average MFI (median fluorescence intensity) or percentage ± SD for the levels of various proteins in CD8$^+$ T cell subpopulations are shown in bar graphs (**g**). n = 5 (**f**) or 3 (**g**) independent biological samples. *p < 0.05; **p < 0.01, ***p < 0.001. **h** Tumor weight (**g**) was correlated with the percentage of Ly6C$^+$ CD39$^-$CD8$^+$ T cells in mouse breast cancer. n = 15 independent biological samples. Statistical comparisons were performed using two-way ANOVA with Sidak's multiple comparisons test (**c**, **e**, **f**, **g**), and two-tailed Pearson's r correlation (**h**). Error bars indicate mean ± SD. Source data and exact p values (**g**) are provided as a Source Data file.

tumor-specific T cell response is required for NcDase-mediated immune regulation. To understand whether the promotive effects incurred by NcDase abrogation are dependent on crosstalk between macrophages and T cells, we compared the tumor growth in Rag1$^{-/-}$ mice and Rag1$^{-/-}$NcDase$^{-/-}$mice upon the transfer of CD8$^+$ T cells isolated from WT mice (designated as Rag1$^{-/-}$ CD8$^{WT}$ and Rag1$^{-/-}$NcDase$^{-/-}$ CD8$^{WT}$, respectively) or CD8$^+$ T cells isolated from NcDase$^{-/-}$ mice (designated as Rag1$^{-/-}$ CD8$^{NCKO}$ and Rag1$^{-/-}$NcDase$^{-/-}$CD8$^{NCKO}$, respectively). As predicted, in the presence of CD8 T cells, tumor growth was significantly delayed (Fig. 3i). Rag1$^{-/-}$ mice receiving transferred WT CD8$^+$ T cells had a more decreased tumor growth rate (Fig. 3i) and exhausted T cells (Fig. 3j) compared to Rag1$^{-/-}$NcDase$^{-/-}$ mice receiving transferred WT CD8$^+$ T cells. In addition, both the tumor growth rate and exhausted T cells in Rag1$^{-/-}$NcDase$^{-/-}$mice receiving NcDase deficient CD8$^+$ T cells (Rag1$^{-/-}$NcDase$^{-/-}$CD8$^{NCKO}$) was also higher than in the Rag1$^{-/-}$ mice receiving NcDase deficient CD8$^+$ T cells (Supplementary Fig. 3f, g), suggesting there is a promotive effect in Rag1$^{-/-}$NcDase$^{-/-}$ mice regardless of the status of NcDase in CD8$^+$ T cells. We next applied clodronate encapsulated liposomes (CELs) treatment to deplete macrophages in tumor-bearing WT Rag1$^{-/-}$ mice and Rag1$^{-/-}$NcDase$^{-/-}$ mice to examine whether macrophages are responsible for the differential tumor growth rates after CD8$^+$ T cells transfer. Rag1$^{-/-}$ mice and Rag1$^{-/-}$NcDase$^{-/-}$ mice displayed similar tumor growth rates and tumor burdens upon treatment with CELs (Supplementary Fig. 3f), indicating that myeloid NcDase deficiency is required for promoting tumor progression by supporting accumulation and survival of TAMs, probably through CD8$^+$ T cells.

## NcDase regulates the function of tumor-associated macrophages

Given that the exhausted Ly6C$^+$CD39$^+$ CD8 T DP subset was found only at sites where tumor cells were present, we wanted to explore the effect of macrophage NcDase on the CD39 upregulation on CD8 T cells. Bone marrow progenitor cells were differentiated into macrophages in the presence of macrophage-CSF (M-CSF). Following differentiation, we polarized them to an M2 state by the addition of IL-4 or treating the macrophages with either normal media or tumor tissue-conditioned media (TCM) from breast cancer tissue. The addition of TCM allowed us to partially mimic the TME, where tumor-derived factors, like prostaglandins, soluble CD44 and succinate, influence the differentiation and polarization of macrophages[36–38]. TCM treatment induced more activity of arginase and higher mRNA levels of TAM markers such as *Fizz1*, *Arg1* and *Ym-1* mRNA in NcDase deficient bone marrow derived macrophages (BMDMs) compared to WT BMDMs (Supplementary Fig. 4a, b). We subsequently co-cultured for 36 h the polarized macrophages with sub-optimally activated T cells (CD3/CD28 and IL-2) isolated from spleens of non-tumor-bearing mice. Flow cytometric analysis revealed that CD8$^+$ T cells co-incubated with PBS-treated macrophages (M0) in normal media had the upregulated expression of CD39$^+$ from Ly6C$^+$CD39$^-$ cells compared to T cells only, regardless of the genotype of the BMDMs (Fig. 4a). Importantly, when T cells were co-incubated with M2-like macrophages differentiated in TCM or IL-4, they display a more increased expression of CD39 compared with expression in T cells that were co-incubated with WT M0

macrophages (Fig. 4a), and an even more enhanced increase in T cells co-incubated with NcDase deficient M2-like macrophages (Fig. 4a). Furthermore, naïve or polarized macrophages can modulate CD39 expression in CD8$^+$ T cells through cell-cell contact-dependent mechanisms (Fig. 4b). To explore if the molecular/signal in T cells mediates the expression of CD39, we added the anti-PD-1 mAb in the co-culture system. PD-1 mAb treatment significantly suppressed the levels of CD39 and Lag3 in CD8$^+$ T cells induced by M2-like macrophages derived from both of genotypes. However, the level of CD39 in CD8$^+$ T cells was highly induced by NcDase deficient M2-like macrophages compared to WT M2-like macrophages after PD-1 mAb treatment (Supplementary Fig. 4c, d). This data indicates that myeloid NcDase provided additional signal contributing to CD8 exhaustion. Of note, IFN-γ in Ly6C$^+$CD39$^+$T cells was significantly lower upon exposure to macrophages cultured in TCM or IL-4 compared to those in Ly6C$^+$CD39$^-$ T cells (Fig. 4c). We also observed a significant increase in the expression of LAG3 and TOX in the Ly6C$^+$CD39$^+$ T cells compared to Ly6C$^+$CD39$^-$ T cells (Fig. 4d), indicating M2-like macrophages prefer to induce Ly6C$^+$CD39$^+$ exhausted T cells. Importantly, a significant increase was found in the exhaustion (LAG3 and TOX) of T cells when they were incubated with IL-4- or TCM-treated BMDMs derived from NcDase$^{-/-}$ mice when compared with those from WT mice (Fig. 4d). Similar increases of CD39$^+$ and LAG3$^+$ in Ly6C$^+$CD8$^+$T cells were found when the Ly6C$^+$CD8$^+$T cells were incubated with TAMs sorted from NcDase$^{-/-}$ PyMT tumor compared to the T cells incubated with TAMs from WT PyMT tumor (Fig. 4e). In addition, macrophages were co-incubated for 72 h with CFSE-stained and sub-optimally activated T cells isolated from non-tumor-bearing mouse spleens. Quantification of CFSE dilution revealed a marked attenuation of T cell proliferation after incubation with M2 BMDMs from NcDase$^{-/-}$ mice compared with T cells after incubation with M2 BMDMs from WT mice (Fig. 4f). We also observed significant decreases in the proliferation (shown by CFSE$^{low/-}$) and cytotoxicity (IFN-γ) of T cells when T cells were co-incubated with M2 BMDMs compared M0 BMDMs, with an even greater reduction in T cells co-incubated with NcDase$^{-/-}$ M2 BMDMs compared to WT M2 BMDMs (Fig. 4f, g). These results indicate that NcDase deficiency induces an immune-suppressive phenotype in tumor-conditioned macrophages, which in turn suppressed the cytotoxic capabilities of CD8 T cells and induced the exhaustion of CD8 T cells.

## NcDase deficiency promotes lipid droplet accumulation and fatty acid oxidation in TAMs

Using the PyMT breast tumor model, we compared transcriptional profiles of sorted TAMs by RNA sequencing (RNA-seq). A fundamental difference in gene expression was found in TAMs when comparing NcDase$^{-/-}$ PyMT mice and WT PyMT control mice (Fig. 5a). Metabolism associated genes were found among the differential genes in WT TAMs versus NcDase$^{-/-}$ TAMs (Supplementary Fig. 5a, b). Further inspection of the genes involved in the primary lipid metabolic pathways revealed that a proportion of genes were associated with lipid droplet (LD) formation (e.g. *Hilpda*, *TREM2*[39]) and LD degradation (e.g. *ATGL*, lipolysis C [*Lip-C*]) (Fig. 5a), which are expressed at higher levels in NcDase$^{-/-}$ TAMs in comparison to WT

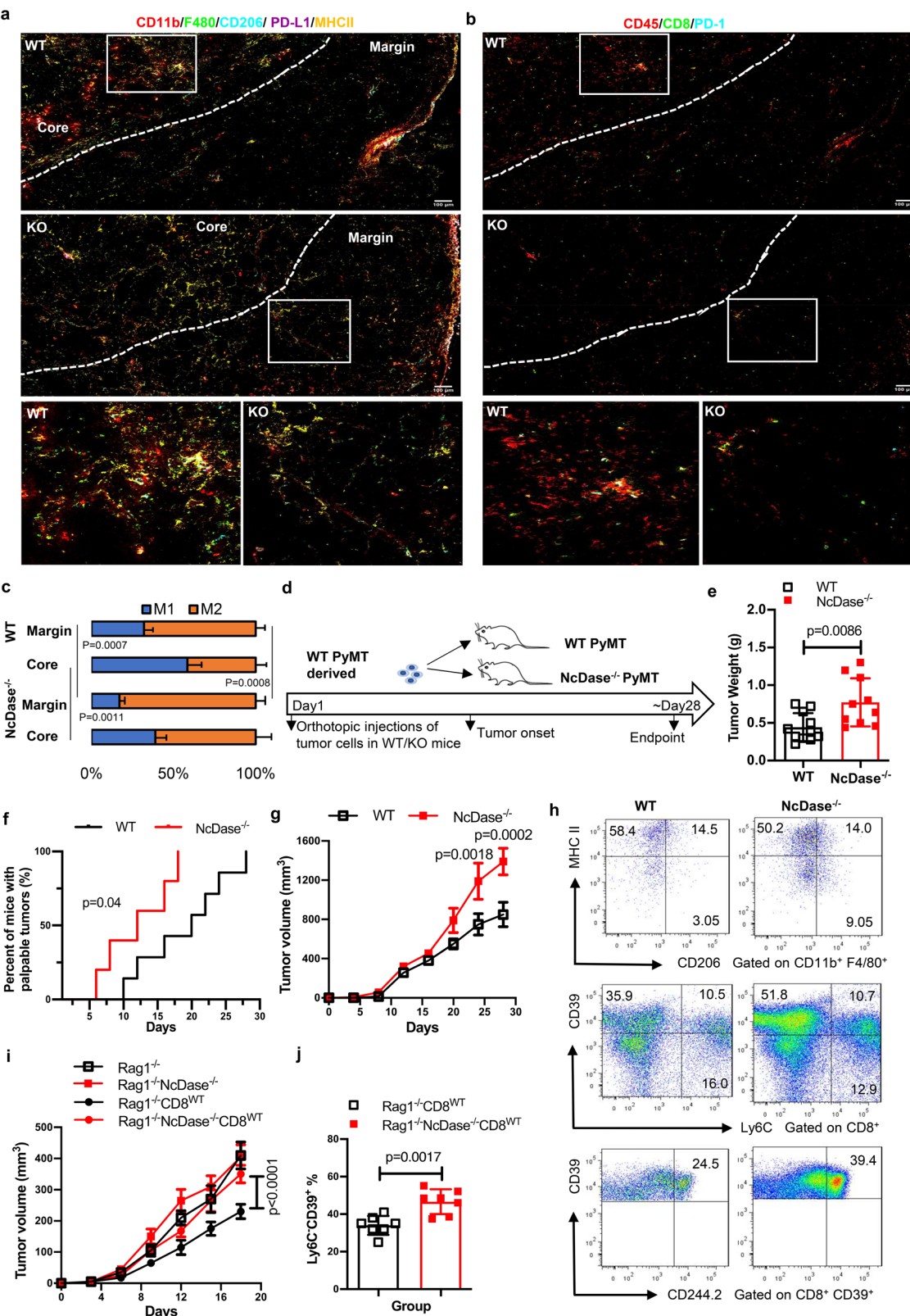

TAMs (Supplementary Fig. 5b). Specifically, real-time PCR confirmed that the expression of *Dgat2*, the enzyme responsible for synthesizing TGs from DGs, and *Gpat3*, which encodes the enzyme which catalyzes the conversion of glycerol-3-phosphate to lysophosphatidic acid in the synthesis of TG, increased in NcDase[-/-] M2 BMDMs (Fig. 5b), but that *Dgat1* was expressed similarly (Supplementary Fig. 5c). Fatty acids for FAO are released from triacylglycerols stored in lipid droplets with cholesterol esters by a process of lipolysis that is initiated by ATGL and is continued by hormone-sensitive lipase[40,41]. Among genes encoding lipases, we confirmed that *Atgl* and *Lip-C* (Fig. 5b), but not *Lip-E* and *Lip-G* (Supplementary Fig. 5c), had higher expression in NcDase[-/-] M2 BMDM compared to WT M2 BMDMs. These data indicate both LD formation and lipolysis are increased in NcDase[-/-] TAMs.

**Fig. 3 | Antitumor efficacy of CD8⁺ T cells is dependent on myeloid NcDase.**
**a, b** Representative IMC images of three independently stained tumor samples with the protein panel colored by different markers. A magnification of the indicated region (white box) from images of WT PyMT and NcDase⁻/⁻ PyMT mice is shown on the bottom. Marker expression was false colored, and markers are indicated above each plot. A Gaussian blur (sigma = 0.65) was applied. Scale bars, 100 µm. **c** Tissue distribution for types of macrophages differs between WT PyMT and NcDase⁻/⁻ PyMT mice. Stacked bar graph showing the distribution of the two macrophage types within the tissue domains. Type 1 macrophages (MHCII⁺) are found mostly in the tumor core domain in WT PyMT mice. Type 2 macrophages (CD206⁺) are found mostly in the margin domain in NcDase⁻/⁻ PyMT mice. The number means for the averages of proportions in MHCII⁺F4/80⁺ cells or CD206⁺F4/80⁺ cells per regions of interest (ROI). $n = 3$ images of obtained from three mice. **d** Experimental setup of the in vivo experiment. Isolated WT PyMT tumor cells were orthotopically injected into the mammary fat pad of WT PyMT mice or NcDase⁻/⁻ PyMT mice without tumor at 7 weeks of age. **e–g** Tumor weight (**e**), kaplan-Meier curves for tumor-onset (**f**) and tumor growth (**g**) described in (**d**). $n = 11$ mice (WT) or 10 mice (NcDase⁻/⁻). **h** Flow cytometry analysis of the expression of CD206/MHCII in CD11b⁺/F4/80⁺ cells; Ly6C/CD39 in CD8⁺ T cells, or CD244.2/CD39 in CD8⁺ T cells. Representative of three independent samples. **i, j** Isolated PyMT WT tumor cells were orthotopically injected into the mammary fat pad of Rag1⁻/⁻ mice or Rag1⁻/⁻ NcDase⁻/⁻ mice that reconstituted with/without $1 \times 10^7$ CD8⁺ T cells from WT mice (T cells were named CD8^WT). Tumor cell growth (**i**) and percentage of Ly6C⁻CD39⁺ CD8⁺ T cells in tumors was analyzed (**j**). $n = 7$ mice per group. Two-way ANOVA with Sidak's multiple comparisons test (**c, i**), two-tailed unpaired $t$-test (**e, g, j**), long-rank test (**f**), were performed. Error bars indicate mean ± SD. Source data are provided as a Source Data file.

Using the lipophilic fluorescent dye BODIPY 493/503, we observed an increase of lipid droplets in F4/80⁺ macrophages in tumor tissue sections (Fig. 5c) and in isolated CD11b⁺ F4/80⁺ macrophages (Fig. 5c, d) from tumor tissue of NcDase⁻/⁻ PyMT mice compared with those from WT PyMT mice. Lipid droplets contain a core of neutral lipid, namely triacylglycerol (TG) and cholesterol ester[42]. We did find that NcDase⁻/⁻ TAMs displayed higher ability to take up cholesterol, but not fatty acids, as evident by the staining of fluorescently labeled palmitate (BODIPY-FL-C₁₆) and cholesterol ester (BODIPY CholEsteryl FL-C₁₂), respectively (Fig. 5d). Next, we examined the accumulation of lipids in WT or NcDase deficient macrophages generated by in vitro culture of BMDMs with TCM for 72 h. These TCM co-cultured NcDase deficient BMDMs displayed a lipid increase compared with WT BMDMs (Fig. 5e). Using confocal microscopy with BODIPY 493/503 staining of neutral lipids, we showed that NcDase deficient BMDMs contained large bright green dot, which were lipid droplets, whereas WT BMDMs contained small, evenly distributed green smears (Fig. 5f and Supplementary Fig. 5d).

Lipolysis is a critical mechanism for the generation of fatty acids for FAO in macrophages and IL-4 drives lipolysis (from LD) for the M2-like activation of macrophages[22]. As expected, no significant difference was observed in lipid accumulation between WT BMDMs and NcDase⁻/⁻ BMDMs under IL-4 culture conditions (Fig. 5g). However, we did observe that more neutral lipids (detected by flow cytometry) (Fig. 5g) and lipid droplets (visualized by confocal microscopy) (Fig. 5h and Supplementary Fig. 5e) accumulated in NcDase⁻/⁻ BMDMs stimulated with IL-4 in the presence of tetrahydrolipistatin (orlistat), a clinically used active site-directed lipase inhibitor[43], compared to those in WT BMDMs. To analyze metabolic differences between TAMs from WT and NcDase⁻/⁻ mice, we measured the mitochondrial oxygen consumption rates (OCR) in sorted TAMs to determine whether the high level of FAO contributed to ATP production-linked oxidative phosphorylation in mitochondria. NcDase⁻/⁻ TAMs showed an enhanced mitochondrial OCRs, ATP production, and an increased spare respiratory capacity (the quantitative difference between the maximal OCR and the initial basal OCR; Fig. 5i, j). No difference in extracellular acidification rate (ECAR) was observed between WT and NcDase⁻/⁻ TAMs (Supplementary Fig. 5f). We also measured OCRs in control BMDMs and IL-4-treated or TCM-treated BMDMs from NcDase WT and KO mice. Our data showed that the basal respiratory rate and ATP production were significantly enhanced in NcDase⁻/⁻ IL-4-treated or TCM-treated BMDMs compared with WT BMDMs (Fig. 5k and Supplementary Fig. 5g). Taken together, these findings confirm our hypothesis that both high levels of lipid accumulation and increased lipolysis by NcDase deficiency promotes TAM FAO and oxidative phosphorylation to generate more energy for TAMs. Given the close link between FAO and M2 activation, we next assessed the effect of inhibiting lipolysis on the coculture of TAMs and CD8 T cells. As expected, TAMs from NcDase⁻/⁻ mice exhibited potent immunosuppressive activity compared to TAMs from WT mice (Fig. 5l), which is also evident by the exhaustion of antigen-specific CD8⁺ T cells that were significantly increased (Fig. 5m). We also found that suppressive activity was higher in NcDase-deficient M2 BMDMs compared with WT M2 BMDMs (Supplementary Fig. 5h). In contrast, treatment with orlistat in TAMs and IL-4-driven BMDMs reversed this effect (Fig. 5l, m and Supplementary Fig. 5h), indicating that NcDase-mediated lipolysis supported FAO, which was critical for M2-like activation. Very importantly, TAMs treated with T863 (a potent DGAT inhibitor), etomoxir (an inhibitor of FAO) and orlistat, all failed to induce the exhaustion of CD8⁺ T cells in a macrophage-T cell co-culture system (Fig. 5m). These data clearly showed that NcDase deficiency not only regulates the lipid accumulation, but also enhances lipolysis and FAO in the M2-like cells and highlighted a role for NcDase in skewing macrophages towards pro-tumorigenic phenotype by the induction of exhausted CD8⁺ T cells.

## Macrophage NcDase deficiency promotes tumor growth

To further investigate whether the expression of NcDase modulates pro-tumorigenic features in TAMs, we generated myeloid-cell-specific NcDase-deficient mice (designated as NcDase^cKO) by crossing NcDase^fl/fl mice with LysM-Cre mice. We next engrafted EO771 breast cancer cells into WT or NcDase^cKO mice and found that genetic ablation of NcDase in myeloid cells promoted tumor growth (Fig. 6a, b). CyTOF revealed that NcDase^cKO tumors were infiltrated with a greater abundance of Ly6C⁺PD-1⁺CD73⁺CD8 T-cell subset and reduced type 1 macrophages (Fig. 6c, d and Supplementary Fig. 6a). Flow cytometry analysis demonstrated that the proportion of DC (CD11C⁺ MHCII⁺) and MDSCs (CD11b⁺Gr-1⁺) in EO771 tumors were comparable between the genotypes (Supplementary Fig. 6b). Tumor DCs expressed similar levels of CD86 and PD-L1 in NcDase^cKO mice compared with WT mice (Supplementary Fig. 6c). We next examined whether NcDase deficiency affects the function of DCs by culturing bone-marrow-derived DCs (BMDC). There were no significant differences in *IL-6*, *IL-12p70*, and *TNF-α* mRNA levels in TCM-changed BMDC from WT and NcDase⁻/⁻ mice (Supplementary Fig. 6d). Interestingly, when DCs were treated by C2 ceramide and examined apoptosis after 24 h, C2 ceramide induced apoptosis in DC (Supplementary Fig. 6e), indicating myeloid NcDase deficiency might be related to DC apoptosis. Furthermore, the increased TAMs expressing PD-L1, CD206 and arginase, and the reduced TNF-α⁺ TAMs were found in NcDase^cKO mice (Fig. 6e, f). In addition, significant increases of exhausted CD8 T-cell co-expression of the markers LAG3, TOX and PD-1 in orthotopic tumors of NcDase^cKO mice were confirmed by flow cytometry (Fig. 6g and Supplementary Fig. 6f), indicating myeloid NcDase deficiency affected CD8⁺ T cell activation in the TME. We also observed decreased frequencies of proliferating CD8⁺ effector T cells in the tumor-draining lymph nodes (TdLN) of NcDase^cKO mice (Supplementary Fig. 6g), suggesting that myeloid NcDase deficiency led to a reduced priming of antitumor CD8⁺ T cells outside the tumor. The frequencies of regulatory T cells (T_reg) (CD4⁺ FoxP3⁺) and type 1 T helper (Th1) cells in tumor and TdLN were

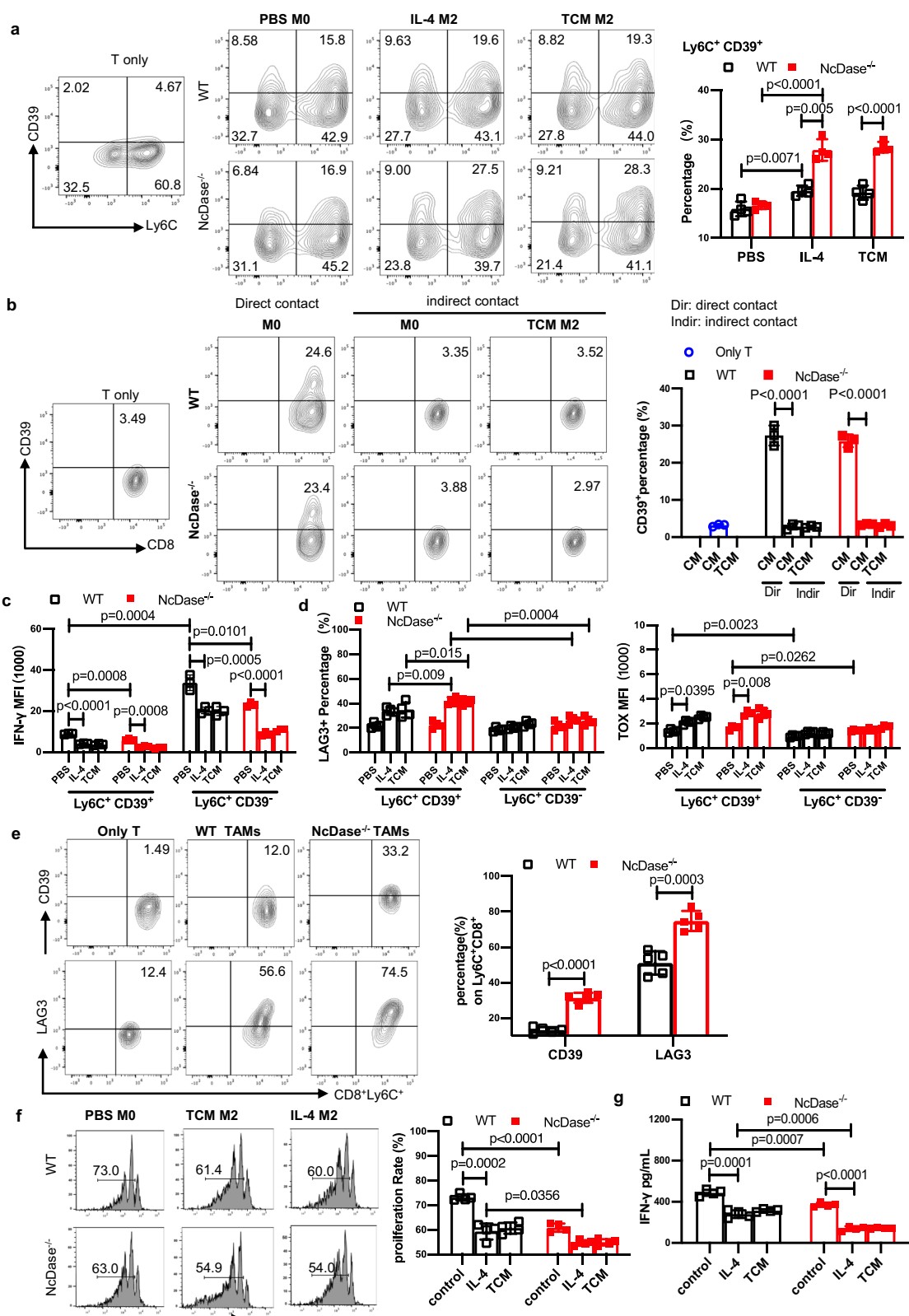

comparable between the genotypes (Supplementary Fig. 6h, i). We found reduced levels of the key inflammatory cytokines IL1β, IL12p70, and TNF-α as well as the inflammatory chemokines CCL3 and CCL4 in NcDase[cKO] mice (Fig. 6h). In addition, direct multiplex analysis of tumor lysates showed the IL-12p70 level was decreased in NcDase[cKO] mice (Fig. 6i). We then wanted to verify that this tumor promotion is mediated by CD8+ T cells in NcDase[cKO] mice. To this end, we depleted

CD8+ T cells from WT and NcDase[cKO] mice with antibody against CD8 and measured the outgrowth of EO771 tumors (Fig. 6j). Depleting CD8+ T cells promoted tumor growth in both types of mice and reduced the difference seen between WT mice and NcDase[cKO] mice (Fig. 6k). To explore if myeloid NcDase affects the effect of anti-PD-1 immunotherapy, we also treated mice bearing established EO771 tumors with the anti-PD-1 antibody RMP1-14 as monotherapy. Anti-PD-1

**Fig. 4 | NcDase regulates TAM function that contributes to the induction of exhaustion of CD8$^+$ T cells. a** Quantification of the expression of Ly6C$^+$CD39$^+$ on CD8$^+$ T cells that were co-cultured with WT or NcDase$^{-/-}$ BMDMs differentiated from PBS, IL-4 or TCM. $n$ = 4 independent biological experiments. **b** Quantification of the expression of Ly6C$^+$CD39$^+$ on CD8$^+$ T cells that were directly co-cultured with PBS-treated BMDMs or indirectly co-cultured with PBS or TCM-treated BMDMs in transwell plates. $n$ = 3 independent biological experiments. **c, d** Quantification of the expression of IFN-γ (**c**) or LAG3 and TOX (**d**) on CD8$^+$ T subsets. Naïve CD8$^+$ T cells were directly co-cultured with WT or NcDase$^{-/-}$ BMDMs differentiated from PBS, IL-4 or TCM. $n$ = 3 (**c**) or $n$ = 4 (**d**) independent biological experiments.

**a**–**d** *$p$ < 0.05, **$p$ < 0.01, ***$p$ < 0.001. **e** Quantification of the expression of CD39$^+$ or LAG3$^+$ on Ly6C$^+$CD8$^+$ T cells that were co-cultured with TAMs sorted from WT PyMT or NcDase$^{-/-}$ PyMT tumors. $n$ = 5 independent biological samples. **f** CFSE dilution and quantification representing the proliferation of CFSE$^{low/-}$ CD8$^+$ T cells after co-culturing with BMDMs from WT and NcDase$^{-/-}$ mice, differentiated in either PBS, IL-4 or TCM. $n$ = 4 independent biological experiments. **g** Quantification of IFN-γ in the supernatant from the same experiment as in (**f**). $n$ = 4 independent biological experiments. Statistical comparisons were performed using two-way ANOVA with Sidak's multiple comparisons test (**a**–**d**, **f**, **g**), two-tailed unpaired $t$-test (**e**). Error bars indicate mean ± SD. Source data are provided as a Source Data file.

antibody RMP1-14 treatment clearly reduced the size of tumors compared with the irrelevant IgG treatment in both of genotypes, however, we observed that the size of tumors in NcDase$^{cKO}$ mice were also significantly larger than those in WT mice treated with anti-PD-1 alone (Fig. 6l). Taken together, these results demonstrate that macrophage NcDase is essential for initiating CD8$^+$ T-cell exhaustion-mediated tumor growth and activation of macrophage NcDase could provide a compensatory effect for PD-1 checkpoint blockade.

## Metabolite ceramide activates TREM2$^+$ lipid associated macrophages that are responsible for the generation of exhausted CD8$^+$ T cells in the TME

Bulk transcriptome analysis revealed *TREM2*, a marker of lipid associated macrophage (LAM) and lipid droplet[39], is also enriched in TAMs from NcDase$^{-/-}$ PyMT tumor (Fig. 5a). FACS and Real-time PCR verified the higher expression of *TREM2* on TAMs from EO771-bearing NcDase$^{cKO}$ mice and NcDase$^{-/-}$ PyMT mice compared to their control mice (Fig. 7a, b). A recent study identified the infiltration of TREM2$^+$ LAMs as an event associated with cancer development[24,44]. We evaluated how expression of NcDase (*ASAH2*) in select human cancers including breast cancer from the TCGA dataset correlated with *TREM2*. In all comparisons, *ASAH2* was negatively correlated with *TREM2* (Fig. 7c and Supplementary Fig. 7a). Functionally, deletion of NcDase prevents the conversion of sphingolipid ceramide into sphingosine. TREM2 signals can sense phospholipids, modulate immune responses and contribute to a highly immunosuppressed TME in mice and humans[45]. However, how TREM2$^+$ macrophages are induced in the TME is not known, and we therefore wondered whether they correspond to the accumulated ceramide in NcDase deficient macrophages. To address this question, we initially incubated BMDMs with crude lipid extracts obtained from EO771 tumors. Tumor lipids induced higher *TREM2* expression in BMDMs derived from WT mice compared to control treatment, and further enhanced its expression in BMDMs from NcDase$^{cKO}$ mice (Supplementary Fig. 7b). To explore if ceramide accumulation in macrophages drives the TREM2 expression in the TME, we first conducted a thorough analysis of sphingolipids including the specific acyl-chain ceramides, sphingosine (SPH), Ceramide 1-phosphate (C1p) and sphingosine-1-phosphate (S1p) in the tumor tissue from MMTV mice and in the TAMs from PyMT NcDase$^{-/-}$ mice and EO771-bearing NcDase$^{cKO}$ mice. The total amount of ceramide in TAMs of EO771-bearing NcDase$^{cKO}$ mice was much higher than that in WT mice (Fig. 7d). As expected, levels of $C_{18}$, $C_{20}$, $C_{22}$, and $C_{24:1}$ ceramide species were increased in NcDase-deficient macrophages from EO771 tumor models (Fig. 7d). However, $C_{14}$ and $C_{16}$ ceramide species were not further elevated by deletion of NcDase (Fig. 7d). In addition, the total amount of sphingosine (Fig. 7d), but not S1p and C1p (Supplementary Fig. 7c), was significantly lower in TAMs of NcDase$^{cKO}$ mice. Although the levels of acylation patterns of ceramide, SPH and S1p were not significantly different in tumor tissues of WT PyMT mice versus NcDase$^{-/-}$ PyMT mice (Supplementary Fig. 7d), the levels of $C_{18}$, $C_{18:1}$, $C_{24}$, and $C_{24:1}$ ceramide species were higher in NcDase-deficient TAMs from MMTV-PyMT cancer models (Fig. 7e). This might be caused by myeloid NcDase deficiency, since no differences were found in the levels of other ceramidases including acid ceramidase (*ASAH1*) and

alkaline ceramidases (*ACER*1, 2 and 3) transcripts in TAMs between two genotypes (Supplementary Fig. 7e). To investigate whether the specificity of TREM2 in NcDase deficient BMDMs is induced by tumor derived sphingolipids, we incubated BMDMs with TCM or ceramide. Both TCM and ceramide stimulated TREM2 signaling as detected by flow cytometry (Fig. 7f). We also treated Raw264.7 macrophages with ceramide for 48 h and found TREM2 expression was significantly induced (Fig. 7f). Since TREM2 is linked with the formation of lipid droplets, we treated the BMDMs that were differentiated from WT or TREM2$^{-/-}$ mice with either DMSO or ceramide. Less formation of lipid droplets was found in TREM2 deficient macrophages compared to that in WT BMDM (Supplementary Fig. 7f). To confirm NcDase-regulated ceramide contributes to TREM2 signaling and lipid droplet accumulation, NcDase was ectopically expressed in BMDMs from NcDase$^{-/-}$ using a lentivirus bearing NcDase (named NcDase$^{RES}$ BMDMs, RES: NcDase rescued). The levels of TREM2 and lipid droplet accumulation were measured in NcDase$^{RES}$ and NcDase$^{-/-}$ BMDMs in the presence of ceramide treatment. The production of lipid droplets (Supplementary Fig. 7g) and the level of TREM2 (Supplementary Fig. 7h) were decreased in NcDase$^{RES}$ BMDMs compared to NcDase$^{-/-}$ BMDMs. These results suggest that due to the NcDase deficiency, the undegraded ceramide alters the activity of the TREM2 signaling pathway in TAMs. Previous studies indicate that the α-secretases disintegrin and metalloproteinase domain-containing protein 17 (Adam17) can cleave the full-length TREM2 protein at the stalk region to release soluble TREM2 (sTREM2) from the plasma membrane[46–48]. Therefore, we hypothesized that ceramide may increase TREM2 protein by preventing its proteolytic cleavage through Adam17. As the proinflammatory cytokine TNF-α induces Adam17 expression and promotes TREM2 protein decline[49], we treated BMDMs with TNF-α in the presence and absence of ceramide. Our data showed ceramide inhibited TREM2 shedding in BMDMs, as shown by the decreased sTREM2 amounts in the culture medium after ceramide stimulation (Fig. 7g). Consistent with this finding, ceramide treatment reduced the expression of Adam17 (Fig. 7h). The activity of Adam17 was found lower, particularly under TNF-α treatment conditions, in cell extracts from BMDMs of NcDase$^{-/-}$ mice as compared with that from WT mice (Fig. 7i). These data suggest that ceramide might be involved in the prevention of full-length TREM2 cleavage through Adam17. To define the extent to which ceramide-induced TREM2$^+$ myeloid cells affect T cell functionality, we co-cultured T cells with ceramide-derived BMDMs that were differentiated from WT or TREM2$^{-/-}$ mice. Flow cytometric analysis revealed that the basal expression of TOX, CD39 and LAG3 in T cells was significantly increased upon exposure to WT macrophages differentiated in ceramide. However, when T cells were co-incubated with TREM2$^{-/-}$ macrophages differentiated in ceramide, the T cells show less increased expression of TOX, CD39, and PD-1 compared with expression in T cells that were co-incubated with DMSO-treated macrophages (Fig. 7j and Supplementary Fig. 7i). Notably, the amount of IFN-γ was significantly reduced in ceramide-derived WT BMDM T-cell co-cultures, compared to DMSO-derived BMDM co-cultures (Supplementary Fig. 7j). However, no decrease was found in the supernatant from ceramide-treated TREM2$^{-/-}$ BMDM-T cell co-cultures compared to that from DMSO-treated TREM2$^{-/-}$ BMDM co-cultures (Supplementary

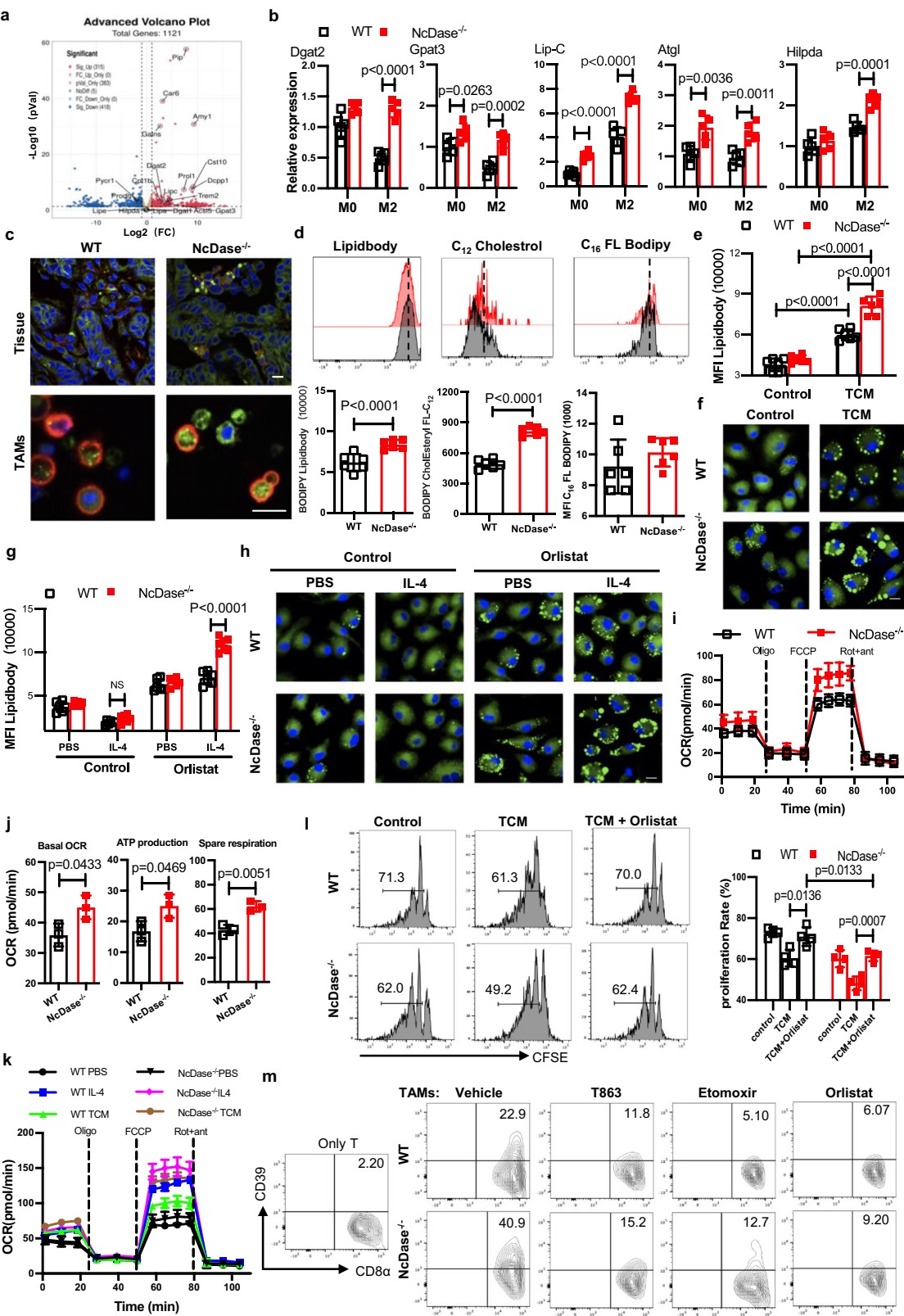

Fig. 7j). To prove NcDase deficiency promotes the exhaustion of CD8⁺ T cells via TREM2⁺ macrophages in the TME, NcDase^KO mice were crossed with TREM2^−/− mice to generate the NcDase^KO XTREM2^−/− double knockout (DKO) mice. Although a significant increase was found in the exhaustion of T cells when they were incubated with TCM-treated BMDMs derived from NcDase^−/− mice compared with those from WT mice, such an increase was not found when they were

incubated with TCM-treated BMDMs derived from DKO mice (Fig. 7k). These results indicate that ceramide accumulation or NcDase deficiency induces a TREM2⁺-related immune-suppressive phenotype in macrophages, which in turn promotes the exhaustion of T cells.

Given the essential role of ceramide in obesity noted in our previous report[50] and that TREM2⁺ macrophages are enriched in obese adipose tissue, we reasoned that a high-fat diet (HFD) would promote

**Fig. 5 | NcDase is responsible for lipid droplet and lipolysis upstream of FAO.**
**a** Volcano plots showing fold change (FC) and *p* value for the comparison of in TAMs sorted from WT PyMT or NcDase$^{-/-}$ PyMT mice. Genes up- or downregulated in TAMs (at FC > 2 and *p* < 0.05) are highlighted (red and blue). **b** Real-time PCR analysis of metabolic genes related to lipid droplets and lipolysis in M0 or M2 BMDMs derived from WT and NcDase$^{-/-}$ mice. **c** Representative images of lipid levels assessed by BODIPY493/503 staining (green) and F4/80 expression (red) in frozen sections of breast tumor (top) or in sorted TAMs (bottom) from PyMT WT mice or PyMT NcDase$^{-/-}$ mice. **d** Representative plots (top) and summary graphs of the MFI (bottom) of lipid quantification using BODIPY493/503 staining (left), the uptake of BODIPY-FL-C16 (middle) or BODIPY CholEsteryl FL-C$_{12}$ (right) in CD11b$^+$ F4/80$^+$ TAMs from WT PyMT or NcDase$^{-/-}$ PyMT mice. **e, f** Lipid quantification using BODIPY493/503 staining by flow cytometry (**e**) or confocal microscopy (**f**) in PBS or TCM-derived BMDMs. Scale bar: 10 μm. **g, h** Lipid quantification using BODIPY493/

503 staining by flow cytometry (**g**) or confocal microscopy (**h**) in PBS or IL-4-derived BMDMs with/without orlistat (100 μM) treatment. Scale bar: 10 μm. NS not significant. **i, j** OCR of TAMs from WT and NcDase$^{-/-}$ mice. **k** OCR of PBS, IL-4 or TCM-derived BMDMs from WT and NcDase$^{-/-}$ mice. **l** The proliferation of CFSE$^{low/}$ $^-$CD8$^+$ T cells after co-culturing with BMDMs. BMDMs were differentiated in either normal media, TCM or TCM+Orlistat (100 μM). **m** Quantification of the expression of Ly6C$^+$CD39$^+$ on CD8$^+$ T cells that were co-cultured with WT or NcDase$^{-/-}$ TAMs. TAMs were treated with vehicle, T863 (50 μM), etomoxir (40 μM) or orlistat (100 μM) for 48 h. Two-sided Wilcoxon's rank sum test (**a**), two-way ANOVA with Sidak's multiple comparisons test (**b, e, g, l**), two-tailed unpaired *t*-test (**d, j**) were performed. Error bars indicate mean ± SD. *n* = 3 (**a, i, j, k, m**), 4 (**l**), 5 (**b, c**), 6 (**d, e, f, g, h**) independent biological samples. Source data are provided as a Source Data file.

---

ceramide generation and accelerate NcDase deficient TAM-impaired T-cell function and enhance tumor growth in vivo. To this end, we orthotopically implanted EO771 mammary tumor cells in HFD-induced obese WT and NcDase$^{-/-}$ mice and found NcDase deficiency significantly promotes tumor growth (Fig. 8a) and caused higher tumor weights (Fig. 8b). Using anti-ceramide monoclonal antibody[51] to check the level of ceramide in TAMs from HFD fed mice, we found that the amount of ceramide in NcDase$^{-/-}$ TAMs was higher as compared to WT TAMs (Fig. 8c). In addition, T cells from NcDase$^{-/-}$ mice exhibited decreased CD8$^+$IFN-γ$^+$ cells in TILs but Th1 cells and T regulatory cells were comparable in WT and NcDase$^{-/-}$ mice (Fig. 8d). Importantly, CyTOF and flow cytometric staining confirmed more tumor-infiltrating exhausted CD8$^+$ T-cell and PD-L1$^+$CD206$^+$ macrophages in NcDase$^{-/-}$ mice than WT mice (Fig. 8e–g).

## Discussion

Here, we provide evidence that deletion of NcDase promotes tumor growth, in part by regulating lipid storage, lipolysis and FAO of TAMs and by inducing the exhaustion of cytotoxic T cells. This function was primarily attributed to ceramide/TREM2 signaling in macrophages that facilitate CD8 T cell exhaustion.

Cytotoxic CD8$^+$ T cell infiltration correlates with a better survival probability in many cancers including breast cancer. We found that NcDase (*ASAH2*) expression negatively correlates with the abundance of tumor-infiltrating macrophages and the exhaustion of CD8$^+$ T cells. We draw conclusions based on the Spearman correlations, which are robust to outliers and more appropriate for variables with non-linear relationships or those that do not follow a normal distribution[52]. The lines overlaid on the plots are based on linear regression models, which assume normal data distribution. In cases of non-normal data distribution, the regression lines may not be suitable. Algorithm included in TIMER 2.0 has unique properties and strengths[53]. TIMER can make estimations on six immune cell populations and CIBERSORT deconvolves more detailed subsets of T-cell signatures. However, we should note that the plots in Fig. 1d were generated using the TIMER2.0 web server, which may not incorporate state-of-the-art methods for handling zero-inflated data at this time. Thus, this data provides the limited data supporting the clinical relevance, however, it could provide the prediction for the association of *ASAH2* expression with breast cancer. We validated the spatial colocalization between NcDase-related macrophage infiltration and CD8$^+$ T cell exhaustion in breast cancer model in mice by imaging mass cytometry and found that CD206$^+$ macrophages were highly infiltrated in tumor margins and colocalized with CD8$^+$ T cells in NcDase$^{-/-}$ mice. Loss of NcDase in the non-tumor tissue and Rag1$^{-/-}$ mice accelerate tumor aggressiveness, thus demonstrating that myeloid NcDase is a pivotal factor in breast cancer progression.

Our study revealed that CD8$^+$ T cells within TILs could be distinguished by differential expression of CD39 and Ly6C. CD39 is an

ectonucleotidase that hydrolyzes extracellular ATP and ADP into AMP and is associated with a specific subset of CD8$^+$ T cells with exhaustion features[33,34]. Ly6C is a GPI-anchored membrane glycoprotein with poorly defined function that can act as a costimulatory molecule for T cells[54]. The study by Marshall et al.[55] recently showed that Ly6C$^{hi}$ effector CD4$^+$ T cells represent more terminally differentiated Th1 cells that were highly activated. Similar to their work, we found Ly6C$^+$ CD8$^+$ T cells expressed greater amounts of IFN-γ and TNF-α relative to the Ly6C$^{low}$ CD8$^+$ T cells, supporting Ly6C$^+$ as a signature for effector T cells in tumors. Given that CD11b$^+$Gr-1$^+$ myeloid-derived suppressor cells (MDSC) consist of two major subsets of Ly6G$^+$Ly6C$^{low}$ granulocytic and Ly6G$^-$Ly6C$^{high}$ monocytic cells, treatment with anti-Gr-1 antibodies may efficiently deplete MDSC. However, our study suggested it could cause significant reduction of cytotoxic Ly6C$^+$ CD8 T cells. In fact, Ly6C$^-$ CD8 T cells produce less IFN-γ and TNF-α. Importantly, our study further shows that the expression of CD39$^+$ within the Ly6C$^+$ CD8 TILs population is essential for transition to the exhaustion of CD8 T cells, as Ly6C$^{hi}$CD39$^+$ CD8$^+$ TILs exhibit intermediate frequency of TOX, LAG3, PD-1 and TIM-3 cells together with the reduced production of IFN-γ$^+$, and TNF-α, all features of an exhausted T cell-like phenotype. Furthermore, the Ly6C$^-$CD39$^+$ CD8 TILs displayed the highest expression of PD1, LAG3 and TOX, but the lowest expression of TNF-α. These findings suggest that Ly6C$^+$CD39$^+$ CD8$^+$ T cells can be identified as an early stage of exhausted CD8 T cells on the basis of the co-expression of exhaustion markers. Particularly, our data revealed that the Ly6C$^+$CD39$^+$ CD8$^+$ T-cells are highly specific for the TME and enriched in NcDase deficient mice, but almost absent in peripheral blood mononuclear cell or non-metastatic tumor draining lymph nodes regardless of mouse genotype. We acknowledge that immune populations, for example, macrophage profiles, CD39 and Ly6C expression pattern in CD8 T cells in MMTV-PyMT breast cancer are different from those in EO771 tumor models.

We next explored the mechanism for induction of the functionally distinct Ly6C$^+$CD39$^+$ CD8$^+$ T cell subsets in the TME. A recent study has reported that prolonged TCR stimulation in the presence of TGF-β is directly necessary for the expression of CD39 on CD8 T cells[34]. Many studies highlight that TAMs mostly come into contact with tumor-infiltrating CD8$^+$ T cells because of their high frequency and can directly induce CD8$^+$ T-cell apoptosis and physically restrict CD8$^+$ T cells from reaching their target cells[56]. Since our spatial imaging mass cytometry suggests that TAMs interact with CD8$^+$ T cells, we took a detailed look at their ability to modulate the onset of T cell exhaustion. Our co-culture experiments revealed TAMs, particularly NcDase deficient TAMs, clearly induce CD8$^+$ T cell exhaustion and further defined the cell-cell contact mechanisms underlying CD39 and LAG3 induction in CD8$^+$ T cells. Although PD-1 blockade can decrease TAM induced CD39 expression in CD8$^+$ T cell in vitro and reduce the tumor size in vivo, macrophage NcDase deficiency also induced the higher level of CD39 expression and the large size of tumor in PD-1 treated mice. Thus, we expect combination of immune checkpoint blockade with a

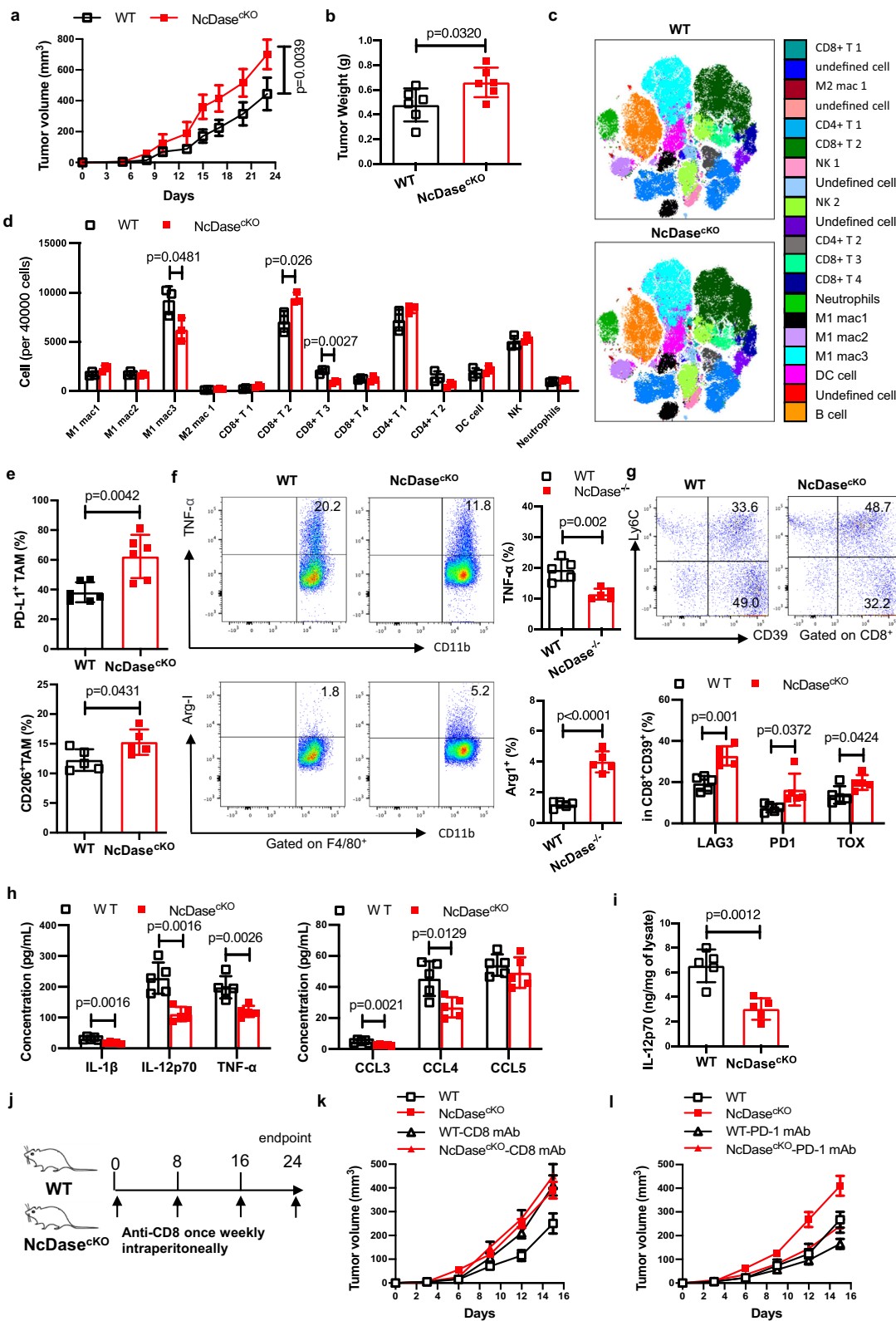

ceramidase-activating agent could yield an even greater antitumor immune response, thus framing a therapeutic design with high translational potential.

Our results suggest that NcDase orchestrates a multitude of cellular and metabolic activities that shape macrophage responses in the TME. The fatty acids needed to meet metabolic and other requirements are released through a coordinated process of lipolysis from triacylglycerols stored in lipid droplets[57–59]. We found higher

expression of genes encoding triacylglycerol lipases (*ATGL*, *Lip-c*) and synthesizing (*DGAT2*) in NcDase[-/-] TAMs, which is critical for full M2 activation of macrophages in response to IL-4. In a recent report, acylceramide generation is catalyzed by DGAT2 and involves the formation of an ACSL5-CerS-DGAT2 complex on lipid droplets[59]. Indeed, our data showed that NcDase deficiency triggers lipid storage and lipolysis and promotes the generation of fatty acids to fuel FAO in infiltrating macrophages. In vitro, inhibition of lipolysis by orlistat

**Fig. 6 | Macrophage NcDase deficiency promotes the progression of breast cancer. a, b** Outgrowth (**a**) and final weight (**b**) of EO771 tumors in syngeneic WT and myeloid NcDase knockout (NcDase^cKO) mice. **a, b** $n = 6$ mice per group. **c, d** Dot plot representation of t-distributed stochastic neighbor embedding (tSNE) analysis (**c**) and analysis of 40,000 cells (**d**) showing different clusters that enable the distinction of immune populations in EO771 tumors from WT and NcDase^cKO mice. $n = 3$ independent biological samples. **e, f** Frequencies of PD-L1^+ and CD206^+ cells (**e**) or TNF-α^+ and Arg-1^+ cells (**f**) in CD11b^+ F4/80^+ TAMs in TILs isolated from EO771 tumors implanted were detected by flow cytometry. $n = 6$ (**e**) or 5 (**f**) independent biological samples. **g** Representative dot plots and frequencies of Ly6C^+CD39^+ exhausted CD8^+ T cells in TILs isolated from EO771 tumors implanted. $n = 5$

independent biological samples. **h, i** Serum concentrations of the cytokines IL1β, IL12p70, and TNF-α and the chemokines CCL3, CCL4, and CCL5 (**h**) or tumor lysates IL-12p70 (**i**) in mice bearing EO771 tumors on day 23. $n = 5$ independent biological samples. **j** Schematic study design for depleting CD8^+ T cells from WT and NcDase^cKO mice with anti-CD8 antibody, during EO771 tumor growth. **k, l** Outgrowth of EO771 tumors in WT and NcDase^cKO mice treated with IgG, anti-CD8 (**k**) (100 μg once weekly, or anti-PD-1 (**l**) (200 ug, every three days) antibodies. $n = 5$ mice per group; *$p < 0.05$; **$p < 0.01$. Statistical comparisons were performed using, two-tailed unpaired $t$-test (**a, b, e, f, g, h, i**), two-way ANOVA with Sidak's multiple comparisons test (**d, k, l**), $P$-values are shown, and error bars indicate mean ± SD. Source data are provided as a Source Data file.

caused significant reduction in the immune suppressive capacity of TAMs on the proliferation of CD8^+ T cells. Furthermore, targeting of lipid droplets and FAO in TAMs led to substantially decreased exhaustion of CD8^+ T cells in NcDase^-/- TAMs. A better understanding of the molecular mechanisms behind induction of DGAT and lipolysis inhibitory factors will potentially allow targeting of multiple FAO-associated factors that could provide more effective approaches for abolishing the NcDase associated lipid-laden phenotype and its detrimental effects.

Our bulk RNA-seq and FACS analysis of TAMs demonstrate TREM2^+ macrophages increase in primary tumors of NcDase^-/- mice. Previous studies have highlighted different markers discriminating TAM subsets in different tumor types in both human and mouse. TREM2^+ macrophages were of monocytic origin and associated with pro-tumoral functions[60–62]. TAM diversity including the HES1^+, FOLR2^+, and TREM2^+ main subsets were recently identified in human breast tumors to reside in distinct niches[44]. It has been demonstrated that TREM2 is required for lipid droplet biogenesis[39] and regulates microglial cholesterol metabolism[63]. Our data demonstrate the contribution of ceramide to the formation of TREM2^+ LAM and are consistent with the hypothesis that TREM2 is a sensor for a broad array of sphingolipids. We acknowledge that ceramide might be involved in the prevention of the full-length TREM2 cleavage through ADAM17, which could be one of mechanistic links with TREM2 expression since recent study showed that transcription factor *YY1* can regulate *TREM2* mRNA expression[64]. Therefore, the other mechanism linking ceramide with *TREM2* mRNA expression could be explored in the future. In addition, it would be interesting to further elucidate the roles of ceramide/ TREM2 signal-derived bioactive mediators in the induction of exhausted CD8^+ T cells within the primary tumor and other tissue microenvironments. Such work will help to develop a systemic understanding, from a metabolic perspective, of TREM2^+ macrophages-CD8^+ T cells crosstalk in breast cancer. We also expect that the acquisition of the NcDase-TREM2-mediated TAMs reprogramming will promote pro-inflammatory pathways by upregulating co-stimulation genes and downregulating those encoding immune-checkpoint molecules.

## Methods
### Ethical approval statement
All the mouse experiments were performed as per University of Louisville (Protocol # 23256) Institutional Animal Care and Use Committee (IACUC) approved protocols.

### Mice
C57BL/6 mice, MMTV-PyMT mice [B6. FVB/N-Tg(MMTV-PyVT)634Mul/J (Stock No: 022974), TREM2^-/- mice and Rag1-deficient mice (Rag1^-/-) mice were obtained from Jackson Laboratory. To generate myeloid cell-specific NcDase-deleted mice, NcDase^fl/fl mice were first generated using CRISPR/Cas9 technology (Biocytogen) and then bred with LysM-cre mice (Jackson Laboratory) to generate control NcDase^fl/fl mice and LysM-cre NcDase^fl/fl cKO mice (designated as NcDase^cKO). NcDase global knockout mice (NcDase^-/-) were from Dr. Yusuf A. Hannun (Stony Brook

University) and had been backcrossed at least 8 generations to C57BL/ 6[65]. NcDase^-/- mice were crossed with MMTV-PyMT mice or Rag1^-/- mice to generate NcDase^-/- PyMT mice or *Rag1*^-/-NcDase^-/- mice. WT PyMT and NcDase^-/- PyMT animals used for breast cancer experiments were female littermates. Tumor onset was monitored by palpation and tumors were measured once a week using a caliper and volume was calculated. Animals were culled at 80–140 days of age once tumors in the control group reached the maximum allowed size. Tumor burden was calculated by adding the volume or the weight of all the tumors from the same animal. For the generation of syngeneic orthotopic tumors, 500,000 viable tumor cells derived from WT PyMT mice or NcDase^-/- PyMT were injected into the fourth right mammary fat pad of anesthetized (with 4% isoflurane) 6-week-old female control PyMT mice or NcDase^-/- PyMT mice. For the orthotopic injection of murine EO771 breast cancer, (ATCC, CRL-3461, $0.5 \times 10^6$) cells in 50 μL of PBS were injected into the fourth right mammary fat pad of anesthetized (with 4% isoflurane) 6-week-old female mice. For some studies, mice were fed either a low-fat diet (LFD, normal diet) or a high-fat diet (HFD, 60 kcal% fat; Research Diets D12492) generally from 6 weeks of age, for up to 4 months as indicated. In all mouse experiments, animals were monitored 3 times a week and tumor growth was measured using a caliper. The maximal diameter allowed for individual tumors was 15 mm. The tumor sizes for MMTV-PyMT breast cancer in Fig. 1b were the total volumes as sum of multiple individual tumors (of diameter less than 15 mm). Tumor volumes were calculated as follows: longer diameter × shorter diameter²/2. Animals were culled once tumors reached the maximum allowed size. Tumors were divided in portions for (a) preparation of tissue sections for H&E and IHC and (b) protein and RNA extraction (snap frozen). Studies were approved by the University of Louisville Institutional Animal Care and Use Committee.

### Gene expression analyses of clinical data sets and bioinformatics analyses
Associations between NcDase (*ASAH2*) mRNA expression and infiltration of different cell types from the TME were analyzed using xCell[66], CIBERSORT[67] and TIMER2.0[53], which incorporates 1100 BRCA samples from TCGA. TIMER2.0 was also used to analyze differential gene expression between tumor and normal tissues. All correlations were calculated with Spearman's rank correlation coefficient. Pearson's correlation coefficients are reported for correlation of expression of NcDase and immune regulation markers. Overall survival and Disease-free survival of patients based on NcDase (*ASAH2*) mRNA expression was calculated using data from the publicly available data sets[68] with the cBioportal[69,70]. Kaplan-Meier curves showing overall survival and disease-free survival of patients from breast cancer according to the expression of NcDase (*ASAH2*) were obtained from the Kaplan-Meier Plotter website[71].

### Preparation of tumor-infiltrating cells
Tumor infiltrating cells were isolated from both subcutaneous and autochthonous tumors at the indicated time points. Briefly, tumor tissues from sacrificed mice were prepared by mechanical disruption followed by digestion for 45 min with collagenase I (Worthington

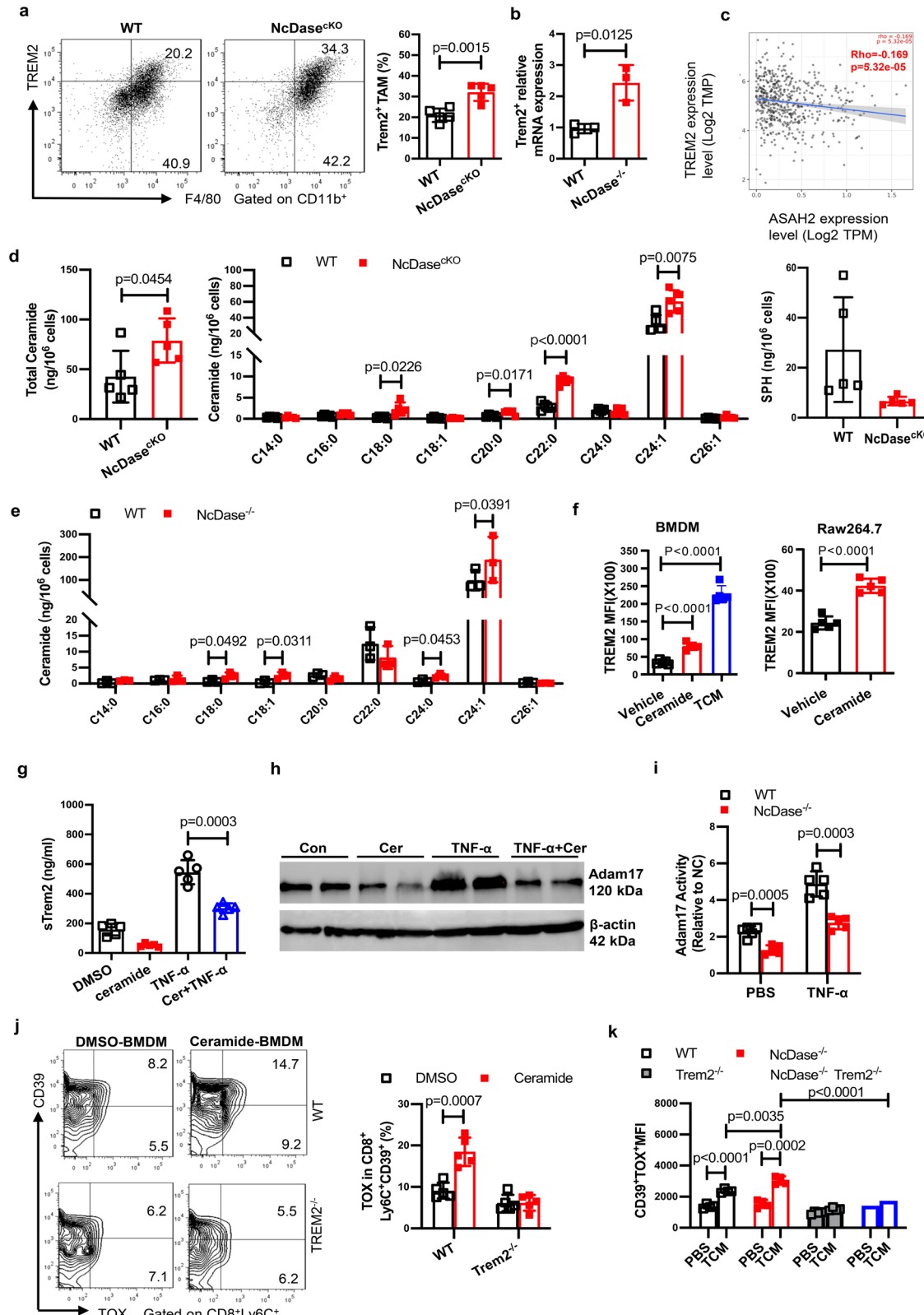

Biochemical, LS004197) and DNase I (1 mg/ml; Roche, 11284932001) at 37 °C. Digested tissues were incubated 5 min at 37 °C with EDTA (0.5 M) to prevent DC/T cell aggregates and mashed through filters.

### Time-of-flight mass cytometry (CyTOF) experiments
Single cell suspensions of tumor infiltrating lymphocytes were centrifuged at 350 × g at 4 °C for 5 min and the supernatant removed by

aspiration. Cells were resuspended in 1 mL PBS containing 0.1% BSA, 2 mM EDTA and 0.05% sodium azide, and blocked with a commercial Fc-blocking reagent. Primary metal-labeled antibody staining was performed on ice for 40 min at a dilution of 1:100, unless otherwise noted. The metal-labeled antibodies included: CD31 (148Nd), Ly6G (141 Pr), CTLA (154Sm), CD11c (142 Nd), CD8 (168Er), CD11b (172Yb), CD19 (149 Sm), F4/80 (146 Nd), CD36 (147 Sm),), PD-1 (159Tb), CD3e

**Fig. 7 | Ceramide contributes to CD8$^+$ T cell exhaustion via TREM2$^+$ TAMs.**
**a** Representative dot plots and frequencies of TREM2$^+$ in CD11b$^+$ F4/80$^+$ TAMs in TILs isolated from WT and NcDase$^{cKO}$ mice bearing EO771 tumors. **b** Real-time PCR analysis of the *TREM2* mRNA expression in CD11b$^+$ F4/80$^+$ TAMs isolated from WT PyMT and NcDase$^{-/-}$ PyMT mice. **c** Correlation between NcDase (*ASAH2*) and *TREM2* expression in human breast cancer (BRCA). $n = 568$ samples. **d** Levels of total ceramide, acyl-chain ceramides and sphingosine (SPH) in CD11b$^+$ F4/80$^+$ TAMs isolated from WT and NcDase$^{cKO}$ mice bearing EO771 tumors. **e** Levels of acyl-chain ceramides in CD11b$^+$ F4/80$^+$ TAMs isolated from WT PyMT and NcDase$^{-/-}$ PyMT mice. **f** Flow cytometry analysis of the MFI of TREM2 expression in BMDMs or RAW 264.7 macrophage treated for 48 h with vehicle, ceramide or TCM (EO771). **g** ELISA analysis of sTREM2 in cell culture medium from BMDMs treated with indicated stimuli for 24 h. Ceramide (Cer, 25 μM), TNF-α, (2 ng/mL). **h** Immunoblot analysis of ADAM17 in BMDMs stimulated with TNF-α and/or ceramide (Cer). **i** Relative

ADAM17 activity was quantified in WT or NcDase deficient BMDMs after 24 h of PBS or TNF-α (2 ng/mL) stimulation. **j** Representative dot plots and frequencies of TOX$^+$CD39$^+$ on CD8$^+$ T cells. WT or TREM2$^{-/-}$ BMDMs were differentiated from DMSO or ceramide and then were subsequently co-cultured with sub-optimally activated CD8$^+$T cells (CD3/CD28 and IL-2) isolated from spleens of naive mice for 72 h. **k** The levels of TOX$^+$CD39$^+$ on CD8$^+$ T cells that were co-cultured with BMDMs. BMDMs were derived from WT, NcDase$^{-/-}$, TREM2$^{-/-}$ or NcDase$^{-/-}$XTREM2$^{-/-}$ DKO mice and treated with DMSO or ceramide. One-way ANOVA with Tukey's multiple comparisons test (**f** left, **g**), two-tailed unpaired *t*-test (**a**, **b**, **d**, **e**, **f** right), two-way ANOVA with Sidak's multiple comparisons test (**i**, **j**, **k**), Pearson's correlation coefficient (PCC) (**c**), were performed Error bars indicate mean ± SD. $n = 3$ (**b**, **e**), 5 (**a**, **d**, **f**, **g**, **h**, **i**, **j**, **k**), independent biological samples. Source data are provided as a Source Data file.

---

(152 Sm), CD274 (153Eu), CD62L (160 Gd), CD25 (151 Eu), CD73 (154 Sm), Ly6c (150Nd), NK1.1 (170 Er), Cx3cr1 (164 Dy), CD103 (163 Dy), TIM3 (162Dy), CD39 (142Nd), LAG3 (174YbCD206 (169 Tm), CD44 (162Dy), CD68 (147Sm), CD4 (145Nd), MHCII (209Bi), B220 (176 Yb). Following primary antibody staining, cells were washed and resuspended in 0.5 mL of a 1/4000 dilution of cisplatin viability reagent (Cell-ID Cisplatin- Fluidigm, Ca) in PBS for 5 min at room temperature. Cisplatin staining was terminated and washed twice by staining buffer by centrifugation. The samples were acquired on a Helios Mass Cytometer (Fluidigm, South San Francisco, CA). The resulting FCS files were normalized using a bead-based normalization algorithm in the CyTOF acquisition software. FCS files were manually pre-gated on 193Ir DNA$^+$CD45$^+$ events to exclude cisplatin-positive dead cells, doublets, and DNA-negative debris.

### Macrophage depletion and T cell transfers
For macrophage depletion by clodronate liposomes (CELs), Rag1$^{-/-}$ mice and Rag1$^{-/-}$NcDase$^{-/-}$ mice were injected with WT PyMT tumor cells. 24 hrs prior to tumor injection mice were injected intravenously with 200 μL of clodronate liposomes (Encapsula Nanosciences). Liposomes were further administered 5 days, 10 days and 15 days after tumor injection. The efficiency of intratumoral macrophage depletion was verified by flow cytometry when the control group reached a tumor size of ~500 mm$^3$. For the purpose of CD8 depletion, C57BL/6 J mice were injected with 200 μg/mouse of a rat anti-CD8 antibody (cat # BE0223, BioXCell) or rat IgG2b anti-KLH isotype control (cat # BE0090, BioXCell) diluted in sterile PBS, 24 h before tumor injection and every 4 days after tumor injection. The efficiency of CD8 depletion was analyzed at the end of the experiment by collecting cardiac blood and performing flow cytometry for T cell subpopulations. For CD8$^+$ T cell transfers, splenic CD8$^+$ T cells were sorted and injected i.v. (1 × 10$^7$ cells/mouse) into Rag1$^{-/-}$ mice and Rag1$^{-/-}$NcDase$^{-/-}$ recipient mice prior to tumor induction.

### In vitro bone marrow-derived macrophage differentiation
Bone marrow cells were differentiated for 7 days in the presence of recombinant mouse macrophage colony-stimulating factor (M-CSF; 20 ng/ml; PeproTech) or L-929 conditional medium in complete medium (RPMI-1640 medium containing 10 mM glucose, 2 mM L-glutamine, 100 U/ml of penicillin-streptomycin and 10% FBS). Macrophages at day 7 were washed and then stimulated for 24 h with various combinations of IL-4 (20 ng/ml; PeproTech) or lipopolysaccharide (20 ng/ml; Sigma) plus IFN-γ (50 ng/ml; R&D Systems) in the presence or absence of 200 μM etomoxir (Sigma, E1905), ceramide (25 μM), breast cancer tissue-derived TCM (30%), 100 μM orlistat (Cayman), or 50 μM T863 (Sigma). Macrophages were then harvested and analyzed by flow cytometry for expression of markers of M1 or M2 activation or were cocultured with CD8$^+$ T cells. Tumors were minced into pieces <3 mm in diameter, washed with PBS 1× and resuspended in RPMI 1640 supplemented with 2 mM L-glutamine, 200 U/ml penicillin plus

50 μg/ml streptomycin, 40 μM β-mercaptoethanol and 10% FBS. The cell free supernatant was collected after 16–18 h of incubation at 37 °C and kept at −80 °C.

### Cell culture experiments
BMDMs were generated in the presence of M-CSF or L-929 conditional medium and pretreated with IL-4, TCM, etomoxir, orlistat, T863 or ceramide before BMDMs were cocultured with CD8$^+$ T cells. BMDCs were generated in the presence of GM-CSF (20 ng/ml) and pretreated with C2 ceramide or TCM. Sorted CD8$^+$ T cells were counted and plated in 96-well plates coated with anti-CD3 antibody (0.5 μg/ml, Cat# 19851, ThermoFisher Scientific) and anti-CD28 antibody (1 μg/ml, Cat# 16-0281-86, Thermo Fisher Scientific) at the desired density (250,000 T cells/50,000 TAMs or BMDMs) in the presence of IL-2 (50 ng/ml) (Cat# 212-12, PeproTech). Seventy-two hours after plating the cells were incubated with PMA and ionomycin for 1 h followed by adding the protein transport inhibitor brefeldin (Cat# 00- 4506-51, Thermo Fisher Scientific) for 5 h after which they were collected and were processed for staining for flow cytometry. CD39 expression was analyzed on the SSC$^{low}$ FSC$^{intermediate}$ CD8$^+$ T cells. For T cell proliferation, BMDMs (2 × 10$^5$ or 5 × 10$^5$, respectively) were cocultured with FACS-sorted Celltrace CFSE-labeled naïve CD8$^+$ T (1 × 10$^5$) in the presence of anti-CD3 antibody, anti-CD28 antibody and IL-2 (50 ng/ml) in 200 μL RPMI complete medium in 96-well round-bottom plates. Some BMDMs were stimulated by IL-4 or TCM with/without 100 μM orlistat (Cayman). After 24 h–72 h, cell proliferation was analyzed by Celltrace CFSE dilution (flow cytometry) and supernatants assessed for IFN-γ quantification by ELISA. RAW 264.7 macrophage (ATCC, TIB-71) were treated for 48 h with vehicle, ceramide (25 μM) or TCM (EO771).

### Oxygen-consumption rate (OCR) and extracellular acidification rate (ECAR)
BMDMs or macrophages sorted from PyMT tumors (TAMs) were plated in XF-96 well cell culture plates (6.5 × 10$^5$ cells/well) and polarized toward M2 with IL-4 for 17 h or treated with TCM for 48 h. Macrophages were rested in a non-CO$_2$ incubator for 45–60 min in XF assay medium at 37 °C before Seahorse XFe96 run. XF-96 Extracellular Flux Analyzer (Seahorse Bioscience) was used for real-time measurements of macrophage ECAR and OCR. Three or more consecutive measurements were obtained under basal conditions and after the sequential addition of 1.5 μM oligomycin, 1.5 μM FCCP (fluoro-carbonyl cyanide phenylhydrazone), and 0.5 μM rotenone plus 0.5 μM antimycin A. To assess glycolysis, three or more consecutive ECAR measurements were obtained under basal conditions and after the sequential addition of 1.5 μM oligomycin and 50 mM 2-DG (2-deoxyglucose).

### Histology, immunofluorescence and image mass cytometry (IMC)
Tissue specimens were fixed in 10% formalin, dehydrated, and then embedded in paraffin. Tissue samples were cut at 5 μm thicknesses and

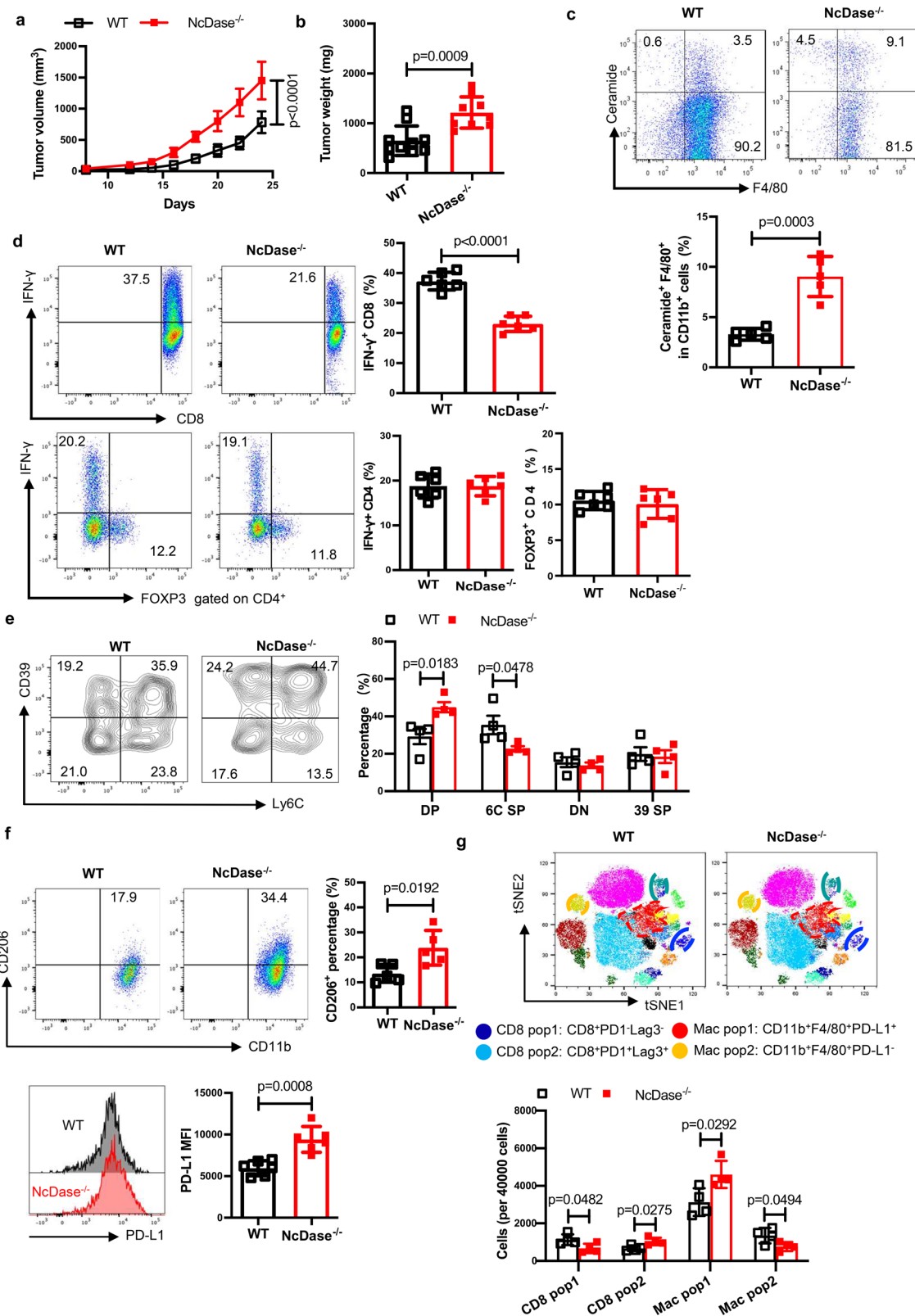

stained with hematoxylin and eosin. Sections were scanned using a PANNORAMIC DESK II DW scanner (3D Histech). For immunofluorescence analysis, OCT (Sakura Finetek)-embedded tissue cryosections (6 µm-thick) were blocked for 1 h at 22 °C with 5% BSA in DPBS and incubated overnight at 4 °C with the primary antibodies, i.e., CD4 (1:1000, catalog ab183685, Abcam), CD8 (1:125, clone 4SM16, Thermo Fisher Scientific), or Foxp3 (1:75, clone D6O8R, Cell Signaling

Technologies). Primary antibodies were detected by Alexa Fluor 488, 594 or 647 conjugated goat anti-mouse, anti-rabbit IgG and anti-rat IgG (1:600, Invitrogen). Tissues were counterstained with DAPI and images were captured on a Zeiss LSM 510 confocal microscope equipped with Zen blue imaging software (Zeiss). For IMC, after the final DPBS wash, slides were incubated with 1:500 Cell-ID Intercalator-Ir (Fluidigm) in DPBS for 10 min and rinsed with MilliQ water before air-drying

**Fig. 8 | NcDase is critical for obesity-promoted breast tumor growth.** WT and NcDase[-/-] mice on the HFD for 4 months were injected with EO771 mammary tumor cells (0.5 × 10[6]) into mammary fat pad. **a** Tumor growth was monitored by measuring tumor size every 2 days for 24 days. **b** EO771 tumor weight on 24 days after tumor injection in mice with HFD. **a, b** n = 10 mice (WT) or 9 mice (NcDase[-/-]). **c** FACS analysis of levels of ceramide in CD11b[+] F4/80[+] TAMs isolated from WT and NcDase[-/-] mice bearing EO771 tumors. n = 5 independent biological samples. **d** Representative dot plots and frequencies of CD4[+] IFN-γ[+], CD4[+] Foxp3[+] or CD8[+] IFN-γ[+] T cells in TILs isolated from HFD-fed WT and NcDase[-/-] mice bearing EO771 tumors. n = 6 independent biological samples. **e, f** Representative dot plots and

frequencies of Ly6C[+]CD39[+] on CD8[+] T cells (**e**, n = 4 independent biological samples), or CD206[+] (n = 5 independent biological samples.), PD-L1[+] (n = 6 independent biological samples.) on TAMs (**f**) in TILs isolated from HFD-fed WT and NcDase[-/-] mice bearing EO771 tumors. **g** Dot plot representation of tSNE analysis and 40,000 cells analysis showing different clusters that enable the distinction of exhausted CD8[+] T cells and macrophages in TILs isolated from HFD-fed WT and NcDase[-/-] mice bearing EO771 tumors. n = 4 independent biological samples. Statistical comparisons were performed using two-tailed unpaired t-test (**a, b, c, d, e**), two-way ANOVA with Sidak's multiple comparisons test (**e, g**), Error bars indicate mean ± SD. Source data are provided as a Source Data file.

overnight. Antibody clones conjugated with metal isotopes were purchased from Fluidigm where available (mouse CyTOF catalogue) or obtained in purified format and conjugated in house using MaxPar conjugation kits (Fluidigm) according to the manufacturer's protocol. IMC images were acquired using Hyperion Imaging Mass Cytometer. To enhance visualization of the small lymphocytes, the channels for CD3, CD206, CD8 and F4/80 were filtered using a bandpass Fast Fourier Transformation. Relative distribution of cell types within the tissues compared between the two genotypes were estimated using marginal means calculation. Each region of interest (ROI) was selected such that it would contain a whole tumor including adjacent normal tissue where possible, or, if required, they were cropped post-acquisition to contain a single tumor. For IMC images, Fiji ImageJ v2.0.0 was used to make composite images of selected channels. Six images obtained from three mice were selected for this study, ranging from 1–9 mm².

### Reagents, antibodies and flow cytometry

Splenocytes, tumor-draining lymph nodes, and TILs were incubated with anti-CD16/32 (Catalog# 14-0161-85, Thermo Fisher Scientific) to prevent nonspecific antibody binding before staining with appropriate surface antibodies for 30 min at 4 °C, washed with PBS + 2% FCS, and used for FACS analysis. Intracellular staining of the transcription factors Foxp3 was performed using the Foxp3 Fix/Perm Buffer Set (eBioscience, Thermo Fisher). For detection of intracellular cytokines, cells were first stimulated for 4 h with 50 ng/ml PMA and 1 μg/ml ionomycin in the presence of brefeldin A (5 μg/ml; All obtained from Sigma), followed by staining for surface markers. Cells were then fixed and permeabilized using the Foxp3 Fix/Perm Buffer Set and stained for intracellular cytokines. The following antibodies were used at a dilution of 1/200–1/600: PerCP-Cy5.5, PE-, FITC- or APC-labeled anti-IL-17A (TC11-18H10.1, 1:100), PE- or APC-labeled anti-IL-4 (11B11, eBioscience, Thermo fisher, 1:200), PE- or APC-labelled anti-IL-10 (JES5–16E3, 1:100), APC- or PE-Cy7-labeled anti-IFN-γ (XMG1.2, 1:200), PE-labeled anti-Foxp3 (FJK-16s, eBioscience, Thermo fisher), PE-, FITC- or APC-labeled anti-CD11b (M1/70, 1:300), PE-, FITC- or APC-labeled anti-CD4 (RM4-5, 1:300), PE-Cy7-labeled anti-CD3 (145-2C11, 1:100), PE-anti-Gr-1 (RB6-8C5, 1:600), PE- or FITC-labeled anti-mouse Ly6G (1A8, 1:200), APC-conjugated CD45.2 (104, 1:400), PE-conjugated anti-CD45.1 (A20, 1:400), FITC-, PerCP-Cy5.5 or Pacific Blue-labelled anti-CD45 (30-F11, 1:400), APC-anti-Lag-3 (C9B7W, 1:100), PE-anti-CD244.2 (2B4, 1:100), PE-anti-TOX (TXRX10, 1:100), APC-anti-CD39 (24DM1, 1:100), and PE-anti-TREM2 (FAB17291A,R&D, 1:50). All antibodies were obtained from ThermoFisher unless otherwise noted. For detection of ceramide in TAMs, TILs were incubated for 30 min at 4 °C with primary monoclonal Anti-Ceramide antibody (Sigma, # C8104-50TST) or mouse IgM isotype. Primary antibodies were detected by Alexa Fluor 488 conjugated goat anti-mouse IgM (1:400, Invitrogen). Ceramide was examined on B220[-] CD11b[+] F4/80[+] TAMs. Flow cytometry data were acquired on a 5-color FACScan (Becton Dickinson) and analyzed using FlowJo software (TreeStar). Cell sorting was performed using a FACSAria II.

### Cytokine analysis

The quantity of IL-1β, TNF-α, IL-22, and IFN-γ (Thermo Fisher, eBioscience) was determined in culture supernatants, serum and tissue

using ELISA kits according to the manufacturer's instructions. The sensitivity of the assays was <20 pg/ml. Mouse tumor supernatant samples were also evaluated for cytokine levels using the U-PLEX mouse cytokine 19-plex kit from Meso Scale Discovery (MSD, Cat. No. K15069M-1). The MSD multiplex assay plates were precoated with capture antibodies. Samples for analysis or kit standards were added at a volume of 50 μl per well after pre-diluting the original sample with assay diluent. The plates were washed after a two-hour incubation at room temperature with agitations. Sulfo-tagged detection antibodies were added and incubated for another two hours at room temperature with agitations. Following the incubation, plates were washed once again. 2X Read Substrate was added and plates were read on MSD reader. All data were analyzed by MSD Discovery Workbench® Software 4.0.

### Assay of ceramidase activity

The neutral ceramidase activity was measured using C12 NBD Ceramide (d18:1/12:0) (Caymanchem) as the substrates according to our previous reports[19,50].

### ADAM17 activity assay

The ADAM17 activities in BMDMs were assessed using a TACE (α - Secretase) Activity Assay Kit, respectively. In brief, BMDMs were seeded in 6-well plates that were pretreated with/without ceramide for 2 h, followed TNF treatment for additional 24 h. Cells were then collected, and total lysates were prepared per the manufacturer's instructions. A volume of 50 μl lysate was added into each well, followed by addition of 50 μl TACE substrate. The reaction mixture was then incubated for 30~60 min in the dark, followed by addition of 50 μl of stop solution to each well. Fluorescence intensity at Ex/Em = 490 nm/520 nm was then measured by bioTek.

### Sphingolipid analysis

Ceramide species, sphingosine and S1P in tumor tissues and TAMs were analyzed with LC-MS/MS by the McGuire Research Institute Mass Spectrometry Facility (Virginia Commonwealth University). In brief, calibration curves were constructed by plotting peak area ratios of synthetic standards corresponding to each target analyte with respect to the appropriate internal standard. The target analyte peak areas from the samples were similarly normalized to their respective internal standard and then compared with the calibration curves using a linear regression model.

### RNA extraction and PCR

RNA was isolated from the tissue or TAMs using Trizol reagent (Thermo Fisher Scientific) and reverse-transcribed with Superscript IV Kit (Invitrogen). cDNA samples were amplified in applied biosystems Realtime System using specific primers (Supplementary Table 1) and SYBR Green Master Mix (Invitrogen) to measure the genes of interest. Fold changes in mRNA expression were determined by the δCT method and normalized to the concentration of *Gapdh* mRNA or *β-actin* mRNA measured in the same samples and expressed as fold increase over baseline levels, which were set at a value of 1. Differences between groups were determined using a two-sided Student's t-test or ANOVA. Error bars on plots represent ± SD. All primers were purchased from Sigma.

## Western blot analysis

BMDMs were disrupted in RIPA lysis buffer with protease and phosphatase inhibitors (Roche) for 30 min on ice. Protein lysates were quantitated using a Bio-Rad protein kit (Bio-Rad) and 100 μg of lysates were separated on 10% SDS polyacrylamide gels and transferred to a nitrocellulose membrane (Bio-Rad). Primary antibodies for Adam17 (ThermoFisher, Catalog # PA5-27395) and β-actin (Sigma) and HRP-conjugated secondary antibodies were used and proteins were detected using the Enhanced Chemiluminescent (ECL) reagent (Thermo Scientific). The images were acquired with ChemiDoc MP System (Bio-Rad). Uncropped and unprocessed scans of the blots were provided in the Source Data file or as a Supplementary Fig. 8.

## Gene expression profiling by RNA-seq

CD11b$^+$F4/80$^+$ TAMs were sorted from PyMT tumor-bearing mice as described above. Total RNA was extracted using Trizol reagent (Thermo Fisher, 15596018) and purified using Dynabeads Oligo (dT) (Thermo Fisher) with two rounds of purification. A Poly(A) RNA sequencing library was prepared following Illumina's TruSeq-stranded-mRNA sample preparation protocol. RNA integrity was checked with Agilent Technologies 2100 Bioanalyzer. The mRNA was fragmented into short fragments using divalent cations under elevated temperature (94 °C) (NEB, cat. e6150). Then the cleaved RNA fragments were reverse-transcribed to cDNA by SuperScript™ II Reverse Transcriptase (Invitrogen, cat. 1896649). Paired-ended sequencing was performed on Illumina's NovaSeq 6000 sequencing system. The reads containing sequencing adaptors, sequencing primers, and sequences with q quality score lower than 20 were removed prior to assembly. The cleaned sequencing reads were aligned to the reference genome using the HISAT2 package, which built a database of potential splice junctions. Multiple alignments with a maximum of two mismatches were allowed for each read sequence. StringTie was used for assembling the aligned reads of individual samples. Transcriptomes from all samples were then merged to reconstruct a comprehensive transcriptome using a proprietary Perl script of LC Sciences (Houston, Texas, U.S.A.). FPKM reads and differential expressed genes were evaluated by StringTie and edgeR, respectively. The differentially expressed mRNAs and genes were selected with log2 (fold change) ≥1 or log2 (fold change) ≤-1, and with $p$ values < 0.05.

## Quantification and statistical analysis

Values are shown as Mean ± SD except where otherwise indicated. Prism (GraphPad Software, version 9.2.0) was used to determine statistical significance. one-way ANOVA with Tukey's test, two-way ANOVA with Sidak's test, two-tailed unpaired $t$-test, log-rank test, two-tailed Pearson's r correlation were used for analysis. Survival analysis was assessed using Kaplan-Meier plots with significance determined using Logrank test. The asterisks indicate significant differences ($p < 0.05$). * indicates $p < 0.05$, **$p < 0.01$, ***$p < 0.001$. $n$ represents the number of independent samples.

## Reporting summary

Further information on research design is available in the Nature Portfolio Reporting Summary linked to this article.

## Data availability

RNA-Seq data were deposited into NIH SRA database under accession number PRJNA936597. The publicly available datasets used in this paper are from the GDAC database (https://gdac.broadinstitute.org) under the accession number (BRCA:20160128; CESC: 20160128; GBM:20160128; ACC:20160128; OV:20160128; UCEC:20160128; LGG:20160128), and the Cancer Genome Atlas (TCGA) database (https://portal.gdc.cancer.gov) under the accession number phs000178. The remaining data are available within the Article, Supplementary Information or Source Data file. Source data are provided with this paper.

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

## Acknowledgements

This work was supported by grants from the NIH R21AA025724, R21AI159194, R01DK131442, R01DK115406 and R01AA030756 (Z.D.). Y.T. is supported by NIH R01HL160927. CyTOF and IMC were performed in the Functional Immunomics Core supported by NIH P20GM135004 (Jun Yan/Jason Chesney, MPI). The authors wish to thank Dr. Hong Li from the Functional Immunomics Core for the help in CyTOF and IMC. Research reported in this publication was partially supported by the National Institute of General Medical Sciences of the National Institutes of Health under Award Number P20GM113226 (Craig McClain, PI). The content is solely the responsibility of the authors and does not necessarily represent the official views of the National Institutes of Health. We thank Dr. J. Ainsworth for editorial assistance.

## Author contributions

R.S. and Z.D. designed the study, analyzed and interpreted the data, and prepared the manuscript; R.S., C.L., Z.X., X.G. and L.C. performed the experiments and interpret the data; L.H. and M.K. performed bioinformatic analysis; Y.T., M.P., K.Y., L.S., R.M. and J.Y. interpreted the findings.

## Competing interests

The authors declare no competing interests.
