## [Peer Review File · Nature Communications]

Neutral ceramidase regulates breast cancer progression by metabolic programming of TREM2-associated macrophagesREVIEWER COMMENTS

Reviewer #1 (Remarks to the Author): with expertise in ceramide, breast cancer

In the present study 'Neutral ceramidase regulates breast cancer progression by metabolic programming Trem2-associated macrophages', authors presented a role of neutral ceramidase (NcDase) in tumor microenvironment as a regulator of tumor-associated macrophages. NcDase expression was found to be inversely correlated with abundance of tumor-associated macrophages, exhausted T-cells as well as tumor burden. Further, authors demonstrated myeloid NcDase deficiency to be responsible for regulation of anti-tumor immunity by promoting CD39 expression on CD8 T-cells that programs it towards exhaustion. NcDase deletion in macrophages alters metabolic activity by stimulating lipid storage and lipolysis as well as fatty acid oxidation that is responsible for programming of macrophages and exhaustion of T-cells. Lastly, they present that Trem2 expression is responsible for this effect which was induced upon ceramide accumulation due to NcDase deficiency. Overall, this study is interesting and sheds light on role of NcDase in macrophages in averting exhaustion of T-cells by altering lipid metabolic events.

However, there are a number of comments/remarks that should be addressed before its acceptance in Nature Communications.

Comments

1. Among 20 immune cells identified in immune profiling studies in WT and NcDase^{-/-} PyMT (Figure 2C), dendritic cells (DC) were also found to be reduced upon NcDase deficiency. Being professional antigen presenting cells, dendritic cells also cross talk with CD8 T-cells like TAMs. Authors have ruled out NcDase expression in tumor cells and CD8 T-cells, but not DCs which can also stimulate T-cell activity in tumor microenvironment. Is it possible that NcDase in DCs is also responsible for effect on T-cells and tumor growth in breast cancer? Please provide an explanation for the same.
2. In Figure S3F, after using clodronate liposome to deplete macrophages, one should observe decrease in tumor volume in both Rag1^{-/-}NcDase^{-/-}CD8NCKO and Rag1^{-/-}CD8NCKO mice as there will be no upregulation of CD39 expression in CD8 T-cells without

NcDase^{-/-} macrophages as claimed by the authors. Please elucidate as why tumor growth pattern were similar in mice with or without depletion of macrophages?

3. Authors have provided tumor conditioned medium to macrophages for its polarization to M2-Like phenotype with IL-4 as control which is further inducing exhaustion in T-cells upon NcDase deficiency. Please specify whether this tumor conditioned medium is being extracted from tissues or from cell line and justify what factors present in this media is responsible for reprogramming of macrophages.

4. In bulk RNA-seq analysis in Figure 5B, Trem2 enrichment is not being shown by gene expression analysis along with other lipid related genes including Dgat2, Gpat3, Lip-C and others. Authors need to show Trem2 expression levels also in figure 5B.

5. Authors have shown that ceramide accumulation induce Trem2 expression in macrophages. Although, Trem2 is reported to be a crucial receptor for TAMs playing a role in immunosuppression and is an immunotherapeutic target in cancer, but its ligand is not yet known. Authors should elaborate how ceramides stimulate Trem2 expression and whether it is a direct or indirect stimulation.

6. In the last section of manuscript, the authors have highlighted the role of ceramide in Trem2 signaling, but they have not shown the ceramide or sphingosine levels in Trem2^{-/-} tumor cells or any other subset of immune cells like macrophages. To prove that ceramide indeed activates Trem2⁺ macrophages authors must quantitate the absolute levels of ceramides and sphingosine in TAMs from NcDase^{-/-} PyMT tumor, E0771-bearing NcDaseCKO TAMs, BMDMs from NcDase CKO and WT mice. Any rescue experiment in NcDaseCKO TAMs in this context will also substantially prove that breakdown of ceramides by overexpression of NcDase should attenuate Trem2⁺ signaling and lipid droplet accumulation.

7. Authors claim of ceramide accumulation upon NcDase deficiency or high fat diet in Figure 6 and 7 is not validated by increase in ceramide species in tumor tissues or sorted TAMs. Authors need to show ceramide species by lipidomic studies as ceramides can be hydrolysed/metabolized into other sphingolipids species including sphingomyelins,

ceramide-1-phosphate or glucosylceramides which might have different effects in tumor microenvironment.

8. In Page 9, Page 13, Page 14, authors have written that NcDase is responsible for tumor growth or T-cell exhaustion, but it should be NcDase deficiency which is responsible for the shown effect. In addition, result heading in Page 13 'Macrophage NcDase deficiency significantly impairs tumor growth' seems to be wrongly written as tumor growth is shown to be increased not impaired upon NcDase deficiency.

9. Interesting correlation has been shown between ceramide induced Trem2 expression and an immunosuppressive phenotype and T cell exhaustion but the authors donot show any mechanism as to how ceramides activate Trem2. This is still unclear which needs to be addressed.

10. Any contribution of the de novo pathway in increasing the ceramide levels in NcDase-/- PyMT TAMs should be confirmed experimentally. There are 5 Ceramidases that breakdown ceramides to sphingosine. In the absence of NcDase status and contribution of acid ceramidase (Ac) and alkaline ceramidases (Acer1, 2 and 3) should be shown.

11. Quality of figures should be improved as some of them are not clear. The font used in the figures should be uniform, different fonts have been used in subsections of same figure. The bar graphs are also not uniform throughout. In addition, there is error in figure S3F where group with clodronate liposomes is wrongly written.

12. In the manuscript, there are many grammatical errors that needs to be corrected for instance, 'we observed increases in lipids that colocalized with F4/80 staining in tumor tissue sections', 'In contrast, treatment with orlistat in TAMs and IL-4-driven BMDMs reversed this effect' and 'We also observed significant decreases in the proliferation (CFSElow/-) and cytotoxicity (IFN- γ) of T cells when T cells were coincubated with M2 BMDMs'. In addition, hyphen should be used when using words like co-culture, co-incubation, co-expression, non-tumor, non-metastatic in the manuscript. Authors need to go through the whole manuscript and correct these errors.

13. The following mistakes should be rectified –

- a. There are several instances where multiple abbreviations are provided for the same words
- b. In page 2, Trem2 full form is not given
- c. Figure 2A & 2B image margins are overlapping due to which some of the texts are missing
- d. In Figure 2B, CyTOF data showing the color coding of undefined cells, M1 mac3 & M1 mac4 is similar which makes it difficult to distinguish the observable differences.
- e. In page 5, the sentence ‘Interestingly, when Ly6C expression was paired with CD39 expression.’ abruptly ends without any result.
- f. In page 6, TIGIT spelling is incorrect.
- g. In several instances statistical significance is missing in graphs – Figure 2G, 3I
- h. In page 13, abbreviation of ECAR is missing
- i. Grammatical error in page 13 – ‘evidenced’ should be ‘evident’
- j. The title of the paper appears incomplete with grammatical error. Please correct it.

Reviewer #2 (Remarks to the Author): with expertise in cancer immunology

Sun and colleagues report on the role of neutral ceramidase in preclinical cancer models. The authors use the MMTV-PyMT model to induce tumors in wildtype and ceramidase-deficient mice and observe an earlier onset of tumors in ceramidase-deficient mice. By using correlative analysis and depletion studies, the authors demonstrate that macrophages are involved in the effect. They further present data supporting that neutral ceramidase induces lipid droplet accumulation in macrophages and ceramide induces immune suppressive macrophages via Trem2. This is an interesting work and experiments are well described. There are some questions that should be addressed before publication of this manuscript.

1. The authors examine the tumor microenvironment by CyTOF. It would be interesting to understand if spatial interactions between CD8 T cells and macrophages are affected by the presence of ceramidase, e.g. by interaction analysis?

2. As CD8 T cells are indirectly affected by the presence of ceramidase, it would be interesting to know if immune checkpoint blockade is influenced by the presence of myeloid ceramidase.

3. Are T cells from NcDase^{-/-} tumors more exhausted on a functional level (not only phenotypically)?

4. Although the authors focus on breast cancer, many other cancer types are affected by myeloid cells. The findings could be potentially more generalized.

5. What influence has ceramide on other immune cell types than Trem2-macrophages?

Reviewer #3 (Remarks to the Author): with expertise in breast cancer, tumor associated macrophages

The current study reports that NcDase deficiency promotes breast cancer growth through metabolic reprogramming in TAMs, leading to T cell exhaustion. Mechanistically, the authors demonstrate that myeloid NcDase deficiency and Ceramide accumulation upregulate Trem2, a marker of pro-tumor lipid-associated TAMs, which in turn promotes T cell exhaustion to accelerate tumor growth. These findings are potentially interesting, but the following concerns need to be addressed.

Major concerns:

1. The survival analysis showed that the NcDase expression is not significantly correlated with the overall survival of tumor patients, however, the tumor models of both complete KO and myeloid cKO demonstrated that NcDase deficiency significantly promoted tumor growth. This discrepancy raises question about the clinical relevance of the findings. It might be better to test the association between the abundance of the NcDase⁺/NcDase⁻ TAMs with overall survival using patient datasets

2. In Fig2, the authors showed that NcDase deficiency regulates the immune profile in TME using CyTOF. However, it is not clear whether this was caused by NcDase deficiency in

myeloid cells or other stromal cells. CyTOF needs to be done using cKO mice.

3. In FigS3F, tumor growth in Rag1^{-/-}NcDase^{-/-}CD8ncKO seems to be slower than Rag1^{-/-}NcDase^{-/-}CD8ncKO +CEs. Is this statistically different? This data seems to indicate that NcDase-deficient macrophages have a tumor suppressive effect, which is inconsistent with the conclusion of the article.

4. The article describes in many places that the NcDase loss causes an increase in M2 and a decrease in M1. However, the terms "M1/M2" put forward 20 years ago are essentially wrong due to oversimplification. A better characterization of the function and molecular heterogeneity of these TAMs is needed.

5. Is there any evidence that a high-fat diet increases the level of Ceramide in macrophages? It is well known that HDF promotes cancer growth, why not feed the mice with ceramide supplement only?

6. Ceramide can be metabolized in multiple metabolic pathways. Will the absence of NcDase really cause substrate accumulation? It would be important to include data that shows the lack of NcDase causes the accumulation of Ceramide in macrophages.

Minor point:

1. The superscripts and subscripts of FigS3F & Fig6I groups are confusing.

2. It seems that the word "impairs" is incorrect in the sixth subtitle "Macrophage NcDase deficiency significantly impairs tumor growth".

3. How many samples in Fig3C, n=?

4. The data in Fig3 and FigS3 only indicates antitumor efficacy of CD8⁺ T cells is dependent on "myeloid" but not "myeloid NcDase".

Reviewer #4 (Remarks to the Author): with expertise in macrophages, metabolism

Sun et. al. propose a Neutral Ceramidase – Trem2 axis in TAMs that results in T cell exhaustion and poor tumor control. Although it appears likely that these observations are linked to some extent, the data is inadequately presented (both statistically and graphically) and the findings repeatedly overstated. The manuscript is not suitable for publication in its current form and the conclusions drawn from it by the authors are not backed by the results included.

Major concerns:

1. Statistical handling of the data presented is poor and, in several cases, erroneous. Panels 1D-G, which the authors use to link NcDase expression, T cells and Macrophages in human samples are incorrectly analysed. P values and correlation coefficients presented are not representative of the data in the graphs. Notice specifically, that of the hundreds of patient samples included in most of these scatter plots less than 5 are responsible for the linear fit models that are forced onto the data distribution. Notice also that frequently high ASAH2 expression is noted in the patients for whom the predicted abundance of macrophages or T cells is lowest (the inverse should be true if this positive and significant correlations were adequately representing the data distribution). This is particularly poignant in panel 1D “Macrophage-TIMER vs. ASAH2” where only one measurement carries the whole linear fit, while the remaining samples appear closer to a negative correlation. Given this poor data handling across all these correlative analyses and the lack of critical assessment of their presentation, results in 1F are hard to accept at face value. Nor is the rationale clear. For instance, the authors use one cancer type in D and then “validate” in a different cancer type in E, using two different (but conceptually related) methodologies, yet not the same methodology as before. This does not constitute a validation (which would involve using an independent method not based on the same type of data and principle). As such most of figure 1 is over interpreted and mischaracterized.
2. The statistical handling of the data throughout the manuscript is likewise poor. There is no indication of the authors applying an analysis of variance for their experimental measurements or any type of multiple testing corrections (either in the methods included in the manuscript, figure legends or supplementary methods). Beyond that the data distribution often reflects a result that is at odds with the reported level of significance (see

panels 7A, 7C, 7I).

3. Confocal imaging presented is never accompanied by formal quantification, so the description, though informative, is not supported by actual data. Co-localization analysis, counts per field, number of images acquired, number of cells observed, number of spots within cells, spot diameter within cells are all amenable for quantification.

4. The authors propose that expression of Ly6C and CD39 in CD8 T cells is indicative of T cell exhaustion and first note the abundance and activation profile of these cells using a multi-panel approach. They later go on to show that macrophages activated with IL-4 or TCM can induce a similar population of cells that expresses exhaustion markers. Several issues with these observations are overlooked in the manuscript:

a. The characterization of macrophages in the CyTOF results is limited to 5 markers (CD11b, Ly6C, MHC-II, CX3CR1 and CD206). Of these, only one is an M2 marker, which appears to be consistently 0 across all populations (2A), yet the authors define 2 types of M2 macrophages. Was this exclusively based on CD11b expression?

b. Following on from these observations, the authors argue that NcDase deficiency results in poor tumor control due to macrophage mediated T cell exhaustion, however, the overall immune infiltration in the tumor is greatly reduced. How do the authors reconcile the increased mitochondrial fitness described in NcDase deficient macrophages (Fig. 5) with their inability to accumulate in the TME? If these cells are reduced in frequency in the TME, how do the authors propose that they are able to exert more immunosuppressive function, via direct contact, on the also less frequent T cells.

c. The CD39 and Ly6C expression profile of CD8 T cells in Fig. 2F and 6C is completely different. Particularly troubling is the inversion in the frequency of DP and 39 SP cells. How do the authors explain these differences, particularly in relation to their exhaustion claims and tumor control.

d. BMDMs treated with IL-4 or TCM are used through out the manuscript to support the authors' claims, but the activation status of these cells is never fully examined. Despite performing RNAseq, it is unclear the extent to which macrophage activation markers are induced in the culture system used or how these change upon NcDase deficiency. The authors should characterize the cells they are constantly using to modulate T cell activation.

5. A major deficiency in the manuscript is the complete lack of validation of the central hypothesis: That ceramide accumulation due to decreased NcDase activity drives the

phenotype observed. There is no direct evidence that NcDase deficient macrophages in fact accumulate ceramides. Only tangential approaches (measuring lipid uptake) are presented. As such, there is no evidence that ceramide accumulates in these cells. Nor any mechanistic link between ceramide accumulation and TREM2 expression.

6. The authors demonstrate that NcDase deficient macrophages display a different metabolic profile, with enhanced lipid uptake and enhanced mitochondrial fitness under IL-4 stimulation, compared to their WT counterparts. Likewise, the authors show that these cells induce a greater proportion of CD39 expressing CD8 T cells. However, the accumulation of lipid droplets between these macrophages only occurred when orlistat was added to the culture system. How do the authors reconcile the ability of NcDase macrophages to promote greater CD8 T cell CD39 expression in the absence of Orlistat, with their theory that lipid accumulation and storage leading to Trem2 expression is responsible for the effect on T cells.

Minor concerns:

1. Poor readability is not improved by the constant shift in color coding. WT is often but not always black, while NcDase is often but not always red. A common scheme should be more strictly defined and adhered to, or at the very least the authors should avoid inverting it. Consistent font type and size across figures would also be welcomed.
2. Findings are repeatedly overstated, with inappropriate language used to describe often minor differences between experimental groups. Avoid for example “markedly” and “interestingly” as these are subjective. Avoid also “weakly correlated” as this is meaningless.
3. In Fig. 7F, MFI is indicated as a % with values up to 5000. This must be an error or else the data is not clearly explained.
4. In the text describing Fig. 7 the authors indicates that LAG3 was measured (7E) but no data regarding LAG3 is shown in this figure.

Reviewer #1:

In the present study 'Neutral ceramidase regulates breast cancer progression by metabolic programming TREM2-associated macrophages', authors presented a role of neutral ceramidase (NcDase) in tumor microenvironment as a regulator of tumor-associated macrophages. NcDase expression was found to be inversely correlated with abundance of tumor-associated macrophages, exhausted T-cells as well as tumor burden. Further, authors demonstrated myeloid NcDase deficiency to be responsible for regulation of anti-tumor immunity by promoting CD39 expression on CD8 T-cells that programs it towards exhaustion. NcDase deletion in macrophages alters metabolic activity by stimulating lipid storage and lipolysis as well as fatty acid oxidation that is responsible for programming of macrophages and exhaustion of T-cells. Lastly, they present that TREM2 expression is responsible for this effect which was induced upon ceramide accumulation due to NcDase deficiency. Overall, this study is interesting and sheds light on role of NcDase in macrophages in averting exhaustion of T-cells by altering lipid metabolic events.

However, there are a number of comments/remarks that should be addressed before its acceptance in Nature Communications.

Comments:

1. Among 20 immune cells identified in immune profiling studies in WT and NcDase^{-/-} PyMT (Figure 2C), dendritic cells (DC) were also found to be reduced upon NcDase deficiency. Being professional antigen presenting cells, dendritic cells also cross talk with CD8 T-cells like TAMs. Authors have ruled out NcDase expression in tumor cells and CD8 T-cells, but not DCs which can also stimulate T-cell activity in tumor microenvironment. Is it possible that NcDase in DCs is also responsible for effect on T-cells and tumor growth in breast cancer? Please provide an explanation for the same.

Response: Our results from macrophage depletion and CD8 adoptive transfer in Rag^{-/-} and Rag^{-/-} NcDase^{-/-} mice (**Figure S3**) illustrated that the function of macrophage NcDase is associated with tumor growth. The data from Lysm^{cre}NcDase^{fl/fl} conditional knockout mice further demonstrated myeloid NcDase contributes to CD8 T cells exhaustion (**Figure 6**). These data suggest the importance of macrophage NcDase in breast cancer. Gene expression profiles are largely overlapped among monocytes/macrophages, granulocytes and DCs due to their close lineage relationship. Therefore, a Cre-transgenic line specific for macrophages is intrinsically

impossible. In addition, LysM is also expressed in most granulocytes and few CD11c⁺ dendritic cells (DCs) in mice¹. We acknowledge that ceramide accumulation and TREM2 expression may be involved not only in macrophages, but also in other immune cells, (e.g. Dendritic cells) contributing to different aspects of TMEs. Therefore, the DC specific nature of ceramide-mediated CD8 function could be explored in the future. In the revised manuscript, we have discussed that more in-depth mechanistic approaches will be required to fully understand the mechanism of myeloid NcDase-mediated exhaustion of CD8 T cells, including studies in mice with DC-specific deletion of NcDase (crossed with *Itgax-cre* mice) to dissect the precise roles of NcDase/TREM2 in breast cancer *in vivo*. This has been added in the Discussion section.

2. In Figure S3F, after using clodronate liposome to deplete macrophages, one should observe decrease in tumor volume in both *Rag1^{-/-}NcDase^{-/-}CD8^{NCKO}* and *Rag1^{-/-}CD8^{NCKO}* mice as there will be no upregulation of CD39 expression in CD8 T-cells without NcDase^{-/-} macrophages as claimed by the authors. Please elucidate as why tumor growth pattern were similar in mice with or without depletion of macrophages?

Response: Unfortunately, we made mistakes during the preparation of this figure. Groups were wrongly labelled. It has been updated.

3. Authors have provided tumor conditioned medium to macrophages for its polarization to M2-Like phenotype with IL-4 as control which is further inducing exhaustion in T-cells upon NcDase deficiency. Please specify whether this tumor conditioned medium is being extracted from tissues or from cell line and justify what factors present in this media is responsible for reprogramming of macrophages.

Response: In the first submission in the main text and in supplementary methods, we mentioned TCM was derived from breast cancer tissue. The preparation of TCM has been modified to contain more detail in the Methods section. We prepared tumor conditioned medium by excising non-ulcerated tumors (~1.5 cm in diameter). Tumors were minced into pieces <3 mm in diameter, washed with PBS 1× and resuspended in RPMI 1640 supplemented with 2 mM L-glutamine, 200 U/ml penicillin plus 50 µg/ml streptomycin, 40 µM β-mercaptoethanol and 10% FBS. The cell free supernatant was collected after 16-18 h of incubation at 37 °C and kept at -80 °C. Tumor

tissue-derived soluble products, including lipids, prostaglandins, IL-6, IL-10, vascular endothelial growth factor (VEGF), and soluble CD44, as well as products from cell metabolism such as succinate, contributes to the immunosuppression of myeloid immune cells within the tumor microenvironment^{2, 3}. In this manuscript, TCM was only used to partially mimic the TME. Discussion was modified to better explain this.

4. In bulk RNA-seq analysis in Figure 5B, TREM2 enrichment is not being shown by gene expression analysis along with other lipid related genes including Dgat2, Gpat3, Lip-C and others. Authors need to **show TREM2 expression levels also in figure 5B**.

Response: Following TREM2 protein expression in TAMs (Figure 7A), the TREM2 mRNA expression in TAMs were added in new **Figure 7B**.

5. Authors have shown that ceramide accumulation induces TREM2 expression in macrophages. Although, TREM2 is reported to be a crucial receptor for TAMs playing a role in immunosuppression and is an immunotherapeutic target in cancer, but its ligand is not yet known. Authors should elaborate how ceramides stimulate **TREM2 expression** and whether it is a direct or indirect stimulation.

Response: Previous studies indicate that the α -secretases disintegrin and metalloproteinase domain-containing protein 17 (ADAM17) can cleave the full-length TREM2 protein at the stalk region to release soluble TREM2 (sTREM2) from the plasma membrane⁴⁻⁶. Therefore, we hypothesized that ceramide may increase TREM2 protein by preventing its proteolytic cleavage through ADAM17. As the proinflammatory cytokine TNF- α induces ADAM17 expression and promotes TREM2 protein decline⁷, we treated BMDMs with TNF- α in the presence and absence of ceramide. Our data showed ceramide inhibited TREM2 shedding in BMDMs, as shown by the decreased amounts of sTREM2 in the culture medium after ceramide stimulation (**Figure 7G**). Consistent with this finding, ceramide treatment reduced the expression of ADAM17 (**Figure 7H**). Indeed, the activity of Adam17 was found to be lower, particularly under TNF- α treatment conditions, in cell extracts from BMDMs of NcDase^{-/-} mice as compared with that from wild-type

mice (**Figure 7I**). These data suggest that ceramide might be involved in full-length TREM2 cleavage through ADAM17.

6. In the last section of manuscript, the authors have highlighted the role of ceramide in TREM2 signaling, but they have not shown the ceramide or sphingosine levels in TREM2^{-/-} tumor cells or any other subset of immune cells like macrophages. To prove that ceramide indeed activates TREM2⁺ macrophages authors must **quantitate the absolute levels of ceramides and sphingosine** in TAMs from NcDase^{-/-} PyMT tumor, E0771-bearing NcDaseCKO TAMs, BMDMs from NcDase CKO and WT mice. Any rescue experiment in NcDaseCKO TAMs in this context will also substantially prove that breakdown of ceramides by **overexpression of NcDase** should attenuate TREM2⁺ signaling and lipid droplet accumulation.

Response: see the response for #7 below

7. Authors claim of ceramide accumulation upon NcDase deficiency or high fat diet in Figure 6 and 7 is not validated by increase in ceramide species in tumor tissues or sorted TAMs. Authors need to show ceramide species by lipidomic studies as ceramides can be hydrolysed/metabolized into other sphingolipids species including sphingomyelins, ceramide-1-phosphate or glucosylceramides which might have different effects in tumor microenvironment.

Response: For concerns #6 and #7 about ceramide accumulation in macrophages, we conducted a thorough analysis of sphingolipids including the specific acyl-chain ceramides, sphingosine, ceramide 1-phosphate (C1p) and sphingosine-1-phosphate (S1p) in the **tumor tissue** from MMTV-PyMT mice and **in the TAMs** from MMTV-PyMT NcDase^{-/-} mice and E0771-bearing NcDase^{CKO} mice. The total amount of ceramide in TAMs of E0771-bearing NcDase^{CKO} mice was much higher than that in WT (**Figure 7D**). As expected, levels of C₁₈, C₂₀, C₂₂, and C_{24:1} ceramide species were increased in NcDase-deficient macrophages from E0771 tumor models (**Figure 7D**). However, C₁₄ and C₁₆ ceramide species were not further elevated by deletion of NcDase (**Figure 7D**). In addition, the total level amount of sphingosine (**Figure 7D**), but not S1p and C1p (**Figure S7C**), was significantly lower in TAMs of NcDase^{CKO} mice. Although the levels of acylation patterns of ceramide, SPH and S1p were not significantly different in tumor tissues of MMTV-PyMT WT versus NcDase^{-/-} mice (**Figure S7D**), the levels of C₁₈, C_{18:1}, C₂₄, and C_{24:1} ceramide species were higher in NcDase-deficient TAMs from MMTV-PyMT cancer models

(**Figure 7E**). This might be caused by myeloid NcDase deficiency, since no differences were found in the levels of other ceramidases including acid ceramidase (Asah1) and alkaline ceramidases (Acer1, 2 and 3) transcripts in TAMs between two genotypes (**Figure S7E**). We also used anti-ceramide monoclonal antibody⁸ to check the level of ceramide in TAMs from HFD fed mice. The amount of ceramide in NcDase^{-/-} TAMs was higher as compared to WT TAMs (**New Figure 8C**).

For rescue experiment, NcDase was ectopically expressed in BMDMs from NcDase cKO using a lentivirus bearing NcDase (named NcDase^{RES} BMDMs, RES: NcDase rescued). The levels of TREM2 and lipid droplet accumulation were measured in NcDase^{RES} and NcDase^{cKO} BMDMs with/without ceramide treatment. The production of lipid droplets and the level of TREM2 were decreased in NcDase^{RES} BMDMs compared to NcDase^{cKO} BMDMs (new **Figures S7G-7H**).

8. In Page 9, Page 13, Page 14, authors have written that NcDase is responsible for tumor growth or T-cell exhaustion, but it should be NcDase deficiency which is responsible for the shown effect. In addition, result heading in Page 13 'Macrophage NcDase deficiency significantly impairs tumor growth' seems to be wrongly written as tumor growth is shown to be increased not impaired upon NcDase deficiency.

Response: These were misstated in the original manuscript. NcDase was changed to NcDase deficiency. The word "impairs" was changed to "promotes".

9. Interesting correlation has been shown between ceramide induced TREM2 expression and an immunosuppressive phenotype and T cell exhaustion but the authors do not **show any mechanism as to how ceramides activate TREM2**. This is still unclear which needs to be addressed.

Response: Please see response 5 above about the effect of ceramide on ADAM17 activity.

10. Any contribution of the de novo pathway in increasing the ceramide levels in NcDase^{-/-} PyMT TAMs should be confirmed experimentally. There are 5 Ceramidases that breakdown

ceramides to sphingosine. In the absence of NcDase status and contribution of acid ceramidase (Ac) and alkaline ceramidases (Acer1, 2 and 3) should be shown.

Response: We analyzed and added the data for the expression of the additional four ceramidases including Asah1, Acer1, 2 and 3 in the TAMs from WT and NcDase KO mice (New Figure S7E).

11. Quality of figures should be improved as some of them are not clear. The font used in the figures should be uniform, different fonts have been used in subsections of same figure. The bar graphs are also not uniform throughout. In addition, there is error in figure S3F where group with clodronate liposomes is wrongly written.

Response: Extra care has been taken to correct these mistakes in the resubmission.

12. In the manuscript, there are many grammatical errors that needs to be corrected for instance, 'we observed increases in lipids that colocalized with F4/80 staining in tumor tissue sections', 'In contrast, treatment with orlistat in TAMs and IL-4-driven BMDMs reversed this effect' and 'We also observed significant decreases in the proliferation (CFSE^{low/-}) and cytotoxicity (IFN- γ) of T cells when T cells were coincubated with M2 BMDMs'. In addition, hyphen should be used when using words like co-culture, co-incubation, co-expression, non-tumor, non-metastatic in the manuscript. Authors need to go through the whole manuscript and correct these errors.

Response: We tried our best to correct the grammatical errors.

13. The following mistakes should be rectified –
a. There are several instances where multiple abbreviations are provided for the same words

Response: These have been corrected.

b. In page 2, TREM2 full form is not given

Response: Full form was added.

c. Figure 2A & 2B image margins are overlapping due to which some of the texts are missing

Response: These have been corrected.

d. In Figure 2B, CyTOF data showing the color coding of undefined cells, M1 mac3 & M1 mac4 is similar which makes it difficult to distinguish the observable differences.

Response: These have been changed to distinctly different colors.

e. In page 5, the sentence 'Interestingly, when Ly6C expression was paired with CD39 expression.' abruptly ends without any result.

Response: We changed "!" To "!".

f. In page 6, TIGIT spelling is incorrect.

Response: Corrected.

g. In several instances statistical significance is missing in graphs – Figure 2G, 3I

Response: Statistical significance was added

h. In page 13, abbreviation of ECAR is missing

Response: Abbreviation of ECAR was added.

i. Grammatical error in page 13 – 'evidenced' should be 'evident'

Response: Corrected.

j. The title of the paper appears incomplete with grammatical error. Please correct it.

Response: Rewritten.

Reviewer #2:

Sun and colleagues report on the role of neutral ceramidase in preclinical cancer models. The authors use the MMTV-PyMT model to induce tumors in wildtype and ceramidase-deficient mice and observe an earlier onset of tumors in ceramidase-deficient mice. By using correlative analysis and depletion studies, the authors demonstrate that macrophages are involved in the effect. They further present data supporting that neutral ceramidase induces lipid droplet accumulation in macrophages and ceramide induces immune suppressive macrophages via TREM2. **This is an interesting work and experiments are well described.** There are some questions that should be addressed before publication of this manuscript.

1. The authors examine the tumor microenvironment by CyTOF. It would be interesting to understand if spatial interactions between CD8 T cells and macrophages are affected by the presence of ceramidase, e.g. by interaction analysis?

Response: In **Figures 3A-3B** and **Figures S3A-3B**, we investigated the spatial distribution of 18 immune cell markers and took a more detailed look at the interactions between TAMs and CD8⁺ T cells within the TME in breast tumors by using **imaging mass cytometry** (IMC)⁹. We did find that most of colocalizations of MHCII⁺ M1 macrophages with CD8⁺ T cells in WT PyMT mice were in the tumor core, whereas colocalizations of CD206⁺ M2 macrophages with CD8⁺ T cells in NcDase^{-/-} PyMT mice were largely located in the tumor margin (**Figures 3A-3B**), suggesting that the interaction between type 1 macrophages and T cells is more direct in WT mice. We have adapted the results section on the macrophages to better reflect these interactions. We have compared the neighborhood enrichment from the two types of macrophages in **Figure 3C**.

2. As CD8 T cells are indirectly affected by the presence of ceramidase, it would be interesting to know if immune checkpoint blockade is influenced by the presence of myeloid ceramidase.

Response: We do expect that the acquisition of the NcDase-TREM2-mediated TAMs reprogramming will promote pro-inflammatory pathways by upregulating co-stimulation genes

and downregulating those encoding immune-checkpoint molecules. We agree that combining immune checkpoint blockade with a ceramidase-activating agent could yield an even greater antitumor immune response, thus framing a new therapeutic design with high translational potential. This has been further delineated in the discussion.

3. Are T cells from NcDase^{-/-} tumors more exhausted on a functional level (not only phenotypically)?

Response: We added the results for the cytotoxic, proliferation and apoptosis of exhausted T cells. We compared IFN- γ -expressing and TNF- α -expressing Ly6C⁺CD39⁺CD8 cytotoxic T cell subsets (**Figure 2G, Figures S2I and S6C**), Granzyme B (**Figures S2E and S6C**) in the tumors, Th1 and Treg (**Figures S6E-S6F**) in the tumors and TdLN. We also added analysis for exhausted T cells proliferation and apoptosis (**Figures S2J-2K and S6D**).

4. Although the authors focus on breast cancer, many other cancer types are affected by myeloid cells. The findings could be potentially more generalized.

Response: We tested the colon cancer MC38 and did not find the difference of tumor growth between two genotypes (data not shown). This result is consistent with the findings that the Asah2 mRNA expression does not correlate with macrophages and CD8 T cells in patient with colon cancer.

5. What influence has ceramide on other immune cell types than TREM2-macrophages?

Response: Please see response #1 to reviewer 1. We acknowledge that ceramide accumulation and TREM2 expression may be involved not only in macrophages, but also in other immune cells, (e.g. Dendritic cells) contributing to different aspects of TMEs. Therefore, the DCs specific nature of ceramide-mediated CD8 function could be explored in the future. In the revised manuscript, we have discussed that more in-depth mechanistic approaches will be required to fully understand the mechanism of myeloid NcDase-mediated exhaustion of CD8 T cells, including studies in mice with a DC-specific deletion of NcDase (crossed with Itgax-cre mice) to dissect the precise roles of NcDase/TREM2 in breast cancer *in vivo*. This has been added in the Discussion section.

Reviewer #3:

Reviewer #3 (Remarks to the Author): with expertise in breast cancer, tumor associated macrophages

The current study reports that NcDase deficiency promotes breast cancer growth through metabolic reprogramming in TAMs, leading to T cell exhaustion. Mechanistically, the authors demonstrate that myeloid NcDase deficiency and Ceramide accumulation upregulate TREM2, a marker of pro-tumor lipid-associated TAMs, which in turn promotes T cell exhaustion to accelerate tumor growth. These findings are potentially interesting, but the following concerns need to be addressed.

Major concerns:

1. The survival analysis showed that the NcDase expression is not significantly correlated with the overall survival of tumor patients, however, the tumor models of both complete KO and myeloid cKO demonstrated that NcDase deficiency significantly promoted tumor growth. This discrepancy raises question about the clinical relevance of the findings. It might be better to test the association between the abundance of the NcDase⁺/NcDase⁻ TAMs with overall survival using patient datasets.

Response: We searched, but could not find a database regarding patients with breast cancer to show the association between the abundance of the NcDase⁺/NcDase⁻ TAMs with overall survival.

2. In Fig2, the authors showed that NcDase deficiency regulates the immune profile in TME using CyTOF. However, it is not clear whether this was caused by NcDase deficiency in myeloid cells or other stromal cells. CyTOF needs to be done using cKO mice.

Response: CyTOF has been added in new **Figures 6C-6D and S6A**.

3. In **FigS3F**, tumor growth in Rag1^{-/-}-NcDase^{-/-}-CD8ncKO seems to be slower than Rag1^{-/-}-NcDase^{-/-}-CD8ncKO +CELS. Is this statistically different? This data seems to indicate that NcDase-deficient macrophages have a tumor suppressive effect, which is inconsistent with the conclusion of the article.

Response: We apologized for the mistakes including the incorrect labels for these groups. **Figure S3F** has been updated.

4. The article describes in many places that the NcDase loss causes an increase in M2 and a decrease in M1. However, **the terms "M1/M2" put forward 20 years ago** are essentially wrong due to oversimplification. A better characterization of the function and molecular heterogeneity of these TAMs is needed.

Response: We thank the reviewer for this suggestion regarding the distinct subsets of macrophages. The cell surface markers used to identify M1/M2 macrophages were added including MHC class II, PD-L1, PD-L2, MMR/CD206, and Arginase 1 activity (ARG1) (**Figure 3H, Figure S4A, Figure 6E and Figure 7F**). Additionally, M2 macrophage markers for polarization, ARG1, FIZZ1 and YM1 were compared by real-time PCR in new **Figure S4B**. We also added the markers for macrophages expressing anti/pro-inflammatory mediators such as TNF- α and arginase in TAMs (new **Figure 6F**). In addition to these makers, we have also reviewed other papers that describe heterogeneity of the TAMs functions¹⁰⁻¹³. We have included several references and a more detailed discussion regarding the macrophage subtypes in both the results and discussion sections.

5. Is there any evidence that a high-fat diet increases the level of Ceramide in macrophages? It is well known that HDF promotes cancer growth, why not feed the mice with ceramide supplement only?

Response: Please also see response to **concerns #6 and #7** from Reviewer #1. We conducted a thorough analysis of sphingolipids including the specific acyl-chain ceramides, sphingosine, ceramide 1-phosphate (C1p) and sphingosine-1-phosphate (S1p) in the **tumor tissue** from MMTV-PyMT mice and **in the TAMs** from MMTV-PyMT NcDase^{-/-} mice and EO771-bearing NcDase^{ckO} mice. The total amount of ceramide in TAMs of EO771-bearing NcDase^{ckO} mice was much higher than that in WT (**Figure 7D**). As expected, levels of C₁₈, C₂₀, C₂₂, and C_{24:1} ceramide species were increased in NcDase-deficient macrophages from EO771 tumor models (**Figure 7D**). However, C₁₄ and C₁₆ ceramide species were not further elevated by deletion of NcDase

(Figure 7D). In addition, the total level amount of sphingosine (Figure 7D), but not S1p and C1p (Figure S7C), was significantly lower in TAMs of NcDase^{CKO} mice. Although the levels of acylation patterns of ceramide, SPH and S1p were not significantly different in tumor tissues of MMTV-PyMT WT versus NcDase^{-/-} mice (Figure S7D), the levels of C₁₈, C_{18:1}, C₂₄, and C_{24:1} ceramide species were higher in NcDase-deficient TAMs from MMTV-PyMT cancer models (Figure 7E). This might be caused by myeloid NcDase deficiency, since no differences were found in the levels of other ceramidases including acid ceramidase (Asah1) and alkaline ceramidases (Acer1, 2 and 3) transcripts in TAMs between two genotypes (Figure S7E). We also used anti-ceramide monoclonal antibody⁸ to check the level of ceramide in TAMs from HFD fed mice. The amount of ceramide in NcDase^{-/-} TAMs was higher as compared to WT TAMs (New Figure 8C).

The *de novo* synthesis of ceramide can be metabolically induced in response to metabolic loading with either serine or palmitate (fatty acyl-CoA)¹⁴ as shown in the below figure. The High fat diet (HFD) is rich in saturated fatty acid such as palmitic acid. Therefore, HFD feeding could increase ceramide as noted in our recent report¹⁵ and is a most economical way to induce ceramide *in vivo*.

6. Ceramide can be metabolized in multiple metabolic pathways. Will the absence of NcDase really cause substrate accumulation? It would be important to include data that shows the lack of NcDase causes the **accumulation of Ceramide in macrophages**.

Response: Thanks for this helpful great comment. Please see response #5 above.

Minor point:

1. The superscripts and subscripts of FigS3F & Fig6I groups are confusing.

Response: These errors have been corrected (Fig6I is now in **Figure 6K**).

2. It seems that the word "**impairs**" is incorrect in the sixth subtitle "Macrophage NcDase deficiency significantly impairs tumor growth".

Response: "Impairs" have been changed to "promotes".

3. How many samples in Fig3C, n=?

Response: "n" was added in the Figure legend.

4. The data in Fig3 and FigS3 only indicates antitumor efficacy of CD8+ T cells is dependent on "myeloid" but not "myeloid NcDase".

Response: In **Figures 3J-3I**, we compared the tumor growth in Rag1^{-/-} mice and Rag1^{-/-}NcDase^{-/-} mice upon the transfer of CD8⁺ T cells isolated from WT mice (designated as Rag1^{-/-}CD8^{WT} and Rag1^{-/-}NcDase^{-/-}CD8^{WT}, respectively). In the presence of CD8 T cells, tumor growth was significantly delayed (**Figure 3I**). Rag1^{-/-} mice receiving transferred WT CD8⁺ T cells experienced a more decreased tumor growth rate (**Figure 3I**) and exhausted T cells compared to Rag1^{-/-} NcDase^{-/-} mice receiving transferred WT CD8⁺ T cells. **These data suggest that NcDase**

expressed in non-T or non-B cells (probably through NcDase macrophages) played such a role. We also applied clodronate liposome (CELs) treatment to deplete macrophages in tumor-bearing WT Rag1^{-/-} mice and Rag1^{-/-}NcDase^{-/-} mice to examine whether macrophages are responsible for the differential tumor growth rates after CD8⁺ T cells transfer. Combined with the data for depletion in **Figure S3F**, we concluded that macrophages NcDase mediates this effect. However, we should acknowledge NcDase in some DCs subset might play a partial role.

Reviewer #4

Sun et. al. propose a Neutral Ceramidase – TREM2 axis in TAMs that results in T cell exhaustion and poor tumor control. Although it appears likely that these observations are linked to some extent, the data is inadequately presented (both statistically and graphically) and the findings repeatedly overstated. The manuscript is not suitable for publication in its current form and the conclusions drawn from it by the authors are not backed by the results included.

Major concerns:

1. Statistical handling of the data presented is poor and, in several cases, erroneous. Panels 1D-G, which the authors use to link NcDase expression, T cells and Macrophages in human samples are incorrectly analysed. P values and correlation coefficients presented are not representative of the data in the graphs. Notice specifically, that of the hundreds of patient samples included in most of these scatter plots less than 5 are responsible for the linear fit models that are forced onto the data distribution. Notice also that frequently high ASAH2 expression is noted in the patients for whom the predicted abundance of macrophages or T cells is lowest (the inverse should be true if this positive and significant correlations were adequately representing the data distribution). This is particularly poignant in panel 1D “Macrophage-TIMER vs. ASAH2” where only one measurement carries the whole linear fit, while the remaining samples appear closer to a negative correlation. Given this poor data handling across all these correlative analyses and the lack of critical assessment of their presentation, results in 1F are hard to accept at face value. Nor is the rationale clear. For instance, the authors use one cancer type in D and then “validate” in a different cancer type in E, using two different (but conceptually related) methodologies, yet not the same methodology as before. This does not constitute a validation (which would involve using an independent

method not based on the same type of data and principle). As such most of figure 1 is over interpreted and mischaracterized.

Response: The statistical analysis has been reperformed. BRCA (Breast invasive carcinoma) can be divided into four subtypes: luminal A (LumA), luminal B (LumB), HER2-enriched (Her2), and basal-like (Basal). We reanalyzed the correlation between subtypes NcDase expression, T cells and macrophages. TIMER 2.0 analysis showed that NcDase mRNA expression (encoding by ASA2) most highly correlated with CD8⁺ T cells and macrophage populations (**Figure 1D**) in lumA and lumB subtypes, but not in Basal subtype. Although Her2 subtype NcDase expression correlated with macrophages, we did not find it correlated with CD8 T cells. For **Figure 1E**, we used the same analysis by introducing **QUANTISEQ** to quantify the fractions of immune cell types from human RNA-seq data and validated that the NcDase is associated with macrophages and CD8⁺ T cells infiltration in uveal melanoma (UVM). **Figure 1F** shows the correlation between NcDase and immune regulatory markers across nine cancer types from TCGA.

2. The statistical handling of the data throughout the manuscript is likewise poor. There is no indication of the authors applying an analysis of variance for their experimental measurements or any type of multiple testing corrections (either in the methods included in the manuscript, figure legends or supplementary methods). Beyond that the data distribution often reflects a result that is at odds with the reported level of significance (see panels 7A, 7C, 7I).

Response: We apologize that this was not clear and have added more information about the analysis in the figure legends and the methods section of the revised manuscript. The errors in Figure 7 were corrected.

3. Confocal imaging presented is never accompanied by formal quantification, so the description, though informative, is not supported by actual data. Co-localization analysis, counts per field, number of images acquired, number of cells observed, number of spots within cells, spot diameter within cells are all amenable for quantification.

Response: We quantified the lipid droplets (BODIPY staining) by area, size, and density and added more description for confocal analysis in the Figure legends. We added, for example, to enhance visualization of the small lymphocytes, the channels for CD3, CD206, CD8 and F4/80 were filtered using a bandpass Fast Fourier Transformation. Relative distribution of cell types

within the tissues compared between the two genotypes were estimated using marginal means calculation. We also added the description for tissue distribution for both macrophages and CD8 T cells. Stacked bar graph in Figure S3C showing the distribution of the two macrophage types within the tissues. Type 1 macrophages are found mostly in the tumor core. Type 2 macrophages are found mostly in the tumor margin. Error bars indicate the standard error of means for the averages of proportions per regions of interest (ROI). We also modified the confocal methods. Immunofluorescence stained slides were imaged with a Zeiss Upright LSM510 microscope with a 20x objective lens using Zen blue imaging software (Zeiss). IMC images were acquired using Hyperion Imaging Mass Cytometer. Each region of interest (ROI) was selected such that it would contain a whole tumor including adjacent normal tissue where possible. If required, they were cropped post-acquisition to contain a single tumor. Twelve images obtained from four mice were selected for this study, ranging from 1-9 mm².

4. The authors propose that expression of Ly6C and CD39 in CD8 T cells is indicative of T cell exhaustion and first note the abundance and activation profile of these cells using a multi-panel approach. They later go on to show that macrophages activated with IL-4 or TCM can induce a similar population of cells that expresses exhaustion markers. Several issues with these observations are overlooked in the manuscript:

a. The characterization of macrophages in the CyTOF results is limited to 5 markers (CD11b, Ly6C, MHC-II, CX3CR1 and CD206). Of these, only one is an M2 marker, which appears to be consistently 0 across all populations (2A), yet the authors define 2 types of M2 macrophages. Was this exclusively based on CD11b expression?

Response: M2 Mac2 subtype was defined by PD-L1^{low}, CX3CR1^{low}, MHC-II^{low} and CD11C^{low}. Since the expression of CD206 on macrophage is striking, we focused on CD206^{hi}/ PD-L1^{hi}/ CX3CR1^{hi}/MHCII macrophages as immune suppressive M2 tumor-associated macrophages in new submission.

b. Following on from these observations, the authors argue that NcDase deficiency results in poor tumor control due to macrophage mediated T cell exhaustion, however, the overall immune infiltration in the tumor is greatly reduced. How do the authors reconcile the increased

mitochondrial fitness described in NcDase deficient macrophages (Fig. 5) with their inability to accumulate in the TME? If these cells are reduced in frequency in the TME, how do the authors propose that they are able to exert more immunosuppressive function, via direct contact, on the also less frequent T cells.

Response: Our results did not show the data that the overall immune infiltration in the tumor is greatly reduced in NcDase^{-/-} mice. We did find the effector Ly6C⁺ CD44⁺ CD8 T cells were decreased in the NcDase^{-/-} PyMT tumors (**Figure 2B-2C**, CD8⁺ T2 cluster and **Figure S2G**). CyTOF profiling revealed that among CD8⁺ tumor infiltrating T lymphocytes (TIL), CD8⁺Ly6C⁺CD39⁻ cells (CD8⁺ T1) were decreased in the tumors from NcDase^{-/-} PyMT versus WT PyMT mice. We also found decreased levels of M1-like macrophages in NcDase^{-/-} PyMT tumors compared with WT tumors (**Figures 2B-2C, M1 mac3**). However, we observed that NcDase^{-/-} PyMT tumors were infiltrated with a greater abundance of CD8 T cells expressing Ly6C⁻CD39⁺ (CD8⁺ T3 and T4) with exhausted markers (Tim-3⁺, PD-1⁺ and LAG3⁺) (**Figures 2D-2E**) and higher levels of M2 macrophages in the tumor margin when compared with WT mice by IMCs (**Figure 3C**) and by CyTOF (**Figure 2C, M2 mac2**). The data about increased mitochondrial fitness in NcDase deficient macrophages is consistent with the regulatory markers like higher levels of Arg1, PD-L1 and lower TNF- α secretion in NcDase deficient macrophages. Please also see Response #2 to the editor's summary and Response #d to this reviewer below.

c. The CD39 and Ly6C expression profile of CD8 T cells in Fig. 2F and 6C is completely different. Particularly troubling is the inversion in the frequency of DP and 39 SP cells. How do the authors explain these differences, particularly in relation to their exhaustion claims and tumor control.

Response: Fig. 2F was produced from MMTV-PyMT breast cancer, a spontaneous tumor model. Fig. 6C (now is Fig. 6G) was generated from EO771 a syngeneic tumor model. We did find that DP cells were much higher in EO771 tumor models. Therefore, the pattern of DP cells was caused by the different tumor models. The difference of 39 SP cells is indeed a mistake, which was wrongly labeled. It should be 6C SP (inversion). We corrected the labels of 39 SP and 6C SP.

d. BMDMs treated with IL-4 or TCM are used throughout the manuscript to support the authors' claims, but the activation status of these cells is never fully examined. Despite performing RNAseq, it is unclear the extent to which macrophage activation markers are induced in the culture system used or how these change upon NcDase deficiency. The authors should characterize the cells they are constantly using to modulate T cell activation.

Response: We appreciate this suggestion. The cell surface markers used to identify M1/M2 macrophages were added including MHC class II, PD-L1, PD-L2, MMR/CD206, and Arginase 1 activity (ARG1) (Figure 3H, Figure S4A, Figure 6E and Figure 7F). Additionally, M2 macrophage markers for polarization, ARG1, FIZZ1 and YM1 were compared by real-time PCR in new Figure S4B. We also added the macrophages expressing anti/pro-inflammatory mediators such as TNF- α and arginase in TAMs (new Figure 6F). In addition to these markers, we have also reviewed other papers that describe heterogeneity of the TAMs functions¹⁰⁻¹³. We have included several references and a more detailed discussion regarding the macrophage subtypes in both the results and discussion sections.

5. A major deficiency in the manuscript is the complete lack of validation of the central hypothesis: That ceramide accumulation due to decreased NcDase activity drives the phenotype observed. There is no direct evidence that NcDase deficient macrophages in fact accumulate ceramides. Only tangential approaches (measuring lipid uptake) are presented. As such, there is no evidence that ceramide accumulates in these cells. Nor any mechanistic link between ceramide accumulation and TREM2 expression.

Response: Thanks again for these great comments. We conducted a thorough analysis of sphingolipids including the specific acyl-chain ceramides, sphingosine, ceramide 1-phosphate (C1p) and sphingosine-1-phosphate (S1p) in the **tumor tissue** from MMTV-PyMT mice and **in the TAMs** from MMTV-PyMT NcDase^{-/-} mice and EO771-bearing NcDase^{CKO} mice. The total amount of ceramide in TAMs of EO771-bearing NcDase^{CKO} mice was much higher than that in WT (Figure 7D). As expected, levels of C₁₈, C₂₀, C₂₂, and C_{24:1} ceramide species were increased in NcDase-deficient macrophages from EO771 tumor models (Figure 7D). However, C₁₄ and C₁₆ ceramide species were not further elevated by deletion of NcDase (Figure 7D). In addition, the total level amount of sphingosine (Figure 7D), but not S1p and C1p (Figure S7C), was significantly lower in TAMs of NcDase^{CKO} mice. Although the levels of acylation patterns of

ceramide, SPH and S1p were not significantly different in tumor tissues of MMTV-PyMT WT versus NcDase^{-/-} mice (**Figure S7D**), the levels of C₁₈, C_{18:1}, C₂₄, and C_{24:1} ceramide species were higher in NcDase-deficient TAMs from MMTV-PyMT cancer models (**Figure 7E**). This might be caused by myeloid NcDase deficiency, since no differences were found in the levels of other ceramidases including acid ceramidase (Asah1) and alkaline ceramidases (Acer1, 2 and 3) transcripts in TAMs between two genotypes (**Figure S7E**). We also used anti-ceramide monoclonal antibody⁸ to check the level of ceramide in TAMs from HFD fed mice. The amount of ceramide in NcDase^{-/-} TAMs was higher as compared to WT TAMs (**New Figure 8C**).

For the question about mechanistic link between ceramide accumulation and TREM2 expression, please see **response #5** to reviewer 1 and below.

Previous studies indicate that the α -secretases disintegrin and metalloproteinase domain-containing protein 17 (ADAM17) can cleave the full-length TREM2 protein at the stalk region to release soluble TREM2 (sTREM2) from the plasma membrane⁴⁻⁶. Therefore, we hypothesized that ceramide may increase TREM2 protein by preventing its proteolytic cleavage through ADAM17. As the proinflammatory cytokine TNF- α induces ADAM17 expression and promotes TREM2 protein decline⁷, we treated BMDMs with TNF- α in the presence and absence of ceramide. Our data showed ceramide inhibited TREM2 shedding in BMDMs, as shown by the decreased amounts of sTREM2 in the culture medium after ceramide stimulation (**Figure 7G**). Consistent with this finding, ceramide treatment reduced the expression of ADAM17 (**Figure 7H**). Indeed, the activity of Adam17 was found to be lower, particularly under TNF- α treatment condition, in cell extracts from BMDMs of NcDase^{-/-} mice as compared with that from wild type mice (**Figure 7I**). These data suggest that ceramide might be involved in full-length TREM2 cleavage through ADAM17.

6. The authors demonstrate that NcDase deficient macrophages display a different metabolic profile, with enhanced lipid uptake and enhanced mitochondrial fitness under IL-4 stimulation, compared to their WT counterparts. Likewise, the authors show that these cells induce a greater proportion of CD39 expressing CD8 T cells. However, the accumulation of lipid droplets between these macrophages only occurred when orlistat was added to the culture system. How

do the authors reconcile the ability of NcDase macrophages to promote greater CD8 T cell CD39 expression in the absence of Orlistat, with their theory that lipid accumulation and storage leading to TREM2 expression is responsible for the effect on T cells.

Response: In **Figure 5F**, we already showed TCM induced more accumulation of lipid droplets in NcDase deficient macrophages, suggesting more triacylglycerols are stored in lipid droplets. Fatty acids for FAO released from TG/droplets is mediated by a process of lipolysis that is initiated by ATGL and is continued by cellular lipases including *Pnpla2*, *Lipe*, *Lipc* and *Lipg*. Since IL-4 can induce an increase of lipolysis by enhancing lipases activity¹⁶, treatment with IL-4 could promote the release of fatty acids from lipid droplets and cause the reduction of droplets (lipolysis). Therefore, in the presence of IL-4, there is no difference in lipid droplet accumulation between WT and KO macrophages as shown in **Figure 5H**. However, treatment of orlistat, an active site-directed lipase inhibitor¹⁷, prevented the lipolysis from lipid droplets, and specifically caused more lipid droplets in NcDase deficient macrophages. These data suggest NcDase/ceramide not only induces the formation of lipid droplets, but also regulates the lipolysis in macrophages (**Figure.5B** showed the expression of lipases). Our data is consistent with recent findings that ceramide is metabolized to acylceramide and stored in lipid droplets¹⁸ and tumor cell-derived glucosylceramide can induce cholesterol imbalance in macrophages¹⁹. In addition, it has also been shown that TREM2- deficient microglia are unable to adapt to excess cholesterol exposure, and thus form fewer lipid droplets²⁰. Our data suggests lipid droplet is a link between ceramide and TREM2. We have revised the discussion in the resubmission to consider these issues.

Minor concerns:

1. Poor readability is not improved by the constant shift in color coding. WT is often but not always black, while NcDase is often but not always red. A common scheme should be more strictly defined and adhered to, or at the very least the authors should avoid inverting it. Consistent font type and size across figures would also be welcomed.

Response: Font type, size and colors have been modified.

2. Findings are repeatedly overstated, with inappropriate language used to describe often minor differences between experimental groups. Avoid for example “markedly” and “interestingly” as these are subjective. Avoid also “weakly correlated” as this is meaningless.

Response: We appreciate the referee's points and have addressed the text accordingly to soften these statements as appropriate.

3. In Fig. 7F, MFI is indicated as a % with values up to 5000. This must be an error or else the data is not clearly explained.

Response: This error has been corrected.

4. In the text describing Fig. 7 the authors indicate that LAG3 was measured (7E) but no data regarding LAG3 is shown in this figure.

Response: LAG3 has been deleted.

1. Faust N, Varas F, Kelly LM, Heck S and Graf T. Insertion of enhanced green fluorescent protein into the lysozyme gene creates mice with green fluorescent granulocytes and macrophages. *Blood*. 2000;96:719-726.
2. Jang JH, Kim DH, Lim JM, Lee J, Jeong SJ, Kim KP and Surh YJ. Breast Cancer Cell-Derived Soluble CD44 Promotes Tumor Progression by Triggering Macrophage IL1 beta Production. *Cancer Res*. 2020;80:1342-1356.
3. Wu JY, Huang TW, Hsieh YT, Wang YF, Yen CC, Lee GL, Yeh CC, Peng YJ, Kuo YY, Wen HT, Lin HC, Hsiao CW, Wu KK, Kung HJ, Hsu YJ and Kuo CC. Cancer-Derived Succinate Promotes Macrophage Polarization and Cancer Metastasis via Succinate Receptor. *Mol Cell*. 2020;77:213-+.
4. Wunderlich P, Glebov K, Kemmerling N, Tien NT, Neumann H and Walter J. Sequential proteolytic processing of the triggering receptor expressed on myeloid cells-2 (TREM2) protein by ectodomain shedding and gamma-secretase-dependent intramembranous cleavage. *J Biol Chem*. 2013;288:33027-36.
5. Feuerbach D, Schindler P, Barske C, Joller S, Beng-Louka E, Worringer KA, Kommineni S, Kaykas A, Ho DJ, Ye CY, Welzenbach K, Elain G, Klein L, Brzak I, Mir AK, Farady CJ, Aichholz R, Popp S, George N and Neumann U. ADAM17 is the main sheddase for the generation of human triggering receptor expressed in myeloid cells (hTREM2) ectodomain and cleaves TREM2 after Histidine 157. *Neurosci Lett*. 2017;660:109-114.
6. Schlepckow K, Kleinberger G, Fukumori A, Feederle R, Lichtenthaler SF, Steiner H and Haass C. An Alzheimer-associated TREM2 variant occurs at the ADAM cleavage site and affects shedding and phagocytic function. *Embo Molecular Medicine*. 2017;9:1356-1365.
7. Wang XC, He QF, Zhou CL, Xu YY, Liu DH, Fujiwara N, Kubota N, Click A, Henderson P, Vancil J, Marquez CA, Gunasekaran G, Schwartz ME, Tabrizian P, Sarpel U, Fiel MI, Diao YR, Sun BC, Hoshida Y, Liang S and Zhong ZY. Prolonged hypernutrition impairs TREM2-dependent efferocytosis to license chronic liver inflammation and NASH development. *Immunity*. 2023;56:58-+.

8. Rotolo J, Stancevic B, Zhang JJ, Hua GQ, Fuller J, Yin XL, Haimovitz-Friedman A, Kim K, Qian M, Cardo-Vila M, Fuks Z, Pasqualini R, Arap W and Kolesnick R. Anti-ceramide antibody prevents the radiation gastrointestinal syndrome in mice. *Journal of Clinical Investigation*. 2012;122:1786-1790.
9. Hoch T, Schulz D, Eling N, Gomez JM, Levesque MP and Bodenmiller B. Multiplexed imaging mass cytometry of the chemokine milieu in melanoma characterizes features of the response to immunotherapy. *Sci Immunol*. 2022;7:eabk1692.
10. Katzenelenbogen Y, Sheban F, Yalin A, Yofe I, Svetlichnyy D, Jaitin DA, Bornstein C, Moshe A, Keren-Shaul H, Cohen M, Wang SY, Li B, David E, Salame TM, Weiner A and Amit I. Coupled scRNA-Seq and Intracellular Protein Activity Reveal an Immunosuppressive Role of TREM2 in Cancer. *Cell*. 2020;182:872-885 e19.
11. Cheng S, Li Z, Gao R, Xing B, Gao Y, Yang Y, Qin S, Zhang L, Ouyang H, Du P, Jiang L, Zhang B, Yang Y, Wang X, Ren X, Bei JX, Hu X, Bu Z, Ji J and Zhang Z. A pan-cancer single-cell transcriptional atlas of tumor infiltrating myeloid cells. *Cell*. 2021;184:792-809 e23.
12. Park MD, Reyes-Torres I, LeBerichel J, Hamon P, LaMarche NM, Hegde S, Belabed M, Troncoso L, Grout JA, Magen A, Humblin E, Nair A, Molgora M, Hou J, Newman JH, Farkas AM, Leader AM, Dawson T, D'Souza D, Hamel S, Sanchez-Paulete AR, Maier B, Bhardwaj N, Martin JC, Kamphorst AO, Kenigsberg E, Casanova-Acebes M, Horowitz A, Brown BD, De Andrade LF, Colonna M, Marron TU and Merad M. TREM2 macrophages drive NK cell paucity and dysfunction in lung cancer. *Nat Immunol*. 2023;24:792-801.
13. Nalio Ramos R, Missolo-Koussou Y, Gerber-Ferder Y, Bromley CP, Bugatti M, Nunez NG, Tosello Boari J, Richer W, Menger L, Denizeau J, Sedlik C, Caudana P, Kotsias F, Niborski LL, Viel S, Bohec M, Lameiras S, Baulande S, Lesage L, Nicolas A, Meseure D, Vincent-Salomon A, Reyat F, Dutertre CA, Ginhoux F, Vimeux L, Donnadiou E, Buttard B, Galon J, Zelenay S, Vermi W, Guernonprez P, Piaggio E and Helft J. Tissue-resident FOLR2(+) macrophages associate with CD8(+) T cell infiltration in human breast cancer. *Cell*. 2022;185:1189-1207 e25.
14. Kitatani K, Idkowiak-Baldys J and Hannun YA. The sphingolipid salvage pathway in ceramide metabolism and signaling. *Cell Signal*. 2008;20:1010-1018.
15. Gu XM, Sun R, Chen L, Chu SH, Doll MA, Li XH, Feng WK, Siskind L, McClain CJ and Deng ZB. Neutral Ceramidase Mediates Nonalcoholic Steatohepatitis by Regulating Monounsaturated Fatty Acids and Gut IgA(+) B Cells. *Hepatology*. 2021;73:901-919.
16. Huang SCC, Everts B, Ivanova Y, O'Sullivan D, Nascimento M, Smith AM, Beatty W, Love-Gregory L, Lam WY, O'Neil CM, Yan C, Du H, Abumrad NA, Urban JF, Artyomov MN, Pearce EL and Pearce EJ. Cell-intrinsic lysosomal lipolysis is essential for alternative activation of macrophages. *Nat Immunol*. 2014;15:846-855.
17. Heck AM, Yanovski JA and Calis KA. Orlistat, a new lipase inhibitor for the management of obesity. *Pharmacotherapy*. 2000;20:270-9.
18. Senkal CE, Salama MF, Snider AJ, Allopenna JJ, Rana NA, Koller A, Hannun YA and Obeid LM. Ceramide Is Metabolized to Acylceramide and Stored in Lipid Droplets. *Cell Metab*. 2017;25:686-697.
19. Di Conza G, Tsai CH, Gallart-Ayala H, Yu YR, Franco F, Zaffalon L, Xie X, Li X, Xiao Z, Raines LN, Falquet M, Jalil A, Locasale JW, Percipalle P, Masson D, Huang SC, Martinon F, Ivanisevic J and Ho PC. Tumor-induced reshuffling of lipid composition on the endoplasmic reticulum membrane sustains macrophage survival and pro-tumorigenic activity. *Nat Immunol*. 2021;22:1403-1415.
20. Gouna G, Klose C, Bosch-Queralt M, Liu L, Gokce O, Schifferer M, Cantuti-Castelvetri L and Simons M. TREM2-dependent lipid droplet biogenesis in phagocytes is required for remyelination. *J Exp Med*. 2021;218.

REVIEWERS' COMMENTS:

Reviewer #1 (Remarks to the Author):

In the present study 'Neutral ceramidase regulates breast cancer progression by metabolic programming TREM2-associated macrophages', authors presented a role of neutral ceramidase (NcDase) in tumor microenvironment as a regulator of tumor-associated macrophages. NcDase expression was found to be inversely correlated with abundance of tumor-associated macrophages, exhausted T-cells as well as tumor burden. Further, authors demonstrated myeloid NcDase deficiency to be responsible for regulation of anti-tumor immunity by promoting CD39 expression on CD8 T-cells that programs it towards exhaustion. NcDase deletion in macrophages alters metabolic activity by stimulating lipid storage and lipolysis as well as fatty acid oxidation that is responsible for programming of macrophages and exhaustion of T-cells. Lastly, they present that TREM2 expression is responsible for this effect which was induced upon ceramide accumulation due to NcDase deficiency. Overall, this study is interesting and sheds light on role of NcDase in macrophages in averting exhaustion of T-cells by altering lipid metabolic events.

In the revised manuscript, authors have addressed most of the comments satisfactorily. However, there are few remarks that are highlighted below which needs corrections/explanation before its final acceptance.

Comments

1. In Figure S3F, after using clodronate liposome to deplete macrophages, one should observe decrease in tumor volume in both Rag1^{-/-}NcDase^{-/-}CD8NCKO and Rag1^{-/-}CD8NCKO mice as there will be no upregulation of CD39 expression in CD8 T-cells without NcDase^{-/-} macrophages as claimed by the authors. Please elucidate as why tumor growth pattern were similar in mice with or without depletion of macrophages?

Response: Unfortunately, we made mistakes during the preparation of this figure. Groups were wrongly labelled. It has been updated.

Comment – The entire tumor volume graph in figure S3F seems different from tumor volume graph reported before and after revision of manuscript. Authors should provide an

explanation for such changes in tumor growth pattern.

2. Authors have shown that ceramide accumulation induces TREM2 expression in macrophages. Although, TREM2 is reported to be a crucial receptor for TAMs playing a role in immunosuppression and is an immunotherapeutic target in cancer, but its ligand is not yet known. Authors should elaborate how ceramides stimulate TREM2 expression and whether it is a direct or indirect stimulation.

Response: Previous studies indicate that the α -secretases disintegrin and metalloproteinase domain-containing protein 17 (ADAM17) can cleave the full-length TREM2 protein at the stalk region to release soluble TREM2 (sTREM2) from the plasma membrane⁴⁻⁶. Therefore, we hypothesized that ceramide may increase TREM2 protein by preventing its proteolytic cleavage through ADAM17. As the proinflammatory cytokine TNF- α induces ADAM17 expression and promotes TREM2 protein decline⁷, we treated BMDMs with TNF- α in the presence and absence of ceramide. Our data showed ceramide inhibited TREM2 shedding in BMDMs, as shown by the decreased amounts of sTREM2 in the culture medium after ceramide stimulation (Figure 7G). Consistent with this finding, ceramide treatment reduced the expression of ADAM17 (Figure 7H), Indeed, the activity of Adam17 was found to be lower, particularly under TNF- α treatment conditions, in cell extracts from BMDMs of NcDase-/- mice as compared with that from wild-type mice (Figure 7I). These data suggest that ceramide might be involved in full-length TREM2 cleavage through ADAM17.

Comment – Authors have written two contrasting statements (highlighted) in response to induction of TREM2 by ceramides. Although data suggests that ceramide prevents TREM2 cleavage by ADAM17 but authors have written that ceramide might be involved in full-length TREM2 cleavage through ADAM17.

Reviewer #2 (Remarks to the Author):

The authors present a revised manuscript.

Some data supporting the author's hypothesis were added.

However, most of my questions have been only addressed by changes in the discussion (e.g. influence on other cell types, addition of T cell stimulation with e.g. PD-1 blockade etc).

It would at least make sense to show data of experiments that were already performed (e.g. MC38) to complete the dataset.

Reviewer #3 (Remarks to the Author):

In the revised version, the authors did not address any of the major concerns raised in my previous comments. Seems they don't understand most of the comments. The current version is not suitable for publication in Nature Communication.

Reviewer #4 (Remarks to the Author):

The additional experimental controls and mechanistic links in the manuscript greatly improved the study. However, I remain concerned with data handling and statistical rigour.

1. The re-analysis performed on human samples is insufficiently rigorous and its interpretation does not support the author's conclusions (Fig. 1D-G).
 - a. The linear regression lines overlaid on the plots are inappropriate for data that is not normally distributed.
 - b. No handling of zero-inflated data is discussed when applying Spearman's rank test in the analysis.
 - c. TIMER 2.0, the analysis tool used, incorporates by default 6 parallel RNAseq deconvolution strategies. These may not be viewed as validations of each other, as the authors propose, but merely complementary and not independent. quanTIseq, the method mentioned here as independent, is included by default in TIMER 2.0. If the authors believe that using parallel deconvolution strategies is a form of validation, they should discuss the abundance of immune cells predicted by each algorithm included in TIMER 2.0 with respect to NcDase expression.
 - d. Even taking the interpretation of RNAseq deconvolution at face value, the evidence presented is contradictory and weak. NcDase expression does not predict survival

outcomes, and no difference was seen between normal and malignant tissue (Fig. S1B-D). e. “weak correlation” is often used for correlation coefficient values between 0.25 and 0.5 that are significant. That is not the case in the results presented (Fig. S1B) which are neither significant nor in that range.

2. The authors did not sufficiently improve the statistical rigour in their analysis. There is no multiple testing correction mentioned in their methods or figure legends. ANOVA is mentioned once in the PCR analysis section. A non-exhaustive list of analysis that is in need of an ANOVA that was not performed includes Fig. 2C,E,F,G, Fig. 4A-F, Fig. 5B, E, G, J...

3. The statistical test in Fig. 7A does not reflect the distribution of values presented. The authors should submit the raw individual values for independent verification. However, as stated before, the overall handling of data in this manuscript greatly diminishes enthusiasm for the findings.

4. The additional quantification associated with imaging results is welcomed. The lack of statistics on Fig. 3C however, dampens that enthusiasm.

Reviewer #5 (Remarks to the Author):

The authors answered most of the questions raised by the reviewer. However, there are still some concerns that would still need to be addressed.

1) The fact that there is no correlation with clinical parameters in human datasets is concerning and wakens the relevance of the findings. I understand that the paper is focused on breast cancer models, but the authors might look at different cohorts too. If there is any correlation in different tumor types, they might at least mention that.

2) The cytof analysis performed in cKO (Figure 6) mice shows a different pattern of myeloid cell populations compared to KO (Figure 2) mice. I would suggest explaining and discussing this difference.

3) The authors show the levels of ceramide in TAMs by flow cytometry. I couldn't find any method section explaining this flow staining. Also, the number of cells look different in the

two plots. Is there a general reduction of macrophages in KO mice?

[editorial note: please note that Reviewer #5 has raised few additional concerns in their confidential remarks to the Editors:

- Related to the authors' claim that ceramide inhibits TREM2 shedding resulting in higher TREM2 expression, it is not explained the increased mRNA expression shown in Fig. 7B.
- staining of TREM2 in TAMs (Fig.7A): TREM2-deficient mouse as a control and more representative plots would be required to confirm the quality of the staining.

Reviewer #1

Comment#1 – The entire tumor volume graph in figure S3F seems different from tumor volume graph reported before and after revision of manuscript. Authors should provide an explanation for such changes in tumor growth pattern.

Response: In the responses for the first submission, we stated that “we made mistakes during the preparation of this figure. Groups were wrongly labelled”. Here we would like to further explain these experiments. The experiments in Figure S3F and Figure 3I were performed simultaneously. The total number of groups for each time were eight and included multiple data points for each mouse from the eight groups. The draft figures appeared correct. However, when we separated these groups to generate new figures using the same template/figure style for manuscript submission, some groups were incorrectly labeled. The main reason for the confusion was a group’s name, for example, CD8^{NCKO} and CD8^{WT} refer to Rag1^{-/-} mice receiving CD8 T cells from NCKO mice or WT mice. We have corrected this unfortunate mistake associated with the experimental group names.

Comment #2. In the previous response: Previous studies indicate that the α -secretases disintegrin and metalloproteinase domain-containing protein 17 (ADAM17) can cleave the full-length TREM2 protein at the stalk region to release soluble TREM2 (sTREM2) from the plasma membrane. Therefore, we hypothesized that ceramide may increase TREM2 protein by preventing its proteolytic cleavage through ADAM17. As the proinflammatory cytokine TNF- α induces ADAM17 expression and promotes TREM2 protein decline We treated BMDMs with TNF- α in the presence and absence of ceramide. Our data showed ceramide inhibited TREM2 shedding in BMDMs, as shown by the decreased amounts of sTREM2 in the culture medium after ceramide stimulation (Figure 7G). Consistent with this finding, ceramide treatment reduced the expression of ADAM17 (Figure 7H), The activity of Adam17 was found to be lower, particularly under TNF- α treatment conditions, in cell extracts from BMDMs of NcDase^{-/-} mice when compared with results from wild-type mice (Figure 7I). These data suggest that ceramide might be involved in full-length TREM2 cleavage through ADAM17.

Comment #2 – Authors have written two contrasting statements (highlighted) in response to induction of TREM2 by ceramides. Although data suggests that ceramide prevents TREM2 cleavage by ADAM17 but authors have written that ceramide might be involved in full-length TREM2 cleavage through ADAM17.

Response: We appreciate this reviewer pointing out the contrasting statements. The last sentence after Figure 7I, was corrected to “These data suggest that ceramide might be involved **in the prevention** of the full-length TREM2 cleavage through ADAM17” (Page 18).

Reviewer #2

Comment: The authors present a revised manuscript. Some data supporting the author's hypothesis were added. However, most of my questions have been only addressed by changes in the discussion (e.g., influence on other cell types, addition of T cell stimulation with PD-1 blockade etc). It would at least make sense to show data of experiments that were already performed (e.g. MC38) to complete the dataset.

Response: In first revision, we addressed the concern #3 (Reviewer #2) “Are T cells from NcDase^{-/-} tumors more exhausted on a functional level (not only phenotypically)?”

Our Response in the first revision was:

We added the results for the cytotoxic, proliferation and apoptosis of exhausted T cells. We compared IFN- γ -expressing and TNF- α -expressing Ly6C⁺CD39⁺CD8 cytotoxic T cell subsets (**Figure 2G, Figures S2I and S6C**), Granzyme B (**Figures S2E and S6C**) in the tumors, Th1 and Treg (**Figures S6E-S6F**) in the tumors and TdLN. We also added analysis for exhausted T cells proliferation and apoptosis (**Figures S2J-2K and S6D**). Therefore, we addressed the concern relating to “functional T cell profile” in additional experiments.

In the second revision, we further experimentally addressed the concerns “the influence on other cell types” and “addition of T cell stimulation with e.g., PD-1 blockade etc.”

Comment #1: “the influence on other cell types”

Response: In the first submission and first revision using the MMTV and NcDase^{CKO} EO771 models, we employed CyToF and FACS to analyze many immune cell populations, including Treg, Th1 cells and DCs. Here we showed more data about DCs by FACS (Fig.1) and included the results in **Figure S6B-S6E** in the second revision.

Flow cytometry analysis demonstrated that the proportion of DC (CD11C⁺ MHCII⁺) and MDSCs (CD11b⁺Gr-1⁺) in EO771 tumors were comparable between the genotypes (Fig. 1A). Tumor DCs expressed similar levels of PD-L1 and CD86 in NcDase^{CKO} mice compared with WT mice (Fig. 1B). We next examined whether NcDase deficiency affects the function of DCs by culturing bone-marrow-derived DCs (BMDC). There were no significant differences in IL-6, IL-12p40, and TNF- α mRNA levels in TCM-changed BMDC from WT and NcDase^{-/-} mice (Fig. 1C). Interestingly, when DCs were treated by C2 ceramide and were examined apoptosis after 24 h, C2 ceramide induced apoptosis in DC (Fig. 1D), indicating NcDase deficiency might be related to DC apoptosis.

Fig. 1. (A) Frequencies of CD11b⁺Gr-1⁺ MDSCs or CD11c⁺MHCII⁺ DCs in TILs isolated from E0771 tumors implanted. (B) Quantification of the expression of MHCII, PD-L1 and CD86 on CD11b⁺CD11c⁺ DCs in the tumors. (C) Real-time PCR analysis of the mRNA levels of IL-12, IL-6 and TNF-α genes on WT or NcDase^{-/-} BMDC differentiated from TCM. (D) Exogenous ceramides induce DC apoptosis. Day 7 DC incubated with or without ceramide for 24 h. *n* =5; ** *p* < 0.01.

Comment#2: “addition of T cell stimulation with e.g., PD-1 blockade etc”

Response: We did have the data about addition of T cell stimulation in vitro. However, we didn't include it in the first submission and show this data to address the reviewer #2 concerns about PD-1 blockade in the first revision. In Figure 4B in the first submission, we showed that naïve or polarized macrophages can modulate CD39 expression in CD8⁺ T cells through cell-cell contact-dependent mechanisms. In this system, we further tried to explore which molecules/signaling are responsible for CD39 expression. We tested the effect of anti-PD-1 antibody and present the results here and added them as **Figures S4C-S4D** in the second revision. We also added in the discussion: “Although PD-1 blockade can decrease TAM induced CD39 expression in CD8⁺ T cell in vitro and reduce the tumor size in vivo, macrophage NcDase deficiency also induced the higher level of CD39 expression and the large size of tumor in PD-1 treated mice. Thus, we expect combination of immune checkpoint blockade with a ceramidase-activating agent could yield an even greater antitumor immune response, thus framing a new therapeutic design with high translational potential” (Discussion, Page 22).

“We also expect that the acquisition of the NcDase-TREM2-mediated TAMs reprogramming will promote pro-inflammatory pathways by upregulating co-stimulation genes and downregulating those encoding immune-checkpoint molecules. Combination of immune checkpoint blockade with a ceramidase-activating agent could yield an even greater antitumor immune response, thus framing a new therapeutic design with high translational potential” (Discussion, Page 23).

To explore if the molecular/signal in T cells mediates the expression of CD39, we added the anti-PD-1 mAb in the co-culture system. PD-1 mAb treatment significantly suppressed the levels of CD39 and Lag3 in CD8⁺ T cells induced by M2 macrophages derived from both of genotypes. However, the level of CD39 in CD8⁺ T cells was highly induced by NcDase deficient M2 macrophages compared to WT M2 macrophages after

PD-1 mAb treatment (Fig.2, **Figure S4C-D in revision**). This data indicates that myeloid NcDase provided additional signal contributing to CD8 exhaustion.

Fig. 2. Quantification of the expression of CD39⁺ or LAG3⁺ on Ly6C⁺CD8⁺ T cells that were co-cultured with TAMs sorted from WT PyMT or NcDase^{-/-} PyMT tumors in the presence of PD-1 (50µg/ml) or IgG. Mean ± SD, n = 5. *p < 0.05.

To explore if myeloid NcDase affects the effect of anti-PD-1 immunotherapy, we also treated mice bearing established EO771 tumors with the anti-PD-1 antibody RMP1-14 as monotherapy. Anti-PD-1 antibody RMP1-14 treatment clearly reduced the size of tumors compared with the irrelevant Ig treatment in both of genotypes, however, we observed that the size of tumor in NcDase^{CKO} mice were also significantly larger than those in WT mice treated with anti-PD-1 alone (Fig 3, **Figure 6L in revision**). Taken together, these results demonstrate that macrophage NcDase is essential for initiating CD8⁺ T-cell exhaustion-mediated tumor growth and activation of macrophage NcDase could provide a compensatory effect for PD-1 checkpoint blockade.

Fig. 3. Outgrowth of EO771 tumors in WT and NcDase^{cKO} mice treated with IgG or anti-PD-1 (200ug, every three days) antibodies. $n = 5$; * $p < 0.05$.

Comment #3: MC38 models.

Response: We apologize that the data for MC38 wasn't included in the first revision. We present it here. We tested the colon cancer MC38 and did not find a difference in tumor growth between two genotypes (Fig. 4A). This result is consistent with the findings that the Asah2 mRNA expression does not correlate with macrophages and CD8 T cells in patients with colon cancer (Fig. 4B).

Fig. 4. (A) Outgrowth of MC38 colon tumors in syngeneic WT and myeloid NcDase knockout (NcDase^{cKO}) mice. Mean \pm SD, $n = 5$. The MC38 tumor cells (1×10^6 cells/mouse in 100 μ L PBS) were subcutaneously injected into the flank of C57BL6/J mice to establish the transplanted tumor models. **(B)** Correlation of NcDase expression (ASAH2) with infiltration level of indicated cell types in Colon adenocarcinoma (COAD). Data were from the TIMER 2.0 web platform ($n=458$). Spearman's correlation coefficients and P values are shown. TPM, transcript count per million reads.

Reviewer #4.

The additional experimental controls and mechanistic links in the manuscript greatly improved the study. However, I remain concerned with data handling and statistical rigor.

1. The re-analysis performed on human samples is insufficiently rigorous and its interpretation does not support the author's conclusions (Fig. 1D-G).

Response: We have addressed this concern by engaging a statistician (Dr. Maiying Kong) who is an expert in zero-inflated models.

a. The linear regression lines overlaid on the plots are inappropriate for data that is not normally distributed.

Response: We generated the plots using the TIMER2.0 web server, as detailed in Li et al¹. To assess the correlation between two variables, the Spearman correlation coefficient employed TIMER2.0. Spearman's correlation is a non-parametric measure that evaluates the monotonic relationship

between two variables. It relies on the ranking of data rather than their actual values, making it robust to outliers and suitable for variables with non-linear relationships or those that do not adhere to a normal distribution².

We acknowledge the Reviewer's concern that "The linear regression lines overlaid on the plots are inappropriate for data that is not normally distributed." Indeed, these lines were automatically generated by the TIMER2.0 web server and were provided with special notice (red circle) on the top of the Figure as shown below (Fig.5). Despite their limitations, these lines still offer a visual representation of the data, which can be informative for readers.

Correlation between ASAH2 and T cell CD8+ TIMER in BRCA-LumA

To give users a general idea of how two variable fit, we just applied a linear regression here. If the data is not linear, there might be a discrepancy between the Rho and the slope of the line.

Fig.5. Figure for correlation was automatically generated from TIMER 2.0 website. An introduction for linear regression is shown on the top of Figure.

In our revised manuscript, we have addressed this issue by explicitly stating, "We draw conclusions based on the Spearman correlations, which are robust to outliers and more appropriate for variables with non-linear relationships or those that do not follow a normal distribution (Gauthier, 2001)². The lines overlaid on the plots are based on linear regression models, which assume normal data distribution. In cases of non-normal data distribution, the regression lines may not be suitable" (**Discussion, Page 20**)

We appreciate the Reviewer's feedback and have taken steps to clarify the rationale for including these lines in our figures while emphasizing the robustness of Spearman correlations in our analysis.

b. No handling of zero-inflated data is discussed when applying Spearman's rank test in the analysis.

Response: The robustness of Spearman's correlation in handling non-normally distributed data and outliers is widely recognized. This robustness extends to cases where data may contain a significant proportion of zeros, where average ranks are assigned to tied observations. Alternatively, a hurdle model could be applied to handle zero-inflated data. A hurdle model essentially consists of two components: one component models the mean response when the x variable is zero, and the other component models the relationship between the response variable y and the non-zero values of the x variable.

It is important to note that the data and algorithms utilized in this study were provided and set by the TIMER2.0 web developer. The options for data transformation and addressing zero-inflated data were not

available within this context. We have addressed this limitation in our paper by explicitly stating, “The plots in Figure 1, Panels D, E, and G, were generated using the TIMER2.0 web server, which may not incorporate state-of-the-art methods for handling zero-inflated data at this time” (Discussion, Page 20).

Timer 2.0 is a powerful tool for estimating associations between immune infiltrates and genetic features in the TCGA cohorts. However, these associations require validating using clinic data and animal models. We explored such associations in various animal breast cancer models. We removed the data for LIHC and LUAD tumors in Figure 1G. We would also prefer to remove all of the data generated from Timer 2.0 if it is deemed that the concerns are not well addressed.

c. TIMER 2.0, the analysis tool used, incorporates by default 6 parallel RNAseq deconvolution strategies. These may not be viewed as validations of each other, as the authors propose, but merely complementary and not independent. quanTIseq, the method mentioned here as independent, is included by default in TIMER 2.0. If the authors believe that using parallel deconvolution strategies is a form of validation, they should discuss the abundance of immune cells predicted by each algorithm included in TIMER 2.0 with respect to NcDase expression.

Response: We understand that six algorithms in TIMER2.0 are complementary and not independent. We replaced the word “validated” to “showed”. QUANTISEQ analysis also showed NcDase association with macrophages and CD8+ T cells infiltration in uveal melanoma (UVM) (Figure 1E).

TIMER2.0 utilizes a particular R package that integrates six algorithms for estimating specific cell types within mixed populations including xCell, MCP-counter, CIBERSORT, EPIC, and quanTIseq. In the second revision, we cited and discussed that “xCell can make estimations on the higher number of different immune cell types but may fail to detect signals from homogeneous samples. CIBERSORT deconvolves more detailed subsets of T-cell signatures. EPIC and quanTIseq have the advantage of directly generating scores interpreted as cell fractions” (Discussion, Page 20).

d. Even taking the interpretation of RNAseq deconvolution at face value, the evidence presented is contradictory and weak. NcDase expression does not predict survival outcomes, and no difference was seen between normal and malignant tissue (Fig. S1B-D). **e.** “weak correlation” is often used for correlation coefficient values between 0.25 and 0.5 that are significant. That is not the case in the results presented (Fig. S1B) which are neither significant nor in that range.

Response: We agree with the reviewer’s point. In our manuscript, the description “weak correlation” is incorrect. Fig. S1B-D should indicate no correlation for other immune populations. We revised the comment with “was not correlated with”. In Fig. 1D, “highly correlated with CD8+ T cells and macrophages” was changed to “correlated with CD8+ T cells and macrophages”.

2. The authors did not sufficiently improve the statistical rigor in their analysis. There is no multiple testing correction mentioned in their methods or figure legends. ANOVA is mentioned once in the PCR analysis section. A non-exhaustive list of analysis that is in need of an ANOVA that was not performed includes Fig. 2C,E,F,G, Fig. 4A-F, Fig. 5B, E, G, J...

Response: We revised the statistical analysis in the Figures and Figure legends mentioned by this reviewer.

3. The statistical test in Fig. 7A does not reflect the distribution of values presented. The authors should submit the raw individual values for independent verification. However, as stated before, the overall handling of data in this manuscript greatly diminishes enthusiasm for the findings.

Response: Since Reviewer 5 suggested TREM2-deficient mice as controls, we provided the control data from TREM2-deficient mouse and adjusted the positive line for Trem2 at 10^4 (Fig. 6, **Figure 7A in revision**). Thus, the percentages for all of mice were recalculated accordingly and new individual values were provided. To enhance viewing the data, we also chose the dot lot format of FACS for presenting figures (equal number 40,000 were shown, **Fig.6.**)

WT	NcDase ^{cKO}
20.2	34.3
16.5	30.4
23.2	25.6
25	35.2
21.2	34.1

Fig.6. In main Figure7A of second revision, a control line from Trem2KO was set for Trem2 positive line. We also chose Dot format for FACS figures (bottom).

4. The additional quantification associated with imaging results is welcomed. The lack of statistics on Fig. 3C however, dampens that enthusiasm.

Response: We added and revised the statistical analysis in the Figure and Figure legend.

Reviewer #5 (Remarks to the Author):

The authors answered most of the questions raised by the reviewer. However, there are still some concerns that would still need to be addressed.

1) The fact that there is no correlation with clinical parameters in human datasets is concerning and weakens the relevance of the findings. I understand that the paper is focused on breast cancer models, but the authors might look at different cohorts too. If there is any correlation in different tumor types, they might at least mention that.

Response: We did look at different cohorts, particularly THYM, KICH and UVM tumor, using different algorithms in Figure 1G-1E. From our viewpoint, NcDase is highly associated with macrophages in THYM and UVM.

2) The CyToF analysis performed in cKO (Figure 6) mice shows a different pattern of myeloid cell populations compared to KO (Figure 2) mice. I would suggest explaining and discussing this difference.

Response: The difference was caused by the different markers (antibodies) for detecting myeloid cells in cKO mice. Another important reason is different tumor models. Figure 2 was generated from MMTV spontaneous breast cancer. cKO in Figure 6 was from the E0771 syngenic tumor model. We also responded to similar concerns from Reviewer #4 about different profiles for CD39 and Ly6C expression of CD8 T cells in the first revision. We acknowledged the different patterns of immune population in two tumor models in the second revision (**Discussion, Page 21**).

3) The authors show the levels of ceramide in TAMs by flow cytometry. I couldn't find any method section explaining this flow staining. Also, the number of cells look different in the two plots. Is there a general reduction of macrophages in KO mice?

Response: Anti-ceramide antibody³ is from Sigma (original antibody with same clone was published in JCI: PMC3336980). We cited this paper in the first revision and added the staining methods in the second revision. The reason the two plots looked different is because the total cells number collected by FACS are different. However, the total percentage of CD11b was not reduced (**Fig.7.**).

Fig.7. The percentage of CD11b in TILs from WT and NcDase^{cKO} tumor-bearing mice.

[editorial note: please note that Reviewer #5 has raised a few additional concerns in their confidential remarks to the Editors: - Related to the authors' claim that ceramide inhibits TREM2 shedding resulting in higher TREM2 expression, it does not explain the increased mRNA expression shown in Fig. 7B.]

Response: Our data discovered one mechanistic links with TREM2 expression and discussed the other possible mechanism. For examples, transcription factor YY1 can affect Trem2 mRNA expression; this result was recently published in JBC during our revision (PMCID: PMC10193014)⁴. We added" *We acknowledge that ceramide might be involved in the prevention of the full-length TREM2 cleavage through ADAM17, which could be one of the mechanistic links with TREM2 expression since a recent study showed that transcription factor YY1 can regulate TREM2 mRNA expression. Therefore, other mechanism(s) linking ceramide with TREM2 mRNA expression could be explored in the future*" (**Discussion, Page 23**).

- staining of TREM2 in TAMs (Fig.7A): TREM2-deficient mouse as a control and more representative plots would be required to confirm the quality of the staining.

Response: We provided the control data from TREM2-deficient mouse and adjusted the positive line for Trem2 to 10⁴. To enhance viewing the data, we also chose the dot plot format for presenting figures (40,000 cells are shown in each plot.)

Fig.8: In main Figure7A of revision, a control line from Trem2KO was set forTrem2 positive line. We also chose Dot format for FACS figures (bottom).

1. Li T, Fu J, Zeng Z, Cohen D, Li J, Chen Q, Li B and Liu XS. TIMER2.0 for analysis of tumor-infiltrating immune cells. *Nucleic Acids Res.* 2020;48:W509-W514.
2. Gauthier TD. Detecting trends using Spearman's rank correlation coefficient. *Environ Forensics.* 2001;2:359-362.
3. Rotolo J, Stancevic B, Zhang JJ, Hua GQ, Fuller J, Yin XL, Haimovitz-Friedman A, Kim K, Qian M, Cardo-Vila M, Fuks Z, Pasqualini R, Arap W and Kolesnick R. Anti-ceramide antibody prevents the radiation gastrointestinal syndrome in mice. *Journal of Clinical Investigation.* 2012;122:1786-1790.
4. Lu Y, Huang X, Liang W, Li Y, Xing M, Pan W, Zhang Y, Wang Z and Song W. Regulation of TREM2 expression by transcription factor YY1 and its protective effect against Alzheimer's disease. *J Biol Chem.* 2023;299:104688.

REVIEWERS' COMMENTS

Reviewer #1 (Remarks to the Author):

In the present study 'Neutral ceramidase regulates breast cancer progression by metabolic programming TREM2-associated macrophages' all queries raised are answered now and therefore is recommended for publication.

Reviewer #2 (Remarks to the Author):

The questions raised before were addressed by the authors.

Reviewer #4 (Remarks to the Author):

The authors have addressed my concerns regarding statistical rigour. I would advise removal of the TIMER 2.0 data, as the authors state that they have no control over the graphical output or statistical methods employed in the analysis and are solely relying on a website for the interpretation and generation of these plots. Critically, NcDase expression does not predict survival outcomes, and no difference was seen between normal and malignant tissue. The clinical significance of these findings is unclear.

Reviewer #5 (Remarks to the Author):

The authors addressed all my questions.

Reviewer #4.

Comments: The authors have addressed my concerns regarding statistical rigor. I would advise removal of the TIMER 2.0 data, as the authors state that they have no control over the graphical output or statistical methods employed in the analysis and are solely relying on a website for the interpretation and generation of these plots. Critically, NcDase expression does not predict survival outcomes, and no difference was seen between normal and malignant tissue. The clinical significance of these findings is unclear.

Response: Most of data from TIMER 2.0 in Figure 1 were removed according to the suggestions. We would like to keep the piece of data about the association of NcDase (*ASAH2*) expression in Breast cancer Luminal A subtype in Fig 1d. *ASAH2* expression was correlated with CD8⁺ T cells ($Rho=0.237$, $p=5.07e-08$) and macrophages ($Rho=0.196$, $p=7.31e-06$) in the lumA subtype. From our view, p value for the correlation is very significant.

In discussion, we state “However, we should note that the plots in Figure 1d were generated using the TIMER2.0 web server, which may not incorporate state-of-the-art methods for handling zero-inflated data at this time. Thus, this data provides the limited data supporting the clinical relevance, however, it could provide the prediction for the association of *ASAH2* expression with breast cancer. We validated the spatial colocalization between NcDase-related macrophage infiltration and CD8⁺ T cell exhaustion in breast cancer model in mice.”